# MIRO: MultI-Reward cOnditioned pretraining improves T2I quality and efficiency

**Nicolas Dufour** [1 2 3] **Lucas Degeorge** [* 1 2 4] **Arijit Ghosh** [* 1] **Vicky Kalogeiton** [† 2] **David Picard** [† 1]

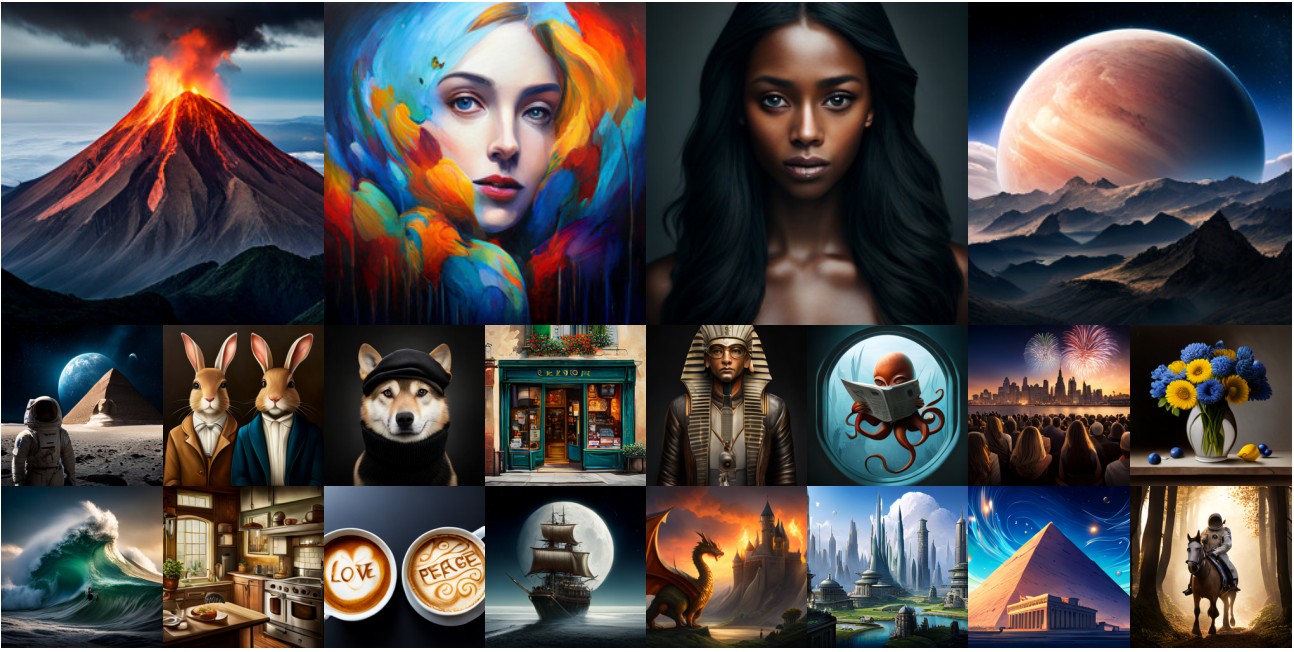

*Figure 1.* Images from our MIRO Synth model on PartiPrompt(Yu et al., 2022).

## Abstract

The default paradigm of post-training text-to-image generators includes post-hoc selection of generated images, and subsequent training with one reward model to align the generator to the reward, typically user preference. This discards informative data as well as optimizes only for a single reward, hence harming diversity, semantic fidelity and efficiency. Instead, we propose **MIRO**, a method that conditions the model on multiple rewards during training, thus letting the model learn user preferences directly. **MIRO** pre-training both improves the visual quality of the generated images and speeds up the training, achieving state of the art on the GenEval compositional benchmark and user-preference scores (PickAScore, ImageReward, HPSv2). Code and weights available here.

## 1. Introduction

A modern text-to-image generator must learn two things: what an image looks like, and what a user wants from one. Following RLHF's success in language models (Christiano et al., 2017; Rafailov et al., 2023) and image generation (Fan et al., 2023), today's best systems pursue both in three training stages: (1) large-scale pretraining on noisy web data, (2) supervised finetuning on a curated high-quality subset, and (3) reinforcement learning from human feedback (RLHF) (Esser et al., 2024; Labs, 2024). Each stage targets a different objective: pretraining the data distribution, finetuning visual quality, and RLHF user preference.

---
[*]Co-second authors. [†]Co-last authors. [1]LIGM, ENPC, IP Paris, CNRS, UGE, France [2]LIX, CNRS, École Polytechnique, IP Paris, France [3]Kyutai [4]AMIAD. Correspondence to: Nicolas Dufour <nicolas.dufour@kyutai.org>.

*Proceedings of the 43rd International Conference on Machine Learning*, Seoul, South Korea. PMLR 306, 2026. Copyright 2026 by the author(s).

Yet each stage pays its own price. **Pretraining** maximizes likelihood over the web's distribution, a target decoupled from what users actually want. **Finetuning** narrows that distribution by discarding informative "low-quality" data (Dufour et al., 2024), sacrificing the signal that taught the model the structure of natural images. **RLHF** adds a separate optimization stage that further collapses the distribution onto a single scalar reward, harming diversity (mode collapse) and semantic fidelity, and locking the reward trade-off at training time, beyond the user's reach at inference. These costs compound: each stage contracts the distribution shaped by the previous, with no way back — pretraining onto the web's bias, finetuning onto the curator's taste, RLHF onto a single reward. The user inherits whatever operating point the trainers picked, with no recourse at inference.

We propose to align *multiple rewards during* pretraining. Each training image is annotated with seven complementary rewards (aesthetics, user preference, text-image alignment, visual reasoning, and scientific correctness), and the generator is conditioned on the resulting reward vector, learning $p(x \mid c, \mathbf{s})$ over images, captions, and *quality*. We call this framework **MIRO** (**M**ult**I**-**R**eward c**O**nditioning); Figure 1 shows it covers the entire reward spectrum without filtering or RLHF.

Three properties follow. First, every training image contributes signal at its own reward level, preserving the full data distribution rather than filtering it. Second, the reward vector becomes a controllable input at inference; users dial each reward, and a multi-reward extension of classifier-free guidance (Ho & Salimans, 2022) steers generation toward jointly-high-reward regions. Third, dense multi-dimensional quality supervision accelerates convergence beyond the unconditional objective.

On a 16M-image setup, a 0.36B-parameter **MIRO** model converges up to $19\times$ faster than its baseline on aesthetic and preference metrics, mitigates reward hacking, and improves compositional alignment. On GenEval (Ghosh et al., 2024) and PartiPrompts (Yu et al., 2022), **MIRO** surpasses FLUX-dev (12B) at $370\times$ less training compute, and tops ImageReward (Xu et al., 2023), HPSv2 (Wu et al., 2023), and PickScore (Kirstain et al., 2023) with $3\times$ less inference compute under sample-based scaling.

Our contributions are:

- **MIRO**, a multi-reward conditioned pretraining framework that integrates alignment directly into training, eliminating filtering and post-hoc RLHF;
- state-of-the-art results on GenEval and user-preference benchmarks (PickScore, ImageReward, HPSv2), with a 0.36B-parameter model surpassing FLUX-dev (12B) at $370\times$ less training compute;
- up to $19\times$ faster convergence than baseline pretrain-

ing and $3\times$ less inference compute than FLUX under sample-based scaling.

## 2. Method

We introduce **MultI-Reward cOnditioning Pretraining (MIRO)**, a framework for conditional image generation that incorporates multiple reward signals directly into the pretraining phase (see Figure 8 in the Supplementary Material for a detailed diagram). Our key insight is that by conditioning the generative model on explicit reward scores during training, we can preserve the full spectrum of quality levels while enabling fine-grained control over multiple objectives at inference time. This approach eliminates the need for separate alignment stages (Fan et al., 2023) while providing unprecedented flexibility in reward trade-offs.

**Method Overview**  Our method consists of three key components: (1) **Dataset Augmentation**, where we enrich the pretraining dataset with reward annotations across multiple quality dimensions; (2) **Multi-Reward Conditioned Training**, where we modify the flow matching objective to incorporate reward signals directly into the generative process; and (3) **Reward-Guided Inference** where we enable fine-grained control through explicit reward conditioning during sampling.

**Problem Formulation**  Let $\mathcal{D} = \{(x^{(i)}, c^{(i)})\}_{i=1}^{M}$ be a large-scale pretraining dataset where $x^{(i)} \in \mathbb{R}^{H \times W \times 3}$ an image and $c^{(i)} \in \mathcal{T}$ the corresponding text condition (e.g., caption, prompt). Traditional pretraining learns a generative model $p_\theta(x|c)$ that captures the joint distribution of images and text without explicit quality control.

In contrast, we consider a set of $N$ reward models $\mathcal{R} = \{r_1, r_2, \ldots, r_N\}$ where each $r_j : \mathbb{R}^{H \times W \times 3} \times \mathcal{T} \to \mathbb{R}$ evaluates different aspects of image quality, with $\mathcal{T}$ being the associated conditioning space.

Our goal is to learn a conditional generative model $p_\theta(x|c, \mathbf{s})$ where $\mathbf{s} = [s_1, s_2, \ldots, s_N]$ represents the desired reward levels, enabling control across multiple quality dimensions.

### 2.1. Dataset Augmentation with Reward Scores

The first step of MIRO involves augmenting the pretraining dataset with comprehensive reward annotations. For each sample $(x^{(i)}, c^{(i)}) \in \mathcal{D}$, we compute reward scores across all $N$ reward models:

$$s_j^{(i)} = r_j(x^{(i)}, c^{(i)}) \quad \forall j \in \{1, 2, \ldots, N\} \tag{1}$$

This process transforms our dataset into an enriched version $\tilde{\mathcal{D}} = \{(x^{(i)}, c^{(i)}, \mathbf{s}^{(i)})\}_{i=1}^{M}$ where $\mathbf{s}^{(i)} = [s_1^{(i)}, s_2^{(i)}, \ldots, s_N^{(i)}]$ contains the multi-dimensional quality assessment for each sample.

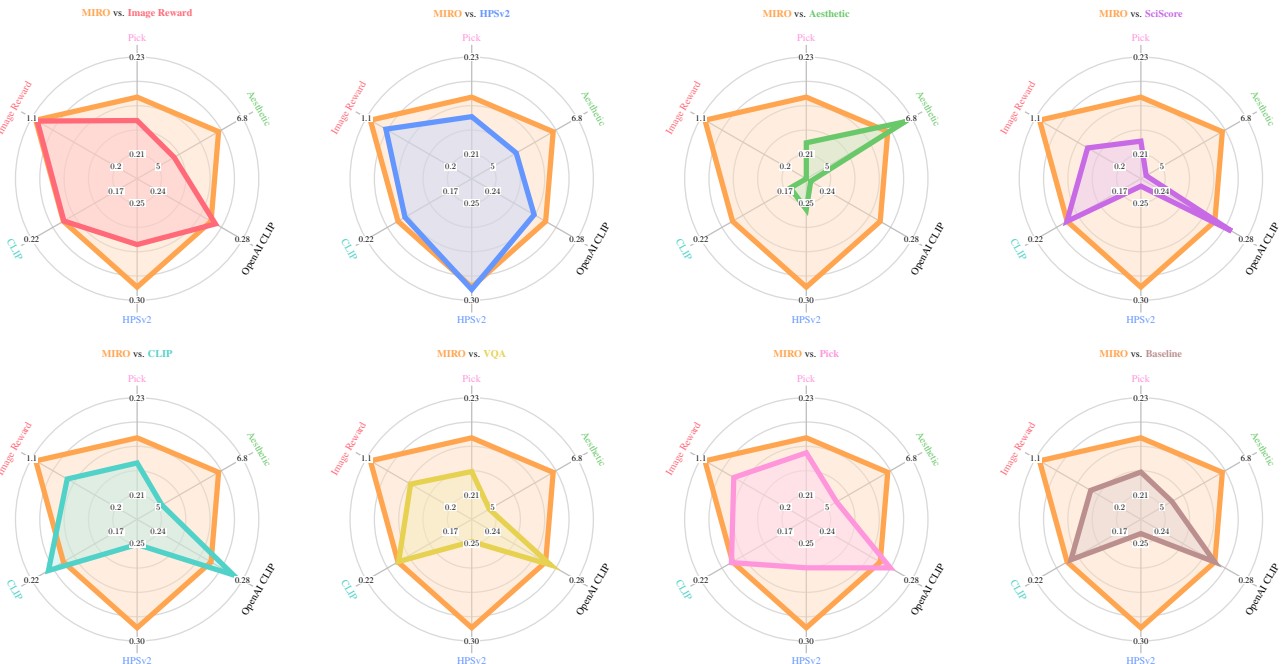

*Figure 2.* Comparison of the MIRO model against eight other specialist/baseline models. Each radar plot shows **MIRO** versus a comparison model across six metrics.

**Score Normalization and Binning.** Raw reward scores often exhibit different scales and distributions across reward models, making direct conditioning challenging. To address this, we employ a uniform binning strategy into $B$ bins that ensures balanced representation across quality levels. Details are found in the Supplementary Material.

### 2.2. Multi-Reward Conditioned Flow Matching

Having augmented our dataset with reward scores, we now incorporate these signals into the generative model architecture. We build upon flow matching (Lipman et al., 2023), a powerful framework for training continuous normalizing flows that has shown excellent performance in high-resolution image generation.

**Training Objective.** Following the standard flow matching formulation, we sample noise $\epsilon \sim \mathcal{N}(0, I)$ and time $t \sim \mathcal{U}(0, 1)$, then compute the noisy sample $x_t = (1 - t)x + t\epsilon$. The multi-reward flow matching loss becomes:

$$\mathcal{L} = \mathbb{E}_{\substack{(x,c,\hat{s}) \sim \tilde{\mathcal{D}}, \epsilon \sim \mathcal{N}(0,I) \\ t \sim \mathcal{U}(0,1)}} \left[ \|v_\theta(x_t, c, \hat{s}) - (\epsilon - x)\|_2^2 \right] \quad (2)$$

This objective trains the model to predict the difference between the noise and the clean image, conditioned on both the text prompt and the desired quality levels. The model learns to associate different reward levels with corresponding visual characteristics, enabling reward-aware generation.

**Training Dynamics.** During training, the model observes

the full spectrum of quality levels for each reward dimension. This exposure allows it to learn the relationship between reward values and visual features, from low-quality samples that exhibit artifacts or poor composition to high-quality samples with superior aesthetics and text alignment.

### 2.3. Inference with Reward-Guided Sampling

At inference time, MIRO provides unprecedented control over the generation process through explicit reward conditioning. This section details the various sampling strategies enabled by our approach.

**High-Quality Generation.** For generating high-quality samples, we condition the model on maximum reward values across all N dimensions: $\hat{s}_{\max} = [B-1, B-1, \ldots, B-1]$. This instructs the model to generate samples that maximize all reward objectives simultaneously.

**Multi-Reward Classifier-Free Guidance.** We extend classifier-free guidance to the multi-reward setting by leveraging the reward conditioning mechanism. Following the Coherence-Aware CFG approach (Dufour et al., 2024), we compute guidance using the contrast between a positive direction and a negative direction in the reward space. We introduce a *positive* and a *negative* reward target, denoted $\hat{s}^+$ and $\hat{s}^-$, which can be chosen by the user for controllability. By default, we use $\hat{s}^+ = \hat{s}_{\max} = [B-1, \ldots, B-1]$ and $\hat{s}^- = \hat{s}_{\min} = [0, \ldots, 0]$ and $\omega$ is the guidance scale:

$$\hat{v}_\theta(x_t, c) = (1+\omega)v_\theta(x_t, c, \hat{\mathbf{s}}^+) - \omega\, v_\theta(x_t, c, \hat{\mathbf{s}}^-) \quad (3)$$

A detailed diagram of the inference process is provided in Figure 9 in the Supplementary Material.

**Theoretical Interpretation.** We now show that this guidance formulation has a principled interpretation as sampling from a reward-tilted distribution.

**Theorem 2.1** (Guidance as Reward-Tilted Sampling). *The MIRO guidance formula corresponds to sampling from a reward-tilted distribution:*

$$p_\omega(x|c) \propto p(x|c, \mathbf{s}^+)\left(\frac{p(\mathbf{s}^+|x,c)}{p(\mathbf{s}^-|x,c)}\right)^\omega \quad (4)$$

*where $\omega \geq 0$ controls the strength of reward guidance. The velocity difference $v_\theta(x_t, c, \mathbf{s}^+) - v_\theta(x_t, c, \mathbf{s}^-)$ approximates the gradient of the log-odds ratio $\nabla_{x_t} \log \frac{p(\mathbf{s}^+|x_t,c)}{p(\mathbf{s}^-|x_t,c)}$. See Appendix G for the proof.*

**Interpretation.** When $\omega = 0$, we sample from the high-reward conditional $p(x|c, \mathbf{s}^+)$. As $\omega$ increases, samples concentrate on regions where the log-odds ratio is maximized—i.e., where high rewards are most likely relative to low rewards. Under conditional independence of rewards, the log-odds decomposes into a sum over individual rewards, naturally steering generation toward the Pareto frontier rather than collapsing to extremes of any single reward (Karras et al., 2024).

**Flexible Reward Trade-offs.** A key advantage of MIRO is the ability to specify custom reward targets at inference time. Users can set $\hat{\mathbf{s}}_{\text{custom}} = [\hat{s}_1, \hat{s}_2, \ldots, \hat{s}_N]$ where each represents the desired level for reward $j$ for image $i$. This enables control over trade-offs between different quality or preference aspects.

## 2.4. Advantages of MIRO over traditional alignment approaches

MIRO offers several key advantages over traditional alignment approaches, stemming from its unified training paradigm and explicit reward conditioning mechanism.

**Training Efficiency.** By incorporating reward alignment directly into pretraining, MIRO eliminates the need for separate fine-tuning or reinforcement learning stages. MIRO converges to reward-aligned behavior without additional training phases achieving faster convergence than regular pretraining and higher quality samples. The single-stage training also reduces the complexity of the training pipeline and eliminates hyperparameter tuning for multiple stages.

**Full-Spectrum Data Utilization and Diversity Preservation.** Unlike post-hoc fine-tuning and RL pipelines that

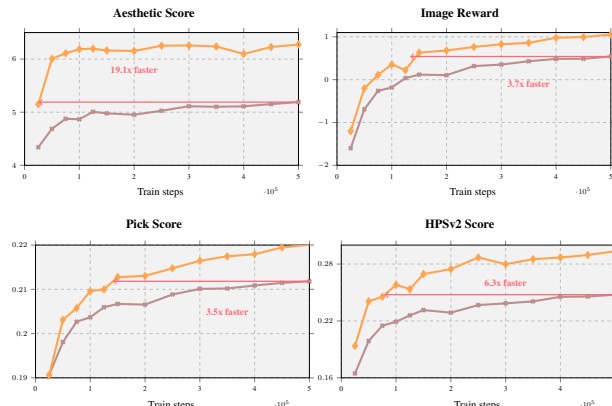

*Figure 3.* Training curves showing reward evolution during training. × Baseline, ◇ MIRO.

concentrate training on a narrow slice of high-reward data, MIRO retains every sample and trains across the entire reward spectrum, with the following theoretical guarantees:

**Theorem 2.2** (MIRO Diversity Preservation). *MIRO's supervised objective preserves the full data distribution while enabling controllable generation:*

1. *(**Full Spectrum Coverage**) MIRO learns $p_\theta(x|c, \mathbf{s}) \approx p_{\text{data}}(x|c, \mathbf{s})$ for all reward levels $\mathbf{s}$, and the marginal $\sum_{\mathbf{s}} p(\mathbf{s}|c)p_\theta(x|c, \mathbf{s})$ recovers the full data distribution.*

2. *(**Entropy Preservation**) The marginal entropy equals the data entropy: $\mathcal{H}(p_\theta^{\text{marginal}}) = \mathcal{H}(p_{\text{data}})$.*

3. *(**Controllable Diversity**) Users can select any point on the diversity-quality spectrum by choosing the conditioning $\mathbf{s}$ at inference time.*

*See Appendix H for the proof.*

Each example contributes signal together with its associated reward vector, so low-, medium-, and high-scoring regions are all modeled. This spectrum-wide supervision reduces collapse toward narrow high-reward modes, yields representations that generalize across quality levels, and produces a single model that can intentionally generate at any desired reward level at inference time.

**Reward Hacking Prevention.** Traditional single-objective optimization often leads to reward hacking, where models exploit specific reward metrics at the expense of overall quality (Luo et al., 2026). MIRO's multi-dimensional conditioning naturally prevents this by requiring the model to balance multiple objectives simultaneously. Users can detect and mitigate reward hacking by adjusting individual reward levels and observing the resulting trade-offs.

**Controllability and Interpretability.** The explicit reward conditioning gives interpretable control over generation quality. Users can predict the effect of different reward

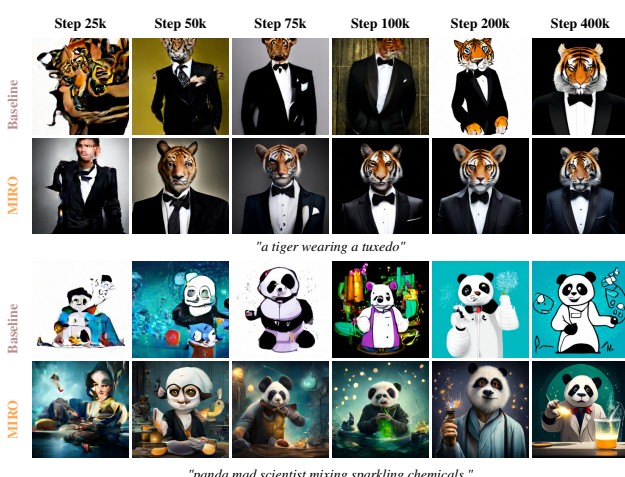

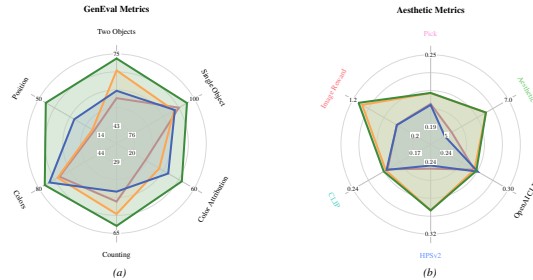

*Figure 5.* MIRO vs baseline trained with real vs synthetic captions on GenEval and Aesthetic metrics. Legend: **Baseline** (real), **MIRO** (real), **Synth Baseline** (50% real + 50% synth), **Synth MIRO** (50% real + 50% synth).

*Figure 4.* Training progression visualization showing generated images at different training steps for the same prompt. Top row shows baseline, bottom row shows MIRO.

settings, enabling intuitive interaction with the model and nuanced trade-offs between visual quality aspects.

## 3. Experiments

### 3.1. Reward-Conditioned Pretraining Improves Model Quality

We demonstrate that pretraining with MIRO produces superior models compared to traditional approaches. We evaluate three training configurations: (1) a baseline model trained without reward conditioning, (2) single-reward models conditioned on individual rewards (similar to Coherence Aware Diffusion (Dufour et al., 2024) but using our reward suite instead of CLIP score), and (3) MIRO conditioned on all seven rewards simultaneously.

**MIRO outperforms single-reward approaches across all metrics.** Figure 2 presents results on the CC12M+LA6 dataset, evaluating models across AestheticScore, PickScore, ImageReward, HPSv2, and JINA CLIP score. We also include OpenAI CLIP score as an out-of-distribution evaluation metric not used during training. MIRO consistently outperforms all baselines across aesthetic and preference metrics, demonstrating the effectiveness of multi-reward conditioning.

**Multi-reward conditioning mitigates reward hacking.** Crucially, we observe that leveraging multiple rewards mitigates reward hacking compared to single-reward optimization. This is particularly evident with AestheticScore: while the single-reward model achieves high aesthetic scores, it severely degrades performance on other metrics. Models trained on ImageReward and HPSv2 show more balanced trade-offs but still underperform MIRO's comprehensive optimization.

**MIRO dramatically accelerates training convergence.** Figure 3 reveals substantial training efficiency gains from multi-reward conditioning. MIRO reaches the baseline model's final performance dramatically faster: 19× speedup for AestheticScore, 6.2× for HPSv2, 3.5× for PickScore, and 3.3× for ImageReward. This acceleration occurs because reward conditioning provides dense supervisory signals throughout training that guide the model toward high-quality generations, rather than requiring the model to discover these qualities through the diffusion objective alone.

**Qualitative results confirm accelerated high-quality generation.** Figure 4 provides qualitative evidence of MIRO's accelerated convergence. For the "tiger in a tuxedo" prompt, MIRO establishes proper compositional layout and generates a visually appealing tiger within 50k training steps, a level of quality that requires 200k steps for the baseline model to achieve. Similarly, for the "mad scientist panda" prompt, MIRO rapidly converges to aesthetically pleasing results while the baseline model fails to generate a recognizable panda until 400k steps. These qualitative improvements complement our quantitative findings, demonstrating that MIRO's multi-reward conditioning enables both faster convergence and superior generation quality.

### 3.2. Improving Text-Image Alignment

Beyond optimizing for specific reward metrics, MIRO demonstrates significant improvements in text-image alignment, measured by comprehensive evaluation benchmarks. Table 1 presents results on GenEval, comparing MIRO against baseline models and single-reward approaches.

**MIRO enhances compositional understanding.** Our multi-reward approach substantially improves the model's ability to generate images that accurately reflect textual descriptions. MIRO achieves an overall GenEval score of 57, representing a 9.6% improvement over the baseline score of 52. This enhancement is particularly pronounced in challenging compositional reasoning tasks: Color Attribution improves from 29 to 38 (+31%), Two Objects from 55 to 68

| Model | Params (B) | Inference TFLOPs | GenEval | | | | | | | PartiPrompts | | | |
|---|---|---|---|---|---|---|---|---|---|---|---|---|---|
| | | | Overall | Single Obj. | Two Obj. | Position | Counting | Colors | Color Attr. | Aesthetic | Image | HPSv2 | PickAScore |
| *SOTA Baselines* | | | | | | | | | | | | | |
| SD v1.5 | 0.9 | - | 43 | 97 | 38 | 4 | 35 | 76 | 6 | 5.68 | 0.24 | 0.25 | 0.213 |
| SD v2.1 | 0.9 | - | 50 | 98 | 51 | 7 | 44 | 85 | 17 | 5.81 | 0.38 | 0.26 | 0.215 |
| PixArt-$\alpha$ | 0.6 | - | 48 | 98 | 50 | 8 | 44 | 80 | 7 | 6.47 | 0.97 | 0.29 | 0.226 |
| PixArt-$\Sigma$ | 0.6 | - | 52 | 98 | 59 | 10 | 50 | 80 | 15 | 6.44 | 1.02 | 0.29 | 0.225 |
| CAD | 0.35 | 20.8 | 50 | 95 | 56 | 11 | 40 | 76 | 22 | 5.56 | 0.69 | 0.26 | 0.214 |
| Sana-0.6B | 0.6 | - | 64 | 99 | 71 | 16 | 63 | 91 | 42 | 6.31 | 1.23 | 0.30 | 0.228 |
| Sana-1.6B | 1.6 | - | 66 | 99 | 79 | 18 | 63 | 88 | 47 | 6.36 | 1.23 | 0.30 | 0.228 |
| SDXL | 2.6 | - | 55 | 98 | 74 | 15 | 39 | 85 | 23 | 5.94 | 0.46 | 0.25 | 0.220 |
| SD3-medium | 2.0 | - | 62 | 98 | 74 | 34 | 63 | 67 | 36 | 6.18 | 1.15 | 0.30 | 0.225 |
| FLUX-dev | 12.0 | 1540 | 67 | 99 | 81 | 20 | 79 | 74 | 47 | 6.56 | 1.19 | 0.30 | **0.229** |
| *CAD-like Models (our models)* | | | | | | | | | | | | | |
| Image Reward | 0.36 | 4.16 | 57 | 97 | 59 | 21 | 56 | 76 | 33 | 5.31 | 1.04 | 0.27 | 0.214 |
| HPSv2 | 0.36 | 4.16 | 56 | 95 | 63 | 15 | 52 | 78 | 31 | 5.47 | 0.90 | 0.29 | 0.215 |
| Aesthetic | 0.36 | 4.16 | 33 | 74 | 37 | 6 | 24 | 42 | 15 | 6.65 | 0.00 | 0.26 | 0.209 |
| SciScore | 0.36 | 4.16 | 58 | 94 | 62 | 24 | 61 | 72 | 35 | 4.62 | 0.56 | 0.24 | 0.209 |
| CLIP | 0.36 | 4.16 | 57 | 97 | 63 | 24 | 57 | 70 | 32 | 5.04 | 0.73 | 0.25 | 0.214 |
| VQA | 0.36 | 4.16 | 57 | 97 | 58 | 20 | 57 | 76 | 37 | 4.88 | 0.64 | 0.25 | 0.212 |
| Pick | 0.36 | 4.16 | 57 | 93 | 62 | 17 | 58 | 75 | 34 | 5.16 | 0.76 | 0.26 | 0.216 |
| *Real Caption Models (our models)* | | | | | | | | | | | | | |
| Baseline | 0.36 | 4.16 | 52 | 94 | 55 | 18 | 49 | 68 | 29 | 5.18 | 0.52 | 0.25 | 0.212 |
| MIRO | 0.36 | 4.16 | 57 | 92 | 68 | 19 | 55 | 69 | 38 | 6.28 | 1.06 | 0.29 | 0.220 |
| *Synthetic Caption Models (50% Real + 50% Synth) (our models)* | | | | | | | | | | | | | |
| Baseline | 0.36 | 4.16 | 57 | 93 | 59 | 30 | 44 | 74 | 43 | 4.96 | 0.52 | 0.24 | 0.211 |
| MIRO | 0.36 | 4.16 | 68 | 97 | 73 | 46 | 61 | 77 | 52 | 6.28 | 1.11 | 0.29 | 0.220 |
| MIRO† | 0.36 | 4.16 | **75** | 98 | 79 | **58** | 71 | 85 | 58 | 5.24 | 1.18 | 0.29 | 0.220 |
| *Inference Scaled + Synthetic Caption Models (MIRO + 128 samples inference scaled) (our models)* | | | | | | | | | | | | | |
| Aesthetic Scaled MIRO | 0.36 | 532 | 63 | 97 | 68 | 40 | 57 | 75 | 45 | **6.81** | 1.04 | 0.29 | 0.219 |
| Image Reward Scaled MIRO | 0.36 | 532 | **75** | 98 | **84** | 52 | 69 | 82 | **65** | 6.28 | **1.61** | 0.30 | 0.223 |
| HPSv2 Scaled MIRO | 0.36 | 532 | 74 | 98 | 83 | 47 | 74 | 80 | **65** | 6.28 | 1.35 | **0.32** | 0.225 |
| PickAScore Scaled MIRO | 0.36 | 532 | 74 | 98 | 83 | 44 | 76 | 81 | 59 | 6.27 | 1.32 | 0.31 | **0.229** |

*Table 1.* GenEval and PartiPrompts Results Comparison Across All Models. Unless noted, inference uses the positive/negative targets $\hat{\mathbf{s}}^+ = [1, 1, \ldots, 1]$ and $\hat{\mathbf{s}}^- = [0, 0, \ldots, 0]$. † denotes a custom positive target with all rewards set to 1 except the aesthetic reward set to 0.625 (i.e., $\hat{s}^+_{\text{aesthetic}} = 0.625$), with $\hat{\mathbf{s}}^-$ fixed.

(+24%), and Counting from 49 to 55 (+12%). These results demonstrate that MIRO's multi-reward conditioning enables better understanding of complex spatial relationships, object interactions, and numerical concepts.

**Single-reward models exhibit varying alignment capabilities.** Our analysis reveals that different reward models contribute differently to text-image alignment. Models optimized solely for aesthetic appeal (AestheticScore) achieve poor GenEval performance (33.0), suggesting that aesthetic optimization can come at the expense of semantic fidelity. In contrast, rewards more directly related to text-image correspondence, such as CLIP score, VQA score, and JINA CLIP score, achieve GenEval scores of 57, matching MIRO's performance. Notably, the SciScore model achieves the highest single-reward GenEval score of 58, though this comes with reduced aesthetic quality as shown in Figure 2.

**Multi-reward conditioning prevents overfitting.** The superior performance of MIRO compared to single-reward models highlights a key advantage of our approach: by optimizing across multiple complementary objectives simultaneously, MIRO avoids the overfitting that occurs when models focus exclusively on a single reward signal. This balanced optimization leads to models that excel across diverse evaluation criteria while maintaining strong performance on individual metrics.

### 3.3. MIRO and Synthetic Captions

Synthetic captioning has emerged as the go-to method for improving text-image alignment in generative models. This approach offers the advantage of retaining all training data without requiring filtering based on caption quality. While CAD (Dufour et al., 2024) proposes a method to avoid filtering, it does not demonstrate results on synthetic captions. We evaluate MIRO using a mixture of 50% synthetic and 50% real captions (see Appendix C).

**Technical implementation.** Applying MIRO to synthetic captions presents a challenge: some reward models cannot process captions longer than 77 tokens, while our synthetic captions are extensive (approximately 200 tokens). To address this limitation, we generate both long captions for training and shorter versions for reward model evaluation.

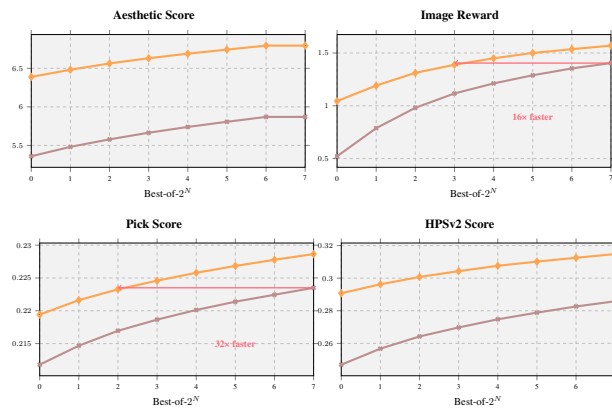

*Figure 6.* Test-time scaling showing performance vs. Best-of-$2^N$ sampling. × Baseline, ◇ MIRO.

**MIRO outperforms synthetic captioning alone.** MIRO without synthetic captions achieves comparable GenEval performance to baselines trained with just synthetic captions. More importantly, Figure 5 shows that MIRO without synthetic captions significantly outperforms the synthetic caption baseline across reward metrics, while being computationally more efficient since reward model scoring is far cheaper than recaptioning with large vision-language models.

**MIRO unlocks synthetic caption potential.** Combining MIRO with synthetic captions yields the strongest overall performance as shown in Table 1. While maintaining equivalent aesthetic quality to MIRO without synthetic captions, this combined approach achieves a remarkable GenEval score of 68, substantially improving over the synthetic caption baseline of 57 (+19%). The improvements are consistent across all compositional reasoning metrics: Position increases from 30 to 46 (+53%), Color Attribution from 43 to 52 (+21%), Single Object from 93 to 97 (+4%), Two Objects from 58 to 73 (+26%), and Counting from 44 to 61 (+39%). These comprehensive gains across all compositional aspects demonstrate that MIRO effectively benefits massively from synthetic captions for text-image alignment, achieving superior compositional understanding while preserving aesthetic quality.

### 3.4. Synergizing with Test-Time Scaling

Test-time scaling has emerged as a popular method to improve reward performance by generating multiple samples and selecting the best one (Ma et al., 2025). We demonstrate that MIRO achieves superior sample efficiency compared to baseline models when combined with test-time scaling.

**Experimental setup.** We evaluate both baseline and MIRO models using the Random Search protocol from (Ma et al., 2025). Figure 6 presents performance across varying sample counts (1 to 128 samples, displayed on a log-2 scale).

For each evaluation, we generate N samples and select the highest-scoring sample according to the respective reward model.

**MIRO demonstrates superior sample efficiency.** Our results reveal that MIRO consistently outperforms the baseline across all reward metrics, often by substantial margins. Most remarkably, for Aesthetic Score and HPSv2 metrics, MIRO achieves with a single sample what the baseline cannot reach even with 128 samples. This dramatic efficiency gain highlights MIRO's ability to generate high-quality samples without requiring extensive test-time computation.

**Quantifying inference-time efficiency improvements.** The gains are particularly striking for specific metrics: for ImageReward, MIRO with 8 samples matches the baseline with 128 samples, a 16× efficiency improvement. For PickScore, MIRO matches the baseline's 128 samples with only 4, a remarkable 32× efficiency gain, establishing MIRO as both a superior training approach and a more efficient inference-time method.

### 3.5. Comparison to State-of-the-Art Models

In Table 1, we evaluate MIRO against state-of-the-art text-to-image models on GenEval, showing superior performance while having significantly lower computational costs.

**GenEval results demonstrate exceptional training efficiency.** MIRO achieves a GenEval score of 68, outperforming FLUX-dev (12B parameters) which scores 67, while requiring dramatically less computation: 4.16 TFLOPs vs 1540 TFLOPs for FLUX-dev, representing a remarkable 370× efficiency improvement. This demonstrates that MIRO's multi-reward conditioning enables compact models to surpass much larger architectures.

**MIRO sets new benchmarks for compositional reasoning.** MIRO excels on challenging compositional metrics that have been challenging for text-to-image models. On the Position metric, MIRO achieves a score of 46, improving upon the previous state-of-the-art of 34 (SD3-Medium) by 31%. For Color Attribution, MIRO advances from FLUX-dev's previous best of 47 to 52 (+11%).

**User preference evaluation confirms scalable efficiency.** On PartiPrompts, MIRO consistently outperforms larger models across multiple reward metrics, leveraging inference time scaling. When optimizing for Aesthetic Score with 128-sample inference scaling, MIRO achieves a state-of-the-art score of 6.81 compared to FLUX-dev's 6.56. For ImageReward optimization, MIRO scores 1.61 versus Sana-1.6B's 1.23. Remarkably, even with this 128-sample inference scaling strategy, MIRO maintains a 3× efficiency advantage over FLUX-dev (532 TFLOPs vs 1540 TFLOPs) while achieving superior performance across all metrics.

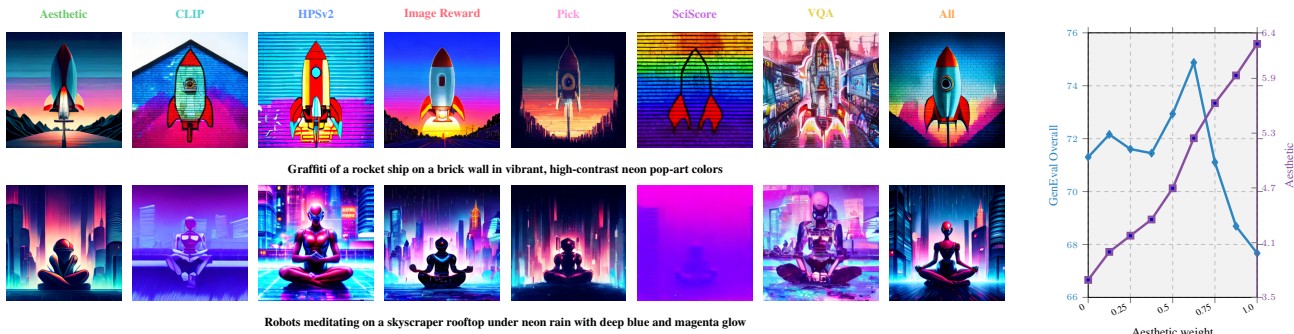

*Figure 7.* **Left: Generations from the Synth MIRO model using multi-reward classifier-free guidance** (see notation in Section 2.3). For each column $j$, we sample with a positive target $\hat{\mathbf{s}}^+ = [1, \dots, 1]$ and a negative target $\hat{\mathbf{s}}^- = [1, \dots, 1]$ except $\hat{s}_j^- = 0$. The guidance vector points purely toward reward $j$. The "All" column uses $\hat{\mathbf{s}}^- = \mathbf{0}$, guiding toward simultaneously high values for all rewards. **Right: Trading off GenEval and Aesthetic scores** by varying $\hat{s}_{\text{aesthetic}}^+$ while keeping other components of $\hat{\mathbf{s}}^+$ equal to 1.

**Multi-reward conditioning enables cross-metric generalization.** Notably, MIRO demonstrates strong performance even when not explicitly optimized for specific metrics. For instance, when optimizing for HPSv2, MIRO achieves an ImageReward score of 1.35, outperforming models specifically trained for that metric. This cross-metric generalization highlights the robustness of MIRO's multi-reward approach and its ability to achieve state-of-the-art results with substantially reduced computational requirements.

### 3.6. Flexible Reward Trade-offs at Inference

**Reward weighting exposes controllable trade-offs between aesthetics and alignment.** Our test-time scaling (TTS) results (Figure 6) show that selecting samples by Aesthetic Score can reduce GenEval performance, indicating a trade-off between aesthetic quality and semantic alignment.

**Sweeping the aesthetic weight identifies an optimal balance.** We vary the aesthetic reward weight at inference and observe the highest GenEval score at a weight of 0.625 (Figure 13). A detailed analysis of aesthetic weight sensitivity across all metrics is provided in Appendix E.3.

**Optimized weighting rivals heavy test-time scaling.** Using this inference strategy, MIRO$^\dagger$ match the GenEval performance of ImageReward-based selection with 128-sample test-time scaling, while using a single weighted selection. Other metrics also improve; for example, ImageReward reaches 1.18, matching FLUX-dev without TTS.

**Visualizing per-reward controllability.** In Figure 7, we visualize this controllability with Synth MIRO using multi-reward classifier-free guidance (Section 2.3). For column $j$, we set $\hat{\mathbf{s}}^+ = [1, \dots, 1]$ and $\hat{\mathbf{s}}^- = [1, \dots, 1]$ with $\hat{s}_j^- = 0$, which cancels the shared direction and isolates reward $j$ while keeping the other rewards anchored to $\hat{\mathbf{s}}^+$.

**Pairwise reward exploration.** To explore the trade-offs between two specific rewards, we perform pairwise interpolation while keeping all other rewards fixed. For rewards $A$ and $B$, we set $\hat{\mathbf{s}}^+ = [1, \dots, 1]$ and $\hat{\mathbf{s}}^- = [1, \dots, 1]$, except for the two rewards of interest: $\hat{s}_A^- = t$ and $\hat{s}_B^- = 1 - t$, where $t \in [0, 1]$ controls the interpolation. This configuration enables smooth exploration of the reward space between two objectives while maintaining high values for all other rewards, revealing the model's ability to navigate trade-offs between specific quality dimensions.

**User-controlled rewards at inference.** MIRO allows choosing reward weights at test time, enabling principled trade-offs across capabilities, giving users control and reducing reward hacking.

**Additional experiments.** We provide comprehensive ablation studies in the Supplementary Material: leave-one-out reward analysis examining the contribution of each reward model (Appendix E.5), binning strategy comparisons evaluating different discretization approaches (Appendix E.6), and post-training efficiency experiments showing that fine-tuning a baseline with MIRO approaches but slightly underperforms full training from scratch (Appendix E.7).

## 4. Conclusion

We presented Multi-Reward cOnditioning (MIRO), a simple pretraining framework that conditions on a vector of reward scores, integrating alignment into training rather than post-hoc. By learning $p(x \mid c, \mathbf{s})$ and exposing reward targets as controllable inputs, MIRO disentangles content from quality, enabling precise control at inference. On a 16M-image setup, MIRO outperforms no-conditioning and single-reward baselines, converges substantially faster, mitigates reward hacking, strengthens compositional alignment, and, despite being much smaller, surpasses FLUX-dev on GenEval and PartiPrompts at a fraction of the compute. We hope this opens a new line of research on exploiting rewards at pretraining.

# Acknowledgements

This work was supported by ANR projects TOSAI ANR-20-IADJ-0009 and sharp ANR-23- PEIA-0008 in the context of the PEPR IA, CIEDS, a Hi!Paris grant, and was granted access to the HPC resources of IDRIS under the allocation 2024-A0171014246 made by GENCI. We would like to thank Alyosha Efros, Tero Karras and Luca Eyring for their helpful comments and Yuanzhi Zhu and Xi Wang for proofreading.

# Impact Statement

### 4.1. Reproducibility Statement

To ensure the reproducibility of our results, we have provided a comprehensive description of the MIRO framework and its implementation. The complete source code for dataset augmentation, pretraining, and the multi-reward inference pipeline will be made available upon publication. Detailed specifications of the model architecture and the flow matching objective are outlined in Section 2 and Appendix B. We utilized publicly available datasets (CC12M and LAION Aesthetics 6+), and the synthetic captioning pipeline is detailed in Appendix C. The exact configurations for the seven reward models, along with the score normalization and binning strategies, are documented in Appendix B. This allows the exact replication of our training environment and experimental baselines.

### 4.2. Ethics Statement

**Data and Privacy** We utilized publicly available datasets: CC12M and LAION Aesthetics 6+. While these datasets are standard in the field, we acknowledge the ongoing community discussions regarding the presence of copyrighted material and private individuals in web-crawled data.

**Bias and Fairness** MIRO conditions generation on reward models such as HPSv2, PickScore, and Aesthetic Score. We caution that these reward models reflect the preferences and biases of the specific user groups that annotated their training data. Optimizing for "high reward" may inadvertently amplify societal biases regarding beauty standards, race, or gender roles. Users should be aware that the "quality" defined by these rewards is subjective and not culturally universal.

**Environmental Impact** A core contribution of this work is efficiency. MIRO converges up to $19\times$ faster than baseline pretraining and requires orders of magnitude less inference compute (e.g., $370\times$ less than FLUX-dev) to achieve comparable quality. We believe this direction significantly contributes to reducing the carbon footprint associated with training and deploying high-quality generative models.

**Misuse** As with all open-domain text-to-image models, there is a risk of generating harmful, offensive, or misleading content. While multi-reward conditioning improves alignment with safe prompts, it does not inherently prevent the generation of malicious content if explicitly prompted.

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

# Supplementary Material Contents

**Overview. Appendix A** reviews related work on diffusion models, flow matching, and reward-based alignment. **Appendix B** details implementation choices including architecture, reward preprocessing, training setup, and baselines. **Appendix C** describes the synthetic captioning pipeline used to enrich training data. **Appendix D** reports computational costs for dataset preprocessing. **Appendix E** presents additional experimental results: CFG sensitivity, single-reward comparisons, aesthetic-weight trade-offs, training dynamics, and ablations on binning and post-training. **Appendix F** shows additional qualitative examples of training progression. **Appendix G** provides the full theoretical framework for MIRO's guidance, proving it samples from a reward-tilted distribution (Theorem G.3) and performs implicit gradient ascent on the log-odds ratio. **Appendix H** formally compares MIRO with RL-based alignment (DDPO), showing DDPO suffers gradient conflicts and mode collapse (Theorem H.6) while MIRO preserves diversity (Theorem H.7).

# A. Related Work

## A.1. Diffusion and Flow-Based Generative Models

Modern text-to-image (T2I) generation is built on diffusion models, which learn to reverse a noise-corruption process to generate samples from complex distributions. Early foundations include denoising diffusion probabilistic models (Sohl-Dickstein et al., 2015; Ho et al., 2020) and score-based generative models (Song & Ermon, 2019; 2020; Song et al., 2021), with Dhariwal & Nichol (2021) demonstrating that diffusion models can surpass GANs in image quality. Flow matching (Lipman et al., 2023) provides an alternative training framework with simpler objectives and improved training dynamics.

Architecture innovations have been equally important. Transformer-based backbones such as DiT (Peebles & Xie, 2023) and SiT (Ma et al., 2024) improve scalability compared to U-Net architectures, while practical training recipes incorporating RMSNorm (Zhang & Sennrich, 2019), GLU activations (Shazeer, 2020), and query-key normalization (Henry et al., 2020) stabilize training at scale.

Beyond diffusion, autoregressive models offer an alternative paradigm: Parti (Yu et al., 2022) demonstrates competitive T2I via sequence modeling, and recent work explores decoder-only LLMs for image generation (Wang et al., 2025). MIRO builds on flow matching with transformer backbones, inheriting their scalability while adding multi-reward conditioning.

## A.2. Conditioning and Controllable Generation

Controllable generation in diffusion models stems from two foundational techniques. *Classifier guidance* (Dhariwal & Nichol, 2021) steers sampling using gradients from an external classifier, trading diversity for quality. *Classifier-free guidance* (CFG) (Ho & Salimans, 2022) eliminates the need for a separate classifier by jointly training conditional and unconditional models, then extrapolating between their predictions at inference. CFG has become the de facto standard for high-quality T2I generation.

Text conditioning is typically implemented via cross-attention between image features and text embeddings from frozen

language models. Latent Diffusion Models (Rombach et al., 2022) popularized this approach in a compressed latent space, while Imagen (Saharia et al., 2022) and DALL-E 2 (Ramesh et al., 2022) scaled it with large language encoders. Beyond text, spatial conditioning through ControlNet (Zhang et al., 2023) enables control via edges, depth, and poses. ELLA (Hu et al., 2024) improves semantic alignment by equipping diffusion models with LLM-based text understanding.

Quality-aware conditioning represents a paradigm shift: rather than filtering low-quality data, models can be conditioned on quality scores. Coherence-aware training (Dufour et al., 2024) conditions on text-image alignment scores, preserving all data while enabling quality control at inference. Karras et al. (2024) show that guiding with a "bad version" of the model itself can improve quality. MIRO extends this paradigm from single quality scores to multiple complementary reward dimensions, enabling fine-grained multi-objective control.

### A.3. Data Efficiency and Training Strategies

Efficient T2I training requires careful consideration of data, architectures, and objectives. Public datasets such as CC12M (Changpinyo et al., 2021), LAION (Schuhmann et al., 2022), YFCC100M (Thomee et al., 2016), and CommonCanvas (Gokaslan et al., 2024) enable reproducible research, while curated subsets like ImageNet (Deng et al., 2009) serve as controlled benchmarks (Degeorge et al., 2025).

Latent-space training dramatically reduces computational cost. Latent Diffusion Models (Rombach et al., 2022) operate in a compressed VAE latent space, and subsequent work such as PixArt-$\alpha$ (Chen et al., 2024b), PixArt-$\sigma$ (Chen et al., 2024a), SDXL (Podell et al., 2024), and SANA (Xie et al., 2024) further optimize efficiency through architecture and training improvements. Matryoshka Diffusion (Gu et al., 2023) enables multi-resolution generation from a single model.

Representation alignment approaches accelerate convergence by leveraging pretrained features. DiffMAE (Wei et al., 2023) connects diffusion to masked autoencoders, while REPA (Yu et al., 2024) shows that aligning diffusion representations with self-supervised features dramatically speeds up training. Synthetic captioning, as demonstrated by DALL-E 3 (Betker et al., 2023), improves text-image alignment by training on detailed machine-generated captions. Large-scale systems like FLUX (Esser et al., 2024) represent the upper bound in capability but require substantial compute.

A key trade-off in current practice is data curation: high-quality subsets improve metrics but discard potentially useful signal. MIRO sidesteps this trade-off by conditioning on reward scores, allowing the model to learn from the full data spectrum while maintaining controllable quality at inference.

### A.4. Reward-Based Alignment

**Reward Models.** Multiple reward models capture complementary aspects of image quality: AestheticScore (Schuhmann et al., 2022) for visual appeal, HPSv2 (Wu et al., 2023) and PickScore (Kirstain et al., 2023) for human preferences, ImageReward (Xu et al., 2023) for instruction following, VQAScore (Lin et al., 2024) for compositional understanding, JINA CLIP (Koukounas et al., 2024) for text-image alignment, and SciScore (Li et al., 2025) for scientific accuracy. These rewards are often complementary and sometimes conflicting, motivating multi-reward approaches.

**RL-Based Fine-Tuning.** Reinforcement learning from human feedback (RLHF) (Christiano et al., 2017) has been adapted for diffusion models. DDPO (Black et al., 2024) and DPOK (Fan et al., 2023) apply policy gradients to fine-tune diffusion models on reward signals, while PRDP (Deng et al., 2025) scales this to large models. Recent work has improved RL stability: DanceGRPO (Xue et al., 2025) adapts Group Relative Policy Optimization for visual generation, achieving up to 181% improvements over baselines, and Flow-GRPO (Liu et al., 2025) extends online RL to flow matching models. However, RL-based approaches still suffer from high variance, require careful hyperparameter tuning, and risk mode collapse when optimizing single rewards (GX-Chen et al., 2025; Kwa et al., 2024).

**Gradient-Based Reward Fine-Tuning.** An alternative to RL is direct backpropagation through the sampling chain. DRaFT (Clark et al., 2024) backpropagates reward gradients through the diffusion process, using truncation and LoRA for efficiency. AlignProp (Prabhudesai et al., 2023) achieves $25\times$ faster training than DDPO through end-to-end reward backpropagation with gradient checkpointing. Reward-Instruct (Luo et al., 2026) provides yet another approach. These methods offer faster convergence than RL but still optimize for fixed reward objectives rather than enabling controllable generation.

**Preference Learning.** Direct Preference Optimization (DPO) (Rafailov et al., 2023) and its diffusion variants (Wallace et al., 2024; Li et al., 2024) learn from pairwise preferences without explicit reward modeling. While effective for single-objective alignment, these methods learn an implicit preference direction rather than providing explicit multi-dimensional control.

**Connections to Inverse and Upside-Down RL.** The relationship between reward learning and alignment has deep roots in reinforcement learning. Inverse RL (Ng & Russell, 2000; Ziebart et al., 2008) learns reward functions from expert demonstrations—analogous to how preference-based reward models are trained from human feedback. More directly relevant, *upside-down RL* (Srivastava et al., 2019) and the Decision Transformer (Chen et al., 2021) condition on desired returns during training rather than optimizing for them, treating RL as supervised learning on (state, action, return) tuples. MIRO applies this "upside-down" paradigm to T2I alignment: instead of optimizing rewards via RL, we condition on reward scores during pretraining, enabling the model to generate at any desired reward level without RL instability.

**Multi-Reward Approaches.** Handling multiple rewards simultaneously is challenging. Rewarded Soups (Rame et al., 2023) trains separate models for each reward and averages weights, but requires $N$ models for $N$ rewards and recomputing averages to change reward mixtures. Parrot (Lee et al., 2024) uses multi-objective RL with Pareto-optimal selection, but inherits RL's instability and mode collapse risks. Rewards-in-Context (RiC) (Yang et al., 2024) conditions LLM responses on reward tokens via supervised fine-tuning, enabling dynamic preference adjustment—conceptually similar to MIRO but developed for language models rather than T2I generation. Control-theoretic formulations (Uehara et al., 2024; Tang & Zhou, 2025; Domingo-Enrich et al., 2025) optimize continuous-time dynamics but are computationally expensive; lighter approaches avoid full trajectory gradients (Oertell et al., 2024; Miao et al., 2024; Jia et al., 2024).

**Test-Time Scaling.** At inference, test-time compute can boost quality via sample-and-select strategies (Ma et al., 2025; Uehara et al., 2025) or reward-guided refinement (Ben-Hamu et al., 2024; Tang et al., 2024). ReNO (Eyring et al., 2024) optimizes the initial noise using reward gradients, and Noise Hypernetworks (Eyring et al., 2025) amortize this cost. These approaches trade inference compute for alignment and are complementary to training-time methods.

**MIRO's Position.** A key distinction of MIRO is that it integrates multi-reward conditioning directly into *pretraining*, rather than requiring a separate post-hoc alignment stage. All prior reward-based alignment methods operate on pretrained models: RL-based approaches (DDPO, DPOK, DanceGRPO, Flow-GRPO, Parrot) fine-tune with policy gradients, gradient-based methods (DRaFT, AlignProp) backpropagate reward signals through a frozen or LoRA-adapted model, preference learning (DPO) requires pairwise preference data, and test-time methods (ReNO) optimize at inference. Rewarded Soups requires $N$ separate fine-tuned models. Even RiC (Yang et al., 2024), the most conceptually similar approach, applies reward conditioning via fine-tuning on LLMs.

In contrast, MIRO learns the full reward-conditioned distribution from scratch during pretraining, using supervised conditioning on reward vectors. This offers several advantages: (1) no separate alignment stage or additional training phases, (2) the model learns how all reward levels manifest visually across the full data spectrum, (3) stable training without RL instability or mode collapse, and (4) inference-time control over reward trade-offs without retraining. MIRO extends coherence-aware training (Dufour et al., 2024) from single quality scores to multiple rewards, bringing the "upside-down" alignment paradigm to T2I pretraining.

## B. Implementation Details

**Architecture Modifications.** Our flow matching network $v_\theta$ takes as input the noisy sample $x_t$, text condition $c$, and the binned reward vector $\hat{\mathbf{s}} = [\hat{s}_1, \hat{s}_2, \ldots, \hat{s}_N]$. The reward conditioning is implemented through:

- **Sinusoidal embeddings**: Each reward bin index $\hat{s}_i$ is encoded using sinusoidal position embeddings, similar to those used in transformer architectures

- **Token space mapping**: The sinusoidal reward embeddings are projected to the same dimensional space as text tokens

- **Token concatenation**: The projected reward embeddings are concatenated to the text token sequence, allowing the model to process rewards and text through the same attention mechanism

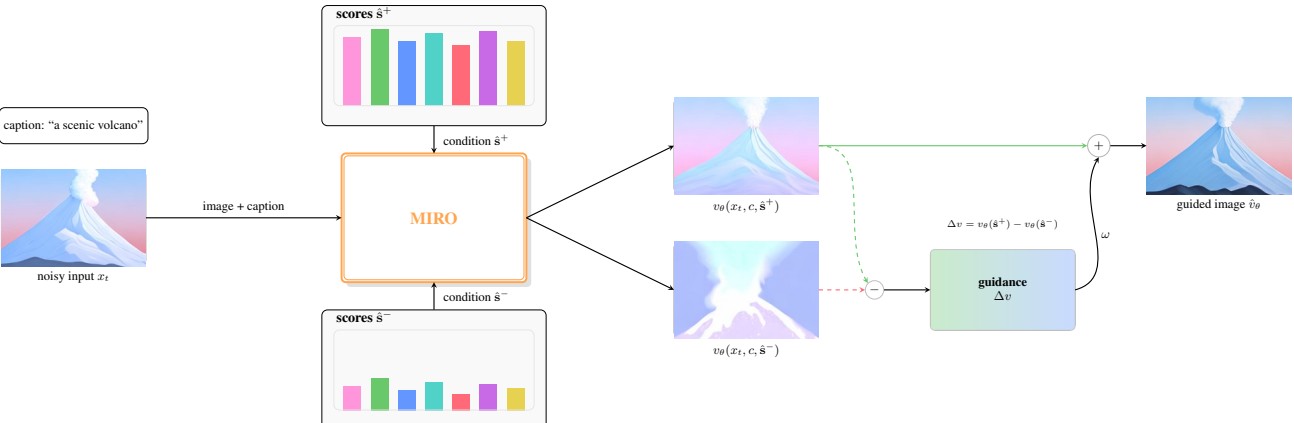

*Figure 8.* MIRO training pipeline. Top: dataset *scoring* with multiple rewards $r_1, \ldots, r_N$ produces a scores vector $\hat{\mathbf{s}}$. Bottom: during *training*, the model conditions on $\hat{\mathbf{s}}$ and a noisy input $x_t = (1-t)x + t\,\epsilon$ to learn to denoise toward high-reward regions.

*Figure 9.* MIRO inference overview (single model). The previous step $x_t$ and caption are fed to one MIRO model while conditioning on two reward histograms: $\hat{\mathbf{s}}^+$ (top) and $\hat{\mathbf{s}}^-$ (bottom), producing $v_\theta(x_t, c, \hat{\mathbf{s}}^+)$ and $v_\theta(x_t, c, \hat{\mathbf{s}}^-)$. The guidance direction $\Delta v = v_\theta(\hat{\mathbf{s}}^+) - v_\theta(\hat{\mathbf{s}}^-)$ is scaled by $\omega$ and added to the high-reward output to obtain the guided image $\hat{v}_\theta$.

**Rewards preprocessing** For each reward model $r_j$, we:

1. Compute scores on a representative subset $\mathcal{D}_{\text{cal}} \subset \mathcal{D}$ of the training data

2. Sort the scores and divide them into $B$ bins with equal population

3. Map each raw score $s_j^{(i)}$ to its corresponding bin index $\hat{s}_j^{(i)} \in \{0, 1, \ldots, B-1\}$

This binning approach provides several advantages: (1) it normalizes different reward scales into a common discrete space, (2) ensures balanced training across all quality levels, and (3) provides interpretable conditioning signals where higher bin indices correspond to better quality.

**Experimental Setup** We used the TextRIN architecture (Dufour et al., 2024) with several modifications: FFN layers replaced with SwiGLU (Shazeer, 2020), LayerNorm replaced with RMSNorm (Zhang & Sennrich, 2019), and QK-Norm (Henry et al., 2020) in attention mechanisms. We employed flow matching instead of diffusion for generation. Models were trained for 500k steps with batch size 1,024 and learning rate 1e-3. We train our model in 256px resolution. We combined CC12M (Changpinyo et al., 2021) and LAION Aesthetics 6+ (Schuhmann et al., 2022) for 16M total image-text pairs,

following (Dufour et al., 2024). We used seven reward models for MIRO: **Aesthetic Score** (Schuhmann et al., 2022) for visual appeal, **HPSv2** (Wu et al., 2023) for human preference alignment, **ImageReward** (Xu et al., 2023) for text-image correspondence and user preference, **PickScore** (Kirstain et al., 2023) for user preference, **VQAScore** (Lin et al., 2024) for visual comprehension, **JINA CLIP Score** (Koukounas et al., 2024) for long captions CLIP score, and **SciScore** (Li et al., 2025) for scientific accuracy.

**SOTA Baselines** We compare MIRO against the following baselines:

- **SD v1.5**: (Rombach et al., 2022)

- **SD v2.1**: (Rombach et al., 2022)

- **PixArt-**$\alpha$: (Chen et al., 2024b)

- **PixArt-**$\Sigma$: (Chen et al., 2024a)

- **CAD**: (Dufour et al., 2024)

- **Sana-0.6B**: (Xie et al., 2024)

- **Sana-1.6B**: (Xie et al., 2024)

- **SDXL**: (Podell et al., 2024)

- **FLUX-dev**: (Labs, 2024)

- **SD3-Medium**: (Esser et al., 2024)

## C. Captioning Pipeline

We caption images using the model `google/gemma-3-12b-it` available on HuggingFace. To generate long captions, we use the following prompt:

```
"Analyze the following image in detail. Identify all prominent objects, their attributes (
    color, material, shape, size, texture), their spatial relationships, the overall scene
     and setting, the lighting conditions, and any relevant style or composition details."
```

```
"Based on your analysis, generate a caption of the image. It should be descriptive enough
    to allow a diffusion model to accurately reconstruct the image. Include specific
    details rather than general descriptions. For example, instead of 'a blue car,'
    describe it as 'a shiny, dark blue vintage sedan with chrome bumpers parked on a
    cobblestone street.'
```

```
"Please ensure the caption is enclosed within <CAPTION> and </CAPTION> tags. "
```

```
"Example of lengths for the caption:"
```

```
"<CAPTION> A plump gray domestic shorthair cat with symmetrical white paws sleeps curled
    into a tight circle on a sunlit oak windowsill, its body occupying about two-thirds of
     the surface. The cat's head rests on its hind legs, with its tail wrapped neatly
    around its body. The windowsill shows distinct wood grain patterns and a sun-bleached
    patch where sunlight consistently hits. To the left, semi-sheer white lace curtains
    with a small floral pattern hang from a wooden rod, partially billowing inward from a
    30-centimeter-wide open window that reveals an out-of-focus garden with green foliage.
     On a round wooden side table to the right, a transparent glass vase holds five pink
    peonies and three white snapdragons in water, with visible pollen grains floating on
    the surface. The table's surface shows faint circular water stains and a light dusting
     of pollen. Behind the table, an armchair with beige linen upholstery features a
    folded gray knit blanket draped over its back. A vintage wall clock with Roman
    numerals and brass hands is mounted above the windowsill. Sunlight streams through the
     window. </CAPTION> "
```

```
"<CAPTION> A Space Gray iPad Pro displays a vibrant beach sunset, positioned on a rustic
    walnut table. The attached Magic Keyboard is folded back, and a Apple Pencil rests
    diagonally across an open leather folio case, revealing its suede-lined interior. To
    the left, a double-walled glass mug of black coffee sits on a cork coaster with a thin
     ring of condensation and a light sprinkle of cinnamon on the foam. A small ceramic
    pot contains a jade pothos plant with six visible leaves, two of which trail over the
    table's edge. The table's surface shows natural wood grain variations, including a
    dark, heart-shaped knot near the center. In the background, a mid-century modern sofa
    in teal velvet has two throw pillows with geometric patterns. A bookshelf against the
    far wall holds a mix of books, a brass desk lamp, and a stacked stone decoration.
    Natural light filters through a casement window with slightly wavy glass panes,
    creating visible light refractions on the table. A ceiling fan casts moving shadows,
    and a seashell wind chime hangs outside the window, occasionally tinkling in the
    breeze.</CAPTION>"
```

```
"<CAPTION> A rectangular farmhouse table is covered with a pressed linen tablecloth (ivory
     with subtle gray stripes) and meticulously set for eight guests. Each place setting
    includes a plate with Wild Strawberry pattern, a five-piece sterling silver flatware
    set, an water goblet, and a wine glass, all arranged with precise alignment. A cloth
    napkin is folded into a rectangle and tied with a burgundy silk ribbon. The
    centerpiece is a floral arrangement in a mercury glass compote, featuring six red
    roses, four white peonies, eight pine sprigs, and three cinnamon sticks. Eight tapered
     candles in brass holders are placed among the flowers. Wooden dining chairs with navy
     velvet upholstery have wool throws draped over their backs. A wrought iron chandelier
     with six Edison bulbs hangs above the table, casting warm light that reflects off the
     crystal glassware. The walls are adorned with cedar garlands embedded with 50 white
    fairy lights, and three framed botanical prints hang in a horizontal row. In the
    background, a fireplace with a visible flame and a stack of birch logs adds warmth to
    the scene. The air smells faintly of pine, cinnamon, and beeswax polish.</CAPTION> "
```

```
"An alt-text corresponding to the image is: <ALT-TEXT>  </ALT-TEXT>"
```

To compute reward scores, we generate short captions of the images. We use the following prompts :

```
"Generate a short caption of the image. Please ensure the caption is enclosed within <
    CAPTION> and </CAPTION> tags. "

"Example of lengths for the caption:"
"<CAPTION>A cat sleeping on a windowsill.</CAPTION> "
"<CAPTION>A beautiful sunset over the mountains with a clear sky.</CAPTION> "
"<CAPTION>A group of people enjoying a picnic in the park on a sunny day.</CAPTION> "
"<CAPTION>A boy playing with a ball in the backyard.</CAPTION> "
```

Table 2 shows examples of long and short captions.

| Images | Captions |
|---|---|
| 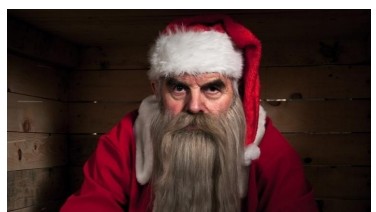 | **Short:** A modern living room features a gray sofa, a vibrant watercolor garden print, and soft throws under a minimalist floor lamp. 

 **Long:** A bright, contemporary living room showcases a spacious gray three-seater sofa with a subtly textured, linen-blend upholstery, positioned against a pristine white wall. The wall is adorned with a large, rectangular canvas print displaying a vibrant, watercolor-inspired illustration of a whimsical garden scene, featuring a diverse array of colorful birds (including blue, orange, yellow, and green-bodied birds) amidst pink, yellow, and white flowers and lush greenery. A tall, minimalist black floor lamp with a geometric, faceted metal shade casts a soft light onto the light gray hardwood floor to the right of the sofa. A dark blue knitted throw is casually draped over the sofa's left armrest, and a lighter blue/gray throw is folded on the right. The overall aesthetic is clean, modern, and inviting, highlighting the artwork as a striking focal point in a minimalist setting. |
| 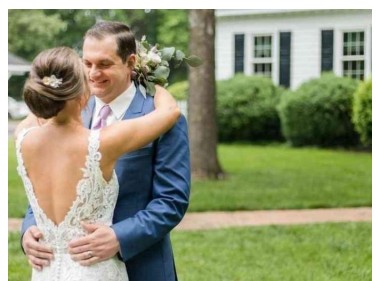 | **Short:** A scowling Santa in a velvet suit glares intensely from inside a wooden hut. 

 **Long:** A strikingly serious portrait of a man portraying Santa Claus, captured in a tight close-up from within a rustic wooden sauna. He is attired in a bright red Santa suit made of a textured velvet-like fabric, complete with white fur trim around the collar and cuffs, and a traditional conical hat featuring a large, plush white pom-pom. His long, thick, and unkempt white beard covers a significant portion of his face. His dark, bushy eyebrows are heavily furrowed, conveying a palpable sense of discontent or annoyance, and his dark eyes gaze directly at the viewer with alarming intensity. The sauna is constructed from light-colored pine planks, exhibiting a natural wood grain and a slightly rough texture, creating a warm but somewhat enclosed feeling. Dramatic directional lighting from the left aggressively illuminates his face, casting heavy shadows to the right, accentuating the wrinkles and emphasizing the seriousness of his expression. The overall effect is a jarring juxtaposition of the familiar Christmas icon with an unsettling and unexpected mood, suggesting a Santa Claus far removed from the joyful spirit typically associated with the holiday. |
|  | **Short:** Newlyweds share a tender embrace on a lush green lawn, she in lace and flowers, he in navy and pink. 

 **Long:** A heartwarming candid moment featuring a bride and groom embracing on a vibrant green lawn, set before a stately two-story white house constructed in a classic colonial architectural style with dark blue, evenly spaced shutters. The bride has light brown hair elegantly styled in an updo accented with a small white floral detail. She wears a flowing white wedding dress with a delicate lace overlay and a low, open back, revealing a glimpse of her skin. Her arms are wrapped tightly around her groom, who is dressed in a navy blue suit, a crisp white dress shirt, and a light pink tie. A shiny silver wedding band adorns his left ring finger. The bride holds a bouquet consisting of a mix of white and pale pink roses interspersed with lush greenery. The background features meticulously trimmed hedges and a mature tree with a thick, textured gray trunk. Soft, diffused natural light bathes the scene, creating gentle shadows across the lawn. The overall impression is one of joy, love, and timeless elegance, characteristic of a wedding day celebration. |

*Table 2.* Example of captions used in the training set.

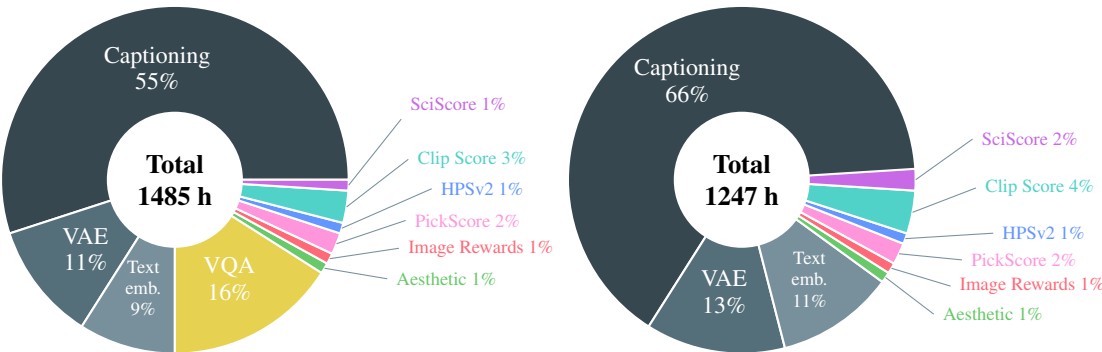

*Figure 10.* **The charts illustrate the time breakdown for preprocessing 16M images** (measured in H100 GPU hours). **Left**: In the full pipeline, precomputing all rewards consumes 25% of the total budget. **Right**: Removing VQA reduces the reward computation overhead to 11% (from 369 to 131 hours). In both configurations, captioning remains the primary bottleneck.

## D. Preprocessing Computational Costs

Figure 10 presents a computational breakdown of the MIRO preprocessing pipeline. Calculating rewards for the 16M image dataset constitutes 25% of the total computational budget. Captioning accounts for over 55% of the runtime. It remains the primary bottleneck. Given that reward computation requires significantly less compute than captioning, MIRO offers a efficient alternative to captioning-based approaches for low-budget training.

## E. Additional Experimental Results

### E.1. CFG Rate Analysis

We analyze how different single-reward models and MIRO respond to varying classifier-free guidance (CFG) rates. Figure 11 presents reward scores across six metrics as a function of CFG rate.

**Analysis:** Several key observations emerge from these plots. First, all models seem to achieve optimal performance around CFG 5–7. At very low CFG rates, conditioning is too weak to guide generation effectively, while excessively high CFG values lead to oversaturation and reduced sample quality. Second, single-reward models demonstrate expected specialization: the **Aesthetic**-only model achieves the highest Aesthetic Score (6.65 at CFG=7) but significantly underperforms on CLIP-based metrics, reflecting a trade-off between visual appeal and text-image alignment. Conversely, **CLIP**-conditioned models excel at their target metric but show lower aesthetic scores. Third, **MIRO** consistently achieves competitive performance across all metrics without sacrificing any single dimension, demonstrating its ability to balance multiple objectives simultaneously. Notably, MIRO achieves the highest Image Reward scores (1.05 at CFG=5), outperforming even the ImageReward-specialized model (1.04), suggesting that multi-reward conditioning can discover complementary signal combinations that exceed single-objective optimization.

### E.2. Single-Reward Comparison on GenEval

We compare MIRO against each single-reward model on the GenEval benchmark, which evaluates compositional generation capabilities across six categories: Single Object, Two Objects, Position, Counting, Colors, and Color Attribution.

**Analysis:** The radar plots reveal that MIRO achieves a well-balanced profile across all GenEval dimensions, whereas single-reward models show pronounced weaknesses. The **Aesthetic**-only model performs poorly across nearly all compositional metrics, particularly on Position (6.0 vs. MIRO's 19.0) and Counting (24.1 vs. MIRO's 55.3), indicating that optimizing purely for visual appeal can severely impair semantic understanding. **VQA** and **SciScore** models achieve the best Single Object scores (97.2 and 94.4), but MIRO remains competitive (92.2) while excelling on multi-object scenarios. Interestingly, **CLIP** and **SciScore** models show the strongest Position performance (24.3), suggesting that text-alignment objectives may encode spatial reasoning capabilities. MIRO's strength lies in its consistent performance across all categories, achieving the highest Two Objects score (67.7) and strong Color Attribution (37.5), demonstrating that multi-reward conditioning produces more semantically faithful generations than any single reward alone.

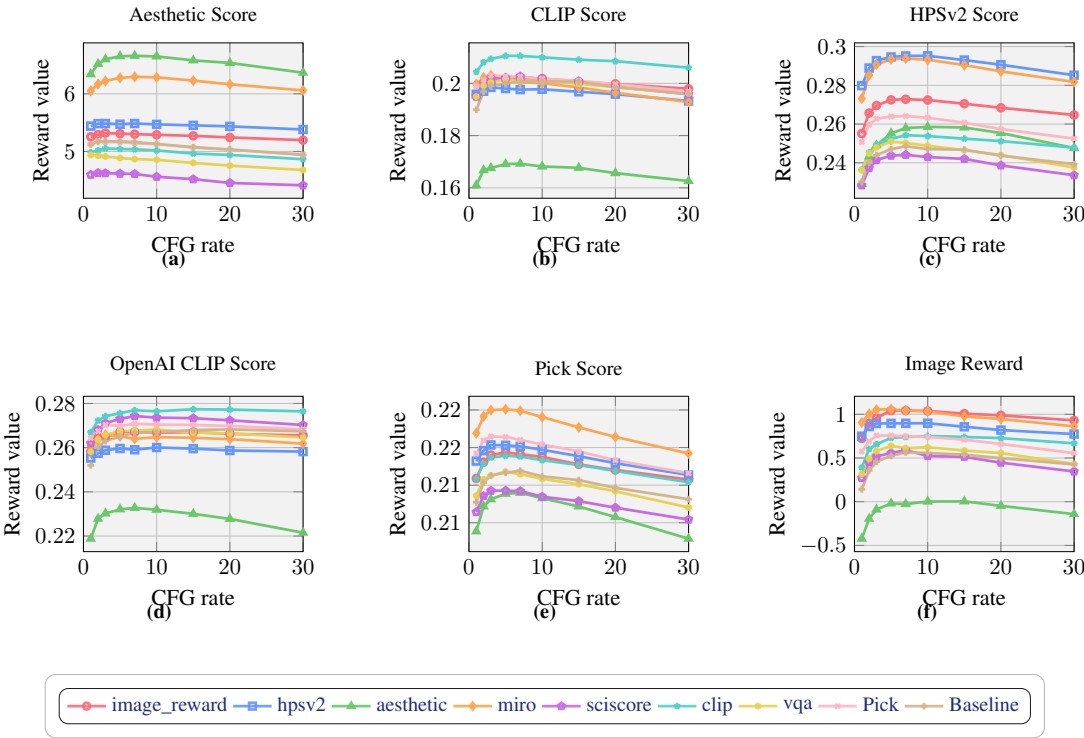

*Figure 11.* Score plots for different reward functions: (a) Aesthetic Score, (b) CLIP Score, (c) HPSv2 Score, (d) OpenAI CLIP Score, (e) Pick Score, and (f) Image Reward. Each plot shows all models color-coded according to the legend.

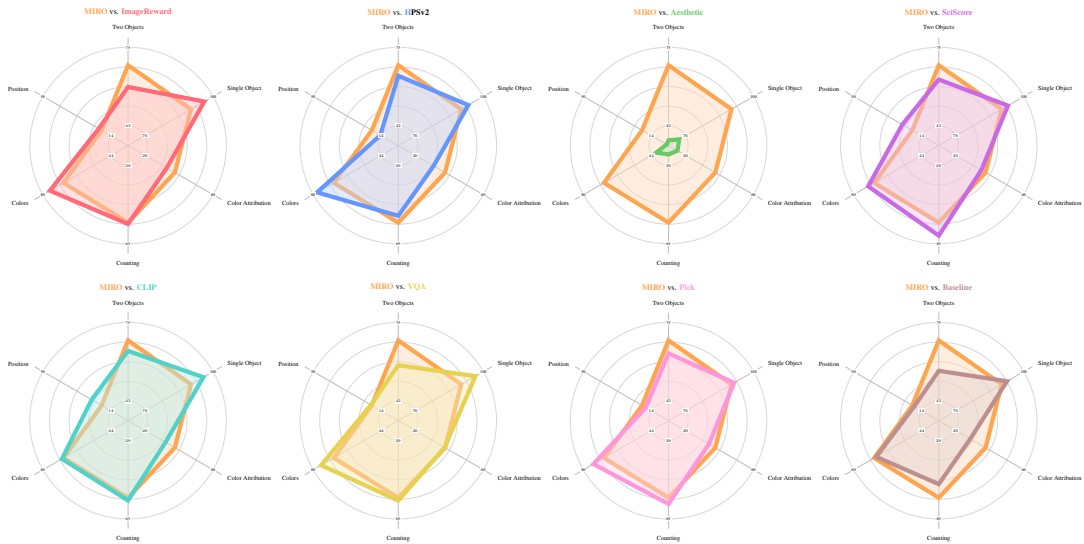

*Figure 12.* Comparison of the MIRO model against eight other specialist/baseline models on GenEval metrics. Each radar plot shows the MIRO model (orange) versus a comparison model across six GenEval categories: Single Object, Two Objects, Position, Counting, Colors, and Color Attribution. Scores range from 0 to 100 for all categories. Min and max values on each axis show the range of actual metric scores and are consistent across all plots.

### E.3. Aesthetic Weight Sensitivity

We investigate how varying the aesthetic reward weight during inference affects both semantic and perceptual quality. This experiment uses the Synth MIRO model and varies only the aesthetic component $\hat{s}^+_{\text{aesthetic}}$ while keeping other rewards fixed.

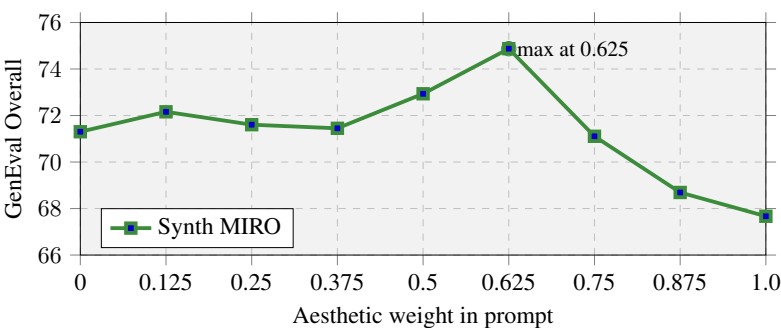

*Figure 13.* GenEval Overall vs aesthetic prompt weight for 'Synth MIRO'. We vary the positive target $\hat{s}^+_{\text{aesthetic}}$ while keeping the other components of $\hat{s}^+$ equal to 1 and $\hat{s}^-$ fixed. Higher is better.

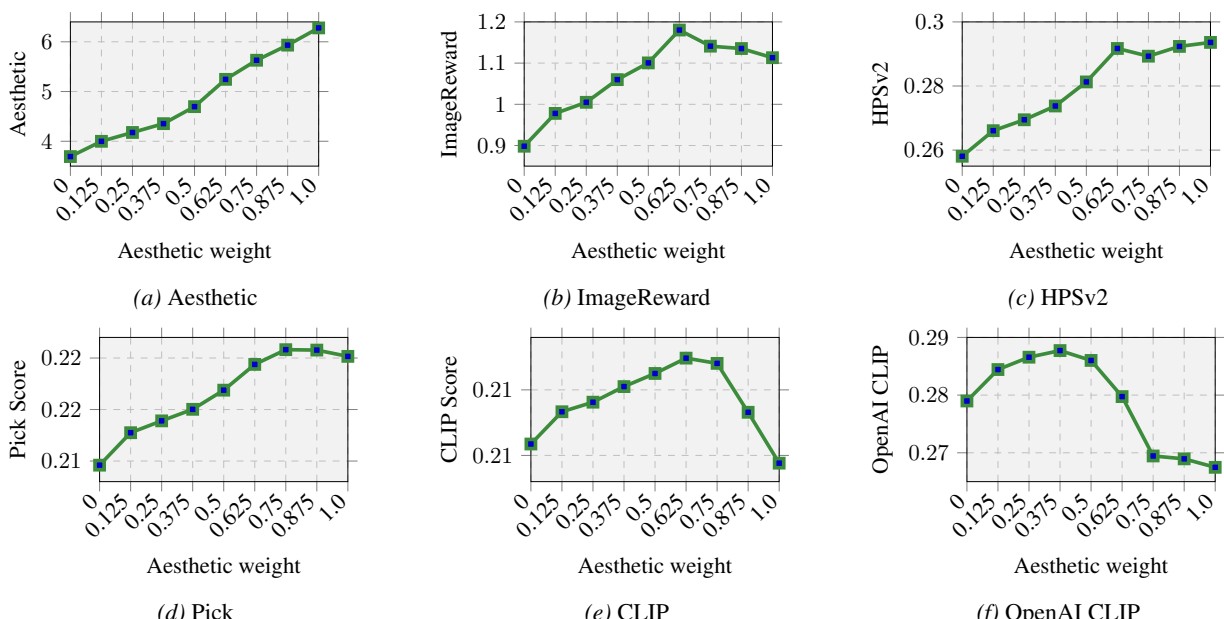

*Figure 14.* Six metrics vs aesthetic prompt weight for 'Synth MIRO'. Each subplot shows the metric value over aesthetic weight in the prompt.

**Analysis:** Figure 13 reveals a non-monotonic relationship between aesthetic weight and GenEval performance. The optimal GenEval score (74.9) is achieved at an intermediate aesthetic weight of 0.625, rather than at the extremes. This suggests that moderate aesthetic guidance can improve compositional coherence—possibly by encouraging more visually structured and less noisy outputs—while excessive aesthetic emphasis (weight 1.0) leads to GenEval degradation (67.7). Figure 14 further illustrates the multi-objective trade-offs: Aesthetic Score monotonically increases with weight (from 3.7 to 6.3), as expected. HPSv2 and Pick Score also increase, reflecting their correlation with visual quality. However, text-alignment metrics (CLIP and OpenAI CLIP) reach a maximum at intermediate weights, with OpenAI CLIP peaking at weight 0.375 before declining. This demonstrates that MIRO's multi-reward formulation enables users to navigate complex Pareto frontiers at inference time, selecting operating points that balance competing objectives according to their specific requirements.

### E.4. Training Dynamics

We examine how reward metrics evolve during training for different conditioning strategies.

**Analysis:** The training curves in Figure 15 reveal several important dynamics. First, **MIRO** (orange) demonstrates remarkably fast convergence across all metrics, reaching competitive performance within the first 100k steps (approximately 20% of total training). This rapid improvement suggests that multi-reward conditioning provides a rich learning signal

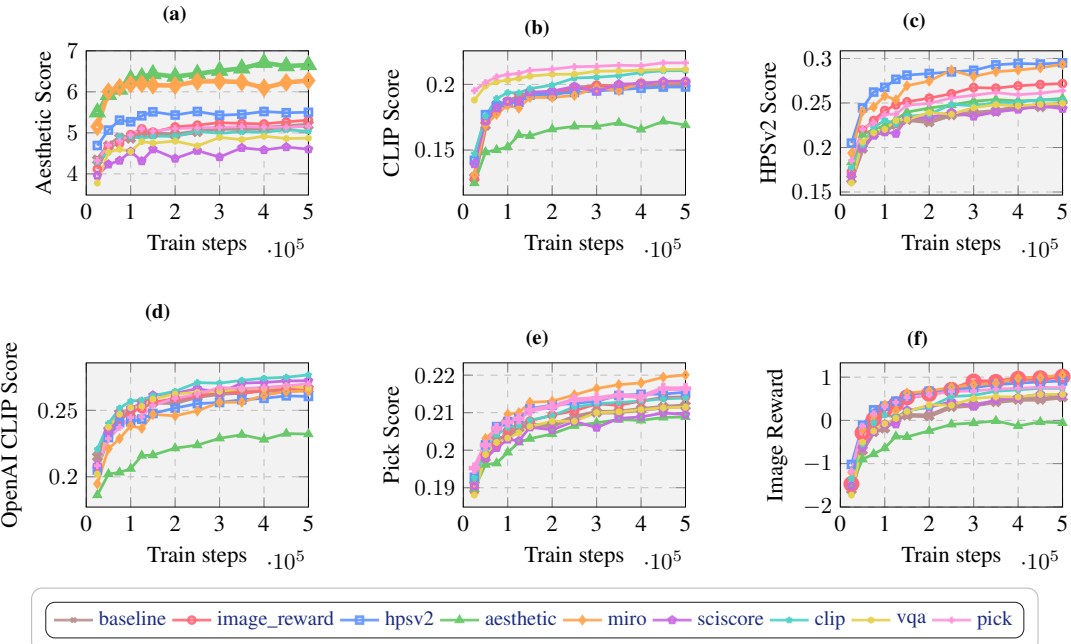

*Figure 15.* Training curves for different reward functions: (a) Aesthetic Score, (b) CLIP Score, (c) HPSv2 Score, (d) OpenAI CLIP Score, (e) Pick Score, and (f) Image Reward. Each plot shows the reward value progression across Train steps for different models including image_reward, hpsv2, aesthetic, miro, sciscore, clip, vqa, and pick.

that accelerates quality gains early in training. Second, single-reward models show strong specialization on their target metrics: the Aesthetic model (green) achieves the highest final Aesthetic Score (6.65), and HPSv2 (blue) leads on HPSv2 Score (0.295). However, these specialists underperform on non-target metrics—the Aesthetic model achieves only 0.17 CLIP Score compared to MIRO's 0.20 and CLIP-conditioned model's 0.21. Third, MIRO achieves Pareto-optimal or near-optimal performance across all metrics simultaneously, matching or exceeding the baseline on every reward while approaching specialist-level performance on several (e.g., Image Reward: MIRO 1.05 vs. ImageReward-specialist 1.01). Fourth, the baseline model (brown) shows the slowest improvement trajectory, particularly on perceptual metrics like Image Reward, where it reaches only 0.54 compared to MIRO's 1.05—a 2× improvement. This validates that MIRO's multi-reward conditioning provides complementary supervision that accelerates learning beyond standard training objectives.

### E.5. Leave-one-out Experiments

We analyze the contribution of each reward model by training MIRO with one reward removed at a time. Figure 16 shows the impact on reward metrics, while Figure 17 shows the impact on Geneval metrics.

**Analysis:** As expected, removing a specific reward generally leads to a decrease in that specific metric. For instance, removing the Aesthetic Score leads to a significant drop in the aesthetic metric (from 6.23 to 5.05). However, interestingly, this removal leads to substantial improvements in Geneval metrics, particularly for **Position** (+11.8 points) and **Two Objects** (+11.4 points). This suggests a potential trade-off between optimizing for pure visual aesthetics and maintaining strict semantic fidelity or spatial composition. Conversely, removing **ImageReward** or **CLIP** tends to hurt semantic metrics like Two Object detection and Color accuracy, highlighting their importance for text-image alignment.

### E.6. Binning Ablation

We compare three binning strategies for the reward conditioning: **Quantile Bins** (our default MIRO), **Refined Quantile Bins** (64 quantiles with the last 8 re-binned), and **Uniform Bins**.

**Analysis:** We chose **Quantile Bins** as our default strategy for several reasons. First, it provides the most balanced performance across the full range of metrics: while **Refined Quantile** achieves slightly higher reward scores (e.g., Aesthetic 6.40 vs 6.23), it underperforms on overall GenEval (54 vs 57 for Quantile). Similarly, **Uniform Bins** show strong

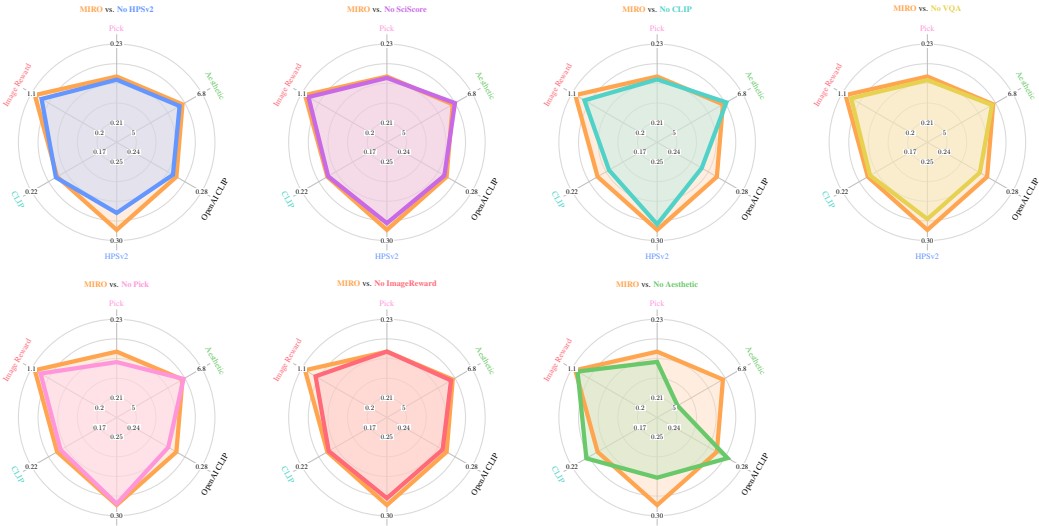

*Figure 16.* Ablation study: MIRO vs MIRO leaving one reward out.

performance on specific metrics like **Position** (25.8 vs 12.2), but at the cost of lower reward scores overall. Second, Quantile Bins is simpler and more robust: it requires no additional hyperparameter tuning (unlike Refined Quantile which requires choosing refinement thresholds) and naturally adapts to the data distribution of each reward model. Third, Quantile Bins ensures balanced training across all quality levels by design, preventing the model from being dominated by the most frequent score ranges. The marginal gains from alternative strategies on specific metrics do not justify the added complexity or the trade-offs on other metrics.

### E.7. Post-training Ablation

We investigate the efficiency of MIRO by comparing a model trained from scratch with MIRO (500k steps) against a **Post-training** approach, where a baseline model trained for 450k steps is fine-tuned with MIRO for only 50k steps.

**Analysis:** The Post-training approach demonstrates that **MIRO** can be applied as a fine-tuning strategy, though with some trade-offs. Despite being trained with **MIRO** for only 10% of the total steps, Post-training approaches the full **MIRO** model on key reward metrics like **Aesthetic Score** (6.22 vs 6.23) and **HPSv2**, but slightly underperforms on the overall GenEval score (56 vs 57). Interestingly, the Post-training model shows improved performance on specific GenEval metrics like **Position** (23.2 vs 12.2), suggesting that initializing with a strong baseline may preserve some semantic knowledge that can be destabilized during long multi-objective training from scratch. However, full training with **MIRO** from scratch remains the recommended approach for optimal overall performance.

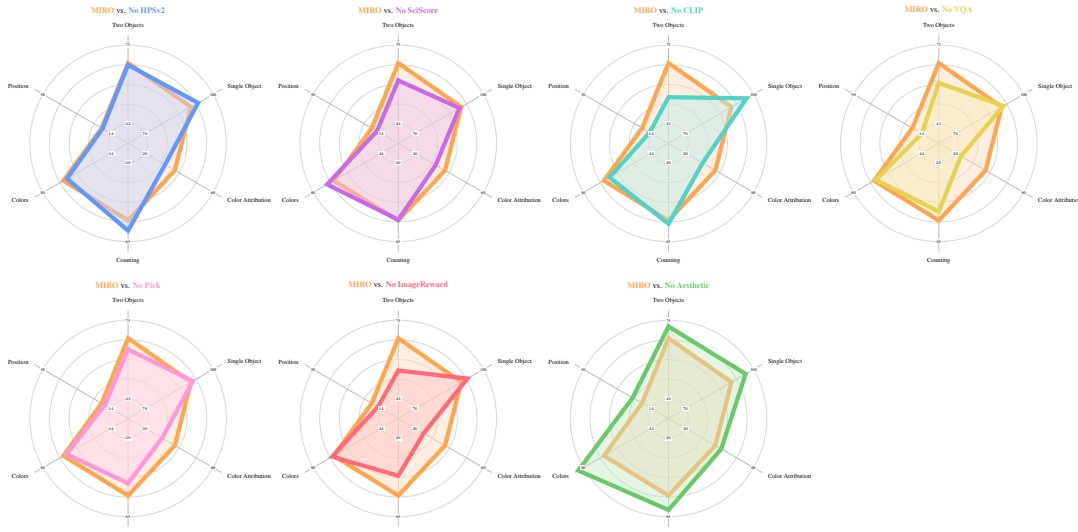

*Figure 17.* Ablation study on GenEval: MIRO vs MIRO leaving one reward out.

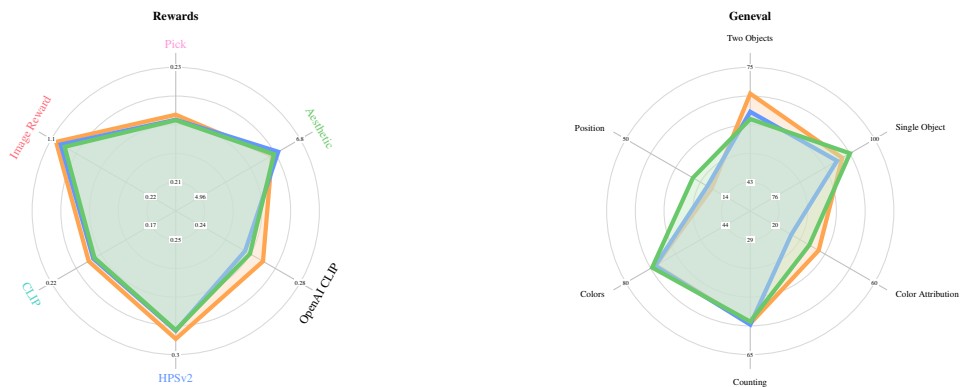

*Figure 18.* Comparison of **MIRO**, **Refined**, and **Uniform** bins on Rewards (Left) and Geneval (Right).

## F. Additional Training Progression Examples

We present additional qualitative examples comparing the visual progression of MIRO and baseline models throughout training. Figures 20 and 21 show side-by-side comparisons for various prompts ranging from simple objects to complex compositional descriptions.

**Analysis of Figure 20:** The progression on simple prompts ("a panda", "a city", "a clock tower", "a ladder") illustrates MIRO's accelerated learning dynamics. At early training stages (25k–50k steps), both models produce noisy, incoherent outputs. However, by 75k steps, MIRO already generates recognizable objects with improved structural coherence, while the baseline remains blurry and ill-defined. The gap widens at later stages: at 200k steps, MIRO produces images with clear details, proper lighting, and coherent composition, whereas the baseline still exhibits artifacts and lacks visual refinement. By 400k steps, MIRO outputs approach photorealistic quality with rich textures and accurate object representation, demonstrating that multi-reward conditioning not only improves final quality but fundamentally accelerates the rate at which models acquire visual competence.

**Analysis of Figure 21:** The more complex prompts ("a taxi", "an elephant", and the detailed compositional prompts involving a beaver in a library and a raccoon in formal attire) further highlight MIRO's advantages. For the compositional prompts, which require generating multiple attributes and objects in specific spatial configurations, MIRO demonstrates superior semantic understanding throughout training. The beaver prompt requires generating glasses, a vest, a tie, books, and a library setting—MIRO progressively assembles these elements coherently, while the baseline struggles to bind attributes to the correct objects. Similarly, for the raccoon prompt combining formal attire, a tophat, a cane, a garbage bag, and

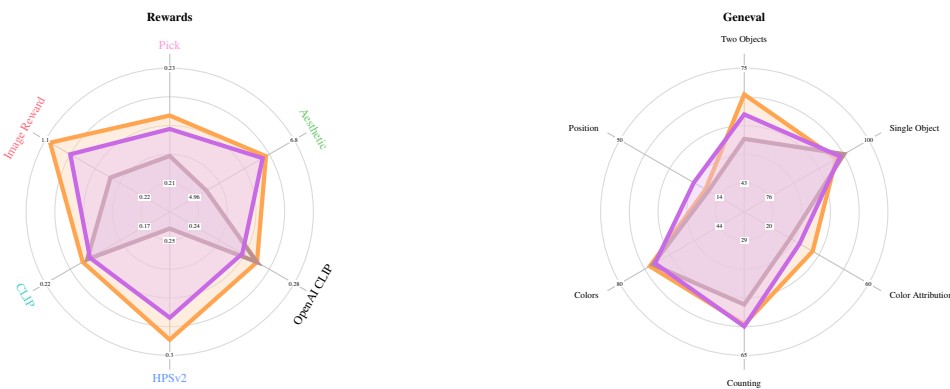

*Figure 19.* Comparison of **MIRO**, **Baseline**, and **Post-Train** on Rewards (Left) and Geneval (Right).

| | GenEval | | | | | | | PartiPrompts | | | |
|---|---|---|---|---|---|---|---|---|---|---|---|
| Model | Overall | Single Obj. | Two Obj. | Position | Counting | Colors | Color Attr. | Aesthetic | Image | HPSv2 | PickAScore |
| *Reference* | | | | | | | | | | | |
| Baseline | 52 | 94 | 55 | 18 | 49 | 68 | 29 | 5.18 | 0.52 | 0.25 | 0.212 |
| MIRO | 57 | 92 | 68 | 19 | 55 | 69 | 38 | 6.28 | **1.06** | **0.29** | **0.220** |
| *Leave-one-out Ablations* | | | | | | | | | | | |
| No Aesthetic | **63** | 97 | **72** | 24 | **62** | **83** | **41** | 5.05 | 1.03 | 0.28 | 0.217 |
| No Pick | 53 | 93 | 63 | 17 | 50 | 68 | 30 | 6.31 | 0.98 | **0.29** | 0.217 |
| No ImageReward | 51 | 94 | 55 | 16 | 46 | 70 | 25 | 6.23 | 0.92 | **0.29** | **0.220** |
| No HPSv2 | 57 | 95 | 67 | 18 | 60 | 67 | 32 | 6.20 | 0.97 | 0.28 | 0.219 |
| No CLIP | 53 | **98** | 54 | 15 | 57 | 66 | 30 | 6.37 | 0.94 | **0.29** | 0.219 |
| No SciScore | 55 | 92 | 61 | 16 | 55 | 73 | 32 | 6.34 | 1.01 | **0.29** | **0.220** |
| No VQA | 52 | 93 | 60 | 14 | 51 | 68 | 23 | 6.25 | 0.99 | **0.29** | 0.219 |
| *Binning Strategies* | | | | | | | | | | | |
| Refined Quantile Bins | 54 | 91 | 63 | 20 | 56 | 69 | 26 | **6.40** | 1.02 | **0.29** | 0.219 |
| Uniform Bins | 57 | 94 | 61 | **26** | 55 | 70 | 34 | 6.32 | 0.98 | **0.29** | 0.219 |
| *Post-Training* | | | | | | | | | | | |
| Post-Training (50k) | 56 | 93 | 62 | 23 | 56 | 67 | 32 | 6.22 | 0.88 | 0.28 | 0.217 |

*Table 3.* Detailed numerical results for all ablation studies on GenEval and PartiPrompts benchmarks. **Bold** indicates best result, underline indicates second best.

an abstract cubism style, MIRO shows emergent compositional ability by 100k steps, successfully combining semantic content with artistic style. These examples demonstrate that MIRO's multi-reward training signal provides not just aesthetic improvements but fundamentally better semantic grounding, enabling reliable generation of complex, multi-attribute scenes that challenge single-objective training approaches.

# G. Log-Odds Maximization Theory

In this section, we provide a formal theoretical justification for MIRO's multi-reward guidance mechanism. We show that our guidance formulation can be derived as sampling from a reward-tilted distribution, providing a principled foundation for steering generation toward high-reward regions.

**Roadmap.** Our derivation proceeds in five logical steps:

1. **Preliminaries** (§G.2): We establish the connection between velocity fields and score functions, and recall how classifier guidance modifies the sampling distribution.

2. **Implicit Classification** (§G.3): We prove that MIRO's reward-conditioned velocity field implicitly learns a classifier over reward levels, eliminating the need for external classifier networks.

3. **Distributional View** (§G.4): We derive the central theorem showing that MIRO's guidance formula corresponds to sampling from a precisely characterized reward-tilted distribution.

4. **Optimization View** (§G.5): We provide a complementary interpretation showing that each denoising step performs implicit gradient ascent on the log-odds ratio.

5. **Multi-Objective Analysis** (§G.6): We analyze how this framework extends to multiple rewards, examining both the general correlated case and the special case of conditional independence.

## G.1. Notation

We first establish the notation used throughout this section:

- $x \in \mathbb{R}^d$ denotes a clean image sample

- $c \in \mathcal{T}$ denotes the text conditioning (e.g., a caption or prompt)

- $t \in [0, 1]$ denotes the flow matching time, where $t = 0$ corresponds to clean data and $t = 1$ to pure noise

- $x_t = (1 - t)x + t\epsilon$ denotes the noisy sample at time $t$, where $\epsilon \sim \mathcal{N}(0, I)$

- $\mathbf{s} = [s_1, \ldots, s_N] \in \{0, \ldots, B - 1\}^N$ denotes the discretized reward vector across $N$ reward models with $B$ bins each

- $\mathbf{s}^+$ and $\mathbf{s}^-$ denote the positive (high-reward) and negative (low-reward) conditioning targets, typically $\mathbf{s}^+ = [B - 1, \ldots, B - 1]$ and $\mathbf{s}^- = [0, \ldots, 0]$

- $v_\theta(x_t, c, \mathbf{s}) : \mathbb{R}^d \times \mathcal{T} \times \{0, \ldots, B - 1\}^N \to \mathbb{R}^d$ denotes the learned velocity field parameterized by $\theta$

- $\omega \geq 0$ denotes the guidance scale controlling the strength of reward guidance

- $p_t(x_t|c, \mathbf{s})$ denotes the distribution of noisy samples at time $t$ conditioned on text $c$ and reward level $\mathbf{s}$

## G.2. Preliminaries: Score Functions and Classifier Guidance

Before deriving MIRO's guidance mechanism, we establish the fundamental connection between velocity fields and probability distributions. This connection is crucial because it allows us to understand how modifying the velocity field changes the distribution we sample from.

In flow matching (Lipman et al., 2023), the velocity field $v_\theta$ is trained to predict the direction from $x_t$ toward the clean sample $x$. The relationship between the velocity field and the score function (gradient of the log-density) is given by:

$$\nabla_{x_t} \log p_t(x_t|c) = -\frac{v_\theta(x_t, c)}{1 - t} \tag{5}$$

where $p_t(x_t|c)$ denotes the marginal distribution of noisy samples at time $t$, conditioned on text $c$.

In classifier-guided diffusion (Dhariwal & Nichol, 2021), one modifies the score function to incorporate an external classifier $p(\mathbf{s}|x_t, c)$:

$$\nabla_{x_t} \log p_t(x_t|c, \mathbf{s}) = \nabla_{x_t} \log p_t(x_t|c) + \nabla_{x_t} \log p(\mathbf{s}|x_t, c) \tag{6}$$

The key insight of MIRO is that by conditioning on reward scores during training, we implicitly learn this classifier within the generative model itself. The next section makes this precise.

## G.3. Multi-Reward Conditioning as Implicit Classification

The classical approach to classifier guidance (Eq. 6) requires training a separate classifier network $p(\mathbf{s}|x_t, c)$ on noisy images. MIRO sidesteps this requirement entirely: by training a single velocity field $v_\theta(x_t, c, \mathbf{s})$ conditioned on reward levels, we implicitly learn the classifier as a byproduct. The following proposition formalizes this insight.

**Proposition G.1** (Implicit Classifier). *Let $v_\theta(x_t, c, \mathbf{s})$ be a velocity field trained with multi-reward conditioning. Then the difference between velocity predictions at different reward levels approximates the gradient of an implicit log-probability ratio:*

$$v_\theta(x_t, c, \mathbf{s}^+) - v_\theta(x_t, c, \mathbf{s}^-) \approx -(1-t)\nabla_{x_t} \log \frac{p(\mathbf{s}^+|x_t, c)}{p(\mathbf{s}^-|x_t, c)} \tag{7}$$

*Proof.* Consider the score functions for the reward-conditioned distributions:

$$\nabla_{x_t} \log p_t(x_t|c, \mathbf{s}^+) = -\frac{v_\theta(x_t, c, \mathbf{s}^+)}{1-t} \tag{8}$$

$$\nabla_{x_t} \log p_t(x_t|c, \mathbf{s}^-) = -\frac{v_\theta(x_t, c, \mathbf{s}^-)}{1-t} \tag{9}$$

By Bayes' rule, we can decompose each conditional score:

$$\nabla_{x_t} \log p_t(x_t|c, \mathbf{s}) = \nabla_{x_t} \log p_t(x_t|c) + \nabla_{x_t} \log p(\mathbf{s}|x_t, c) - \nabla_{x_t} \log p(\mathbf{s}|c) \tag{10}$$

Since $p(\mathbf{s}|c)$ does not depend on $x_t$, its gradient vanishes. Taking the difference:

$$\nabla_{x_t} \log p_t(x_t|c, \mathbf{s}^+) - \nabla_{x_t} \log p_t(x_t|c, \mathbf{s}^-) \tag{11}$$

$$= \nabla_{x_t} \log p(\mathbf{s}^+|x_t, c) - \nabla_{x_t} \log p(\mathbf{s}^-|x_t, c) \tag{12}$$

$$= \nabla_{x_t} \log \frac{p(\mathbf{s}^+|x_t, c)}{p(\mathbf{s}^-|x_t, c)} \tag{13}$$

Substituting the velocity field expressions:

$$-\frac{v_\theta(x_t, c, \mathbf{s}^+) - v_\theta(x_t, c, \mathbf{s}^-)}{1-t} = \nabla_{x_t} \log \frac{p(\mathbf{s}^+|x_t, c)}{p(\mathbf{s}^-|x_t, c)} \tag{14}$$

Rearranging yields the result. $\square$

Proposition 1 establishes that the velocity difference $v_\theta(x_t, c, \mathbf{s}^+) - v_\theta(x_t, c, \mathbf{s}^-)$ encodes the gradient of the log-odds ratio. This is the key building block for our main result: we now show how MIRO's guidance formula leverages this implicit classifier to sample from a reward-tilted distribution.

### G.4. Main Result: Guidance as Sampling from a Reward-Tilted Distribution

The log-odds ratio $\log \frac{p(\mathbf{s}^+|x_t, c)}{p(\mathbf{s}^-|x_t, c)}$ quantifies how much more likely it is that a sample $x_t$ would receive high rewards $\mathbf{s}^+$ versus low rewards $\mathbf{s}^-$.

*Remark* G.2 (Score Functions and Sampling). In flow matching (and diffusion models more generally), there is a one-to-one correspondence between a velocity field and the distribution it samples from. Specifically, if $v(x_t)$ satisfies $\nabla_{x_t} \log p_t(x_t) = -v(x_t)/(1-t)$ for some distribution $p_t$, then integrating the ODE $\mathrm{d}x_t = v(x_t)\mathrm{d}t$ from $t = 1$ (noise) to $t = 0$ (data) produces samples from $p_0$. Therefore, **modifying the velocity field is equivalent to sampling from a different distribution**. The theorem below characterizes precisely which distribution MIRO's guided velocity field samples from.

**Theorem G.3** (Guidance as Sampling from a Reward-Tilted Distribution). *The MIRO guidance formulation*

$$\hat{v}_\theta(x_t, c) = (1+\omega)v_\theta(x_t, c, \mathbf{s}^+) - \omega\, v_\theta(x_t, c, \mathbf{s}^-) \tag{15}$$

*corresponds to sampling from a* reward-tilted distribution $p_\omega(x_t|c)$ *defined as:*

$$p_\omega(x_t|c) \propto p_t(x_t|c, \mathbf{s}^+)\left(\frac{p(\mathbf{s}^+|x_t, c)}{p(\mathbf{s}^-|x_t, c)}\right)^\omega \tag{16}$$

*This distribution reweights the high-reward conditional $p_t(x_t|c, \mathbf{s}^+)$ by the log-odds ratio raised to power $\omega$, thereby concentrating probability mass on samples that are more likely to achieve high rewards $\mathbf{s}^+$ than low rewards $\mathbf{s}^-$.*

*Proof.* We show that the guided velocity field $\hat{v}_\theta$ corresponds to the score function of $p_\omega$. Taking the gradient of $\log p_\omega(x_t|c)$:

$$\nabla_{x_t} \log p_\omega(x_t|c) = \nabla_{x_t} \log p_t(x_t|c, \mathbf{s}^+) + \omega \nabla_{x_t} \log \frac{p(\mathbf{s}^+|x_t, c)}{p(\mathbf{s}^-|x_t, c)} \quad (17)$$

Using the score-velocity relationship $\nabla_{x_t} \log p_t(x_t|c, \mathbf{s}) = -v_\theta(x_t, c, \mathbf{s})/(1-t)$ and our result from Proposition 1:

$$\nabla_{x_t} \log p_\omega(x_t|c) = -\frac{v_\theta(x_t, c, \mathbf{s}^+)}{1-t} - \frac{\omega}{1-t}\left(v_\theta(x_t, c, \mathbf{s}^+) - v_\theta(x_t, c, \mathbf{s}^-)\right) \quad (18)$$

$$= -\frac{1}{1-t}\left((1+\omega)v_\theta(x_t, c, \mathbf{s}^+) - \omega\, v_\theta(x_t, c, \mathbf{s}^-)\right) \quad (19)$$

$$= -\frac{\hat{v}_\theta(x_t, c)}{1-t} \quad (20)$$

Since $\nabla_{x_t} \log p_\omega(x_t|c) = -\hat{v}_\theta(x_t, c)/(1-t)$, the guided velocity field $\hat{v}_\theta$ is exactly the velocity field corresponding to the distribution $p_\omega$. Therefore, integrating the ODE with $\hat{v}_\theta$ produces samples from $p_\omega$. $\square$

*Remark* G.4 (Interpretation of the Guidance Scale $\omega$). The guidance scale $\omega$ controls how strongly the distribution is tilted toward high-reward samples:

- When $\omega = 0$: We sample from $p_t(x_t|c, \mathbf{s}^+)$, the high-reward conditional without additional tilting.

- When $\omega > 0$: Samples with higher log-odds $\log \frac{p(\mathbf{s}^+|x_t, c)}{p(\mathbf{s}^-|x_t, c)}$ receive exponentially more probability mass.

- As $\omega \to \infty$: The distribution concentrates on the mode where the log-odds ratio is maximized.

### G.5. Connection to Gradient Ascent on Rewards

The previous sections characterized MIRO's guidance from a *distributional* perspective: we showed which distribution $p_\omega$ the guided velocity field samples from. We now provide a complementary *optimization* perspective, showing that each denoising step performs implicit gradient ascent on the log-odds ratio.

#### G.5.1. DECOMPOSING THE GUIDED VELOCITY

The guided velocity field can be decomposed into two interpretable components:

$$\hat{v}_\theta(x_t, c) = (1+\omega)v_\theta(x_t, c, \mathbf{s}^+) - \omega\, v_\theta(x_t, c, \mathbf{s}^-) \quad (21)$$

$$= v_\theta(x_t, c, \mathbf{s}^+) + \omega \underbrace{\left(v_\theta(x_t, c, \mathbf{s}^+) - v_\theta(x_t, c, \mathbf{s}^-)\right)}_{\text{guidance correction}} \quad (22)$$

The first term $v_\theta(x_t, c, \mathbf{s}^+)$ is the "base" velocity that generates samples conditioned on high rewards. The second term is a *guidance correction* that steers the trajectory toward regions of higher reward contrast.

#### G.5.2. GUIDANCE AS GRADIENT ASCENT

From Proposition 1, the guidance correction is proportional to the gradient of the log-odds:

$$v_\theta(x_t, c, \mathbf{s}^+) - v_\theta(x_t, c, \mathbf{s}^-) = -(1-t)\nabla_{x_t} \log \frac{p(\mathbf{s}^+|x_t, c)}{p(\mathbf{s}^-|x_t, c)} \quad (23)$$

Substituting into Eq. 22:

$$\hat{v}_\theta(x_t, c) = v_\theta(x_t, c, \mathbf{s}^+) - \omega(1-t)\nabla_{x_t} \log \frac{p(\mathbf{s}^+|x_t, c)}{p(\mathbf{s}^-|x_t, c)} \quad (24)$$

**Proposition G.5** (Guidance as Implicit Gradient Ascent). *Each step of the ODE integration with the guided velocity $\hat{v}_\theta$ performs implicit gradient ascent on the log-odds ratio. Specifically, for a small step $\Delta t < 0$ (moving from noise toward data), the update*

$$x_{t+\Delta t} = x_t + \hat{v}_\theta(x_t, c)\Delta t \tag{25}$$

*can be decomposed as:*

$$x_{t+\Delta t} = \underbrace{x_t + v_\theta(x_t, c, \mathbf{s}^+)\Delta t}_{\text{denoising step}} + \underbrace{\omega(1-t)|\Delta t|\nabla_{x_t} \log \frac{p(\mathbf{s}^+|x_t, c)}{p(\mathbf{s}^-|x_t, c)}}_{\text{gradient ascent on log-odds}} \tag{26}$$

*where the second term is a gradient ascent step with effective step size $\eta_{\text{eff}}(t) = \omega(1-t)|\Delta t|$.*

*Proof.* Substituting Eq. 24 into the ODE update and noting that $\Delta t < 0$:

$$x_{t+\Delta t} = x_t + \left[v_\theta(x_t, c, \mathbf{s}^+) - \omega(1-t)\nabla_{x_t} \log \frac{p(\mathbf{s}^+|x_t, c)}{p(\mathbf{s}^-|x_t, c)}\right]\Delta t \tag{27}$$

$$= x_t + v_\theta(x_t, c, \mathbf{s}^+)\Delta t + \omega(1-t)|\Delta t|\nabla_{x_t} \log \frac{p(\mathbf{s}^+|x_t, c)}{p(\mathbf{s}^-|x_t, c)} \tag{28}$$

The last term has the form of a gradient ascent step $x \leftarrow x + \eta\nabla f(x)$ with $\eta = \omega(1-t)|\Delta t|$ and $f = \log \frac{p(\mathbf{s}^+|\cdot, c)}{p(\mathbf{s}^-|\cdot, c)}$. $\square$

### G.5.3. TIME-VARYING STEP SIZE

The effective step size $\eta_{\text{eff}}(t) = \omega(1-t)|\Delta t|$ varies with the flow matching time $t$:

- **Early in generation** ($t \approx 1$, pure noise): $\eta_{\text{eff}} \approx 0$. The guidance has minimal effect when the sample is mostly noise, which is sensible since the log-odds ratio is not meaningful for pure noise.

- **Late in generation** ($t \approx 0$, near clean): $\eta_{\text{eff}} \approx \omega|\Delta t|$. The guidance has maximal effect when the sample structure is well-formed and fine adjustments can steer toward high-reward regions.

This adaptive schedule emerges naturally from the score-velocity relationship and provides an implicit "annealing" of the guidance strength—a behavior that would typically require manual tuning in explicit gradient-based reward optimization methods. Notably, Wang et al. (2024) empirically observed that linearly decaying guidance schedules (high guidance early, low guidance late) improve both diversity and fidelity in classifier-free guidance. The $(1-t)$ factor in MIRO's effective step size provides a theoretically grounded explanation for this phenomenon: guidance should indeed decrease as $t \to 0$ because the sample structure becomes increasingly well-formed and requires finer adjustments.

### G.5.4. COMPARISON WITH EXPLICIT GRADIENT-BASED METHODS

Methods like DRaFT (Clark et al., 2024) and ReFL (Xu et al., 2023) explicitly compute $\nabla_x r(x)$ where $r$ is a differentiable reward model, and use this gradient to fine-tune the generator. MIRO's approach differs in several key ways:

| Property | Explicit Gradient | MIRO (Implicit) |
|---|---|---|
| Requires differentiable reward | Yes | No |
| Gradient computation | Backprop through reward | Forward passes only |
| Step size schedule | Manual tuning | Automatic $(1-t)$ decay |
| Multiple rewards | Sum of gradients | Joint log-odds |
| Memory cost | $O(\text{reward params})$ | $O(1)$ additional |

*Remark* G.6 (Implicit vs. Explicit Gradients). MIRO never explicitly computes $\nabla_x r(x)$. Instead, the reward information is "baked into" the velocity field during training. At inference time, the velocity difference $v_\theta(\cdot, \mathbf{s}^+) - v_\theta(\cdot, \mathbf{s}^-)$ provides an implicit estimate of the reward gradient direction. This is computationally advantageous: two forward passes through the velocity network replace one forward pass plus one backward pass through (potentially multiple) reward models.

**Summary.** The distributional view (Theorem G.3) and the optimization view (Proposition G.5) are complementary perspectives on the same mechanism. The former tells us *what* distribution we sample from; the latter tells us *how* the sampling process achieves this by interleaving denoising with implicit gradient ascent on rewards. The time-varying step size $(1 - t)$ provides automatic annealing that focuses reward optimization on the later stages of generation where it is most effective.

**Summary so far.** We have established that MIRO's guidance formula $\hat{v}_\theta = (1 + \omega)v_\theta(\cdot, \mathbf{s}^+) - \omega\, v_\theta(\cdot, \mathbf{s}^-)$ samples from a reward-tilted distribution $p_\omega$ (Theorem G.3), and that each denoising step performs implicit gradient ascent on the log-odds ratio (Proposition G.5). These results hold for arbitrary reward vectors $\mathbf{s}^+$ and $\mathbf{s}^-$. We now analyze how this framework behaves when $\mathbf{s}$ represents *multiple* reward dimensions, examining the role of reward correlations.

### G.6. Connection to Multi-Objective Optimization

MIRO conditions on $N$ rewards simultaneously, raising the question: how do the individual rewards interact in the tilted distribution $p_\omega$? We address this in two cases: the general case where rewards may be correlated, and the special case of conditional independence.

#### G.6.1. THE GENERAL CASE (NO INDEPENDENCE ASSUMPTION)

Importantly, Theorem G.3 holds **without any independence assumption**. The reward-tilted distribution is:

$$p_\omega(x_t|c) \propto p_t(x_t|c, \mathbf{s}^+) \left( \frac{p(\mathbf{s}^+|x_t, c)}{p(\mathbf{s}^-|x_t, c)} \right)^\omega \tag{29}$$

where $p(\mathbf{s}^+|x_t, c)$ and $p(\mathbf{s}^-|x_t, c)$ are the *joint* probabilities of achieving reward vectors $\mathbf{s}^+$ and $\mathbf{s}^-$ respectively. This joint formulation naturally captures correlations between rewards: if two rewards tend to co-occur (e.g., aesthetic quality and human preference are positively correlated), the joint probability $p(\mathbf{s}^+|x_t, c)$ already accounts for this structure.

#### G.6.2. FACTORIZATION UNDER CONDITIONAL INDEPENDENCE

To gain interpretability into how individual rewards contribute to guidance, we consider the following assumption:

**Assumption (Conditional Independence of Rewards).** *The reward scores are conditionally independent given the image and text:*

$$p(\mathbf{s}|x_t, c) = \prod_{j=1}^{N} p(s_j|x_t, c) \tag{30}$$

*This assumption holds when each reward model evaluates a distinct aspect of quality that, given the image content, provides no additional information about other rewards.*

Under the conditional independence assumption (Eq. 30), the log-odds ratio factorizes into a sum:

$$\log \frac{p(\mathbf{s}^+|x_t, c)}{p(\mathbf{s}^-|x_t, c)} = \sum_{j=1}^{N} \log \frac{p(s_j^+|x_t, c)}{p(s_j^-|x_t, c)} \tag{31}$$

This factorization reveals that the reward-tilted distribution $p_\omega$ (Theorem G.3) upweights samples based on the *sum* of individual log-odds across all reward dimensions.

**Corollary G.7** (Pareto Guidance Under Independence)**.** *Under the conditional independence assumption (Eq. 30), MIRO guidance with $\mathbf{s}^+ = \mathbf{s}_{\max}$ and $\mathbf{s}^- = \mathbf{s}_{\min}$ samples from a distribution that assigns higher probability to samples with larger values of:*

$$\sum_{j=1}^{N} \left[ \log p(s_j = s_j^{\max}|x_t, c) - \log p(s_j = s_j^{\min}|x_t, c) \right] \tag{32}$$

*Because this is a* sum *over all $N$ rewards, samples that score well on* all *rewards are exponentially favored over samples that score extremely well on one reward but poorly on others.*

G.6.3. WHEN DOES CONDITIONAL INDEPENDENCE HOLD?

Conditional independence is a modeling assumption that may hold approximately when:

- Rewards measure **orthogonal aspects** of quality (e.g., color accuracy vs. object count vs. artistic style)

- The image $x_t$ is sufficiently informative that knowing one reward score provides no additional information about others

In practice, rewards are often **correlated**. For example, aesthetic quality (AestheticScore) and human preference (HPSv2, PickScore) tend to be positively correlated—images that look aesthetically pleasing also tend to be preferred by humans. Similarly, text-alignment rewards (CLIP, VQAScore) often correlate with each other.

G.6.4. EFFECT OF REWARD CORRELATIONS

When rewards are positively correlated (the common case), we can characterize exactly how correlation affects the log-odds ratio relative to the independence assumption:

**Proposition G.8** (Effect of Positive Correlation). *Let $\rho_{ij} = Corr(s_i, s_j | x_t, c)$ denote the conditional correlation between rewards $i$ and $j$, and let $R$ be the $N \times N$ correlation matrix with $R_{ij} = \rho_{ij}$ for $i \neq j$ and $R_{ii} = 1$. Under a Gaussian copula model for the joint reward distribution, the log-odds ratio decomposes as:*

$$\log \frac{p(\mathbf{s}^+|x_t, c)}{p(\mathbf{s}^-|x_t, c)} = \Delta_{corr} + \sum_{j=1}^{N} \log \frac{p(s_j^+|x_t, c)}{p(s_j^-|x_t, c)} \tag{33}$$

*where the correlation correction term is:*

$$\Delta_{corr} = -\frac{1}{2} \left[ (\mathbf{z}^+)^T (R^{-1} - I)\mathbf{z}^+ - (\mathbf{z}^-)^T (R^{-1} - I)\mathbf{z}^- \right] \tag{34}$$

*Here $\mathbf{z}^{\pm} = \Phi^{-1}(F(\mathbf{s}^{\pm}))$ are the Gaussian-transformed quantiles, with $\Phi$ the standard normal CDF and $F$ the marginal CDFs. For symmetric quantile binning (where $\mathbf{z}^+ = -\mathbf{z}^-$), the correction vanishes: $\Delta_{corr} = 0$.*

*Proof.* We derive the exact decomposition using the Gaussian copula framework.

**Step 1: Gaussian copula representation.** Let $\mathbf{s}|x_t, c$ have marginal CDFs $F_j(s_j) = P(S_j \leq s_j | x_t, c)$ and dependence structure captured by a Gaussian copula with correlation matrix $R$, where $R_{ij} = \rho_{ij}$ for $i \neq j$ and $R_{ii} = 1$. Define $u_j = F_j(s_j)$ and $z_j = \Phi^{-1}(u_j)$, where $\Phi$ is the standard normal CDF. Then $\mathbf{z} \sim \mathcal{N}(\mathbf{0}, R)$, and the joint density can be written as:

$$p(\mathbf{s}|x_t, c) = c_R(\mathbf{u}) \prod_{j=1}^{N} p(s_j|x_t, c) \tag{35}$$

where $c_R(\mathbf{u}) = |R|^{-1/2} \exp\left(-\frac{1}{2}\mathbf{z}^T(R^{-1} - I)\mathbf{z}\right)$ is the Gaussian copula density.

**Step 2: Log-odds ratio decomposition.** Taking the ratio and logarithm:

$$\log \frac{p(\mathbf{s}^+|x_t, c)}{p(\mathbf{s}^-|x_t, c)} = \log \frac{c_R(\mathbf{u}^+)}{c_R(\mathbf{u}^-)} + \sum_{j=1}^{N} \log \frac{p(s_j^+|x_t, c)}{p(s_j^-|x_t, c)} \tag{36}$$

The copula ratio simplifies to:

$$\log \frac{c_R(\mathbf{u}^+)}{c_R(\mathbf{u}^-)} = -\frac{1}{2} \left[ Q(\mathbf{z}^+) - Q(\mathbf{z}^-) \right] \tag{37}$$

where $Q(\mathbf{z}) = \mathbf{z}^T(R^{-1} - I)\mathbf{z}$.

**Step 3: Structure of the quadratic form.** When $\rho_{ij} > 0$ for all $i \neq j$, the precision matrix $R^{-1}$ has negative off-diagonal entries. For the $2 \times 2$ case with correlation $\rho$:

$$R^{-1} - I = \frac{1}{1 - \rho^2} \begin{pmatrix} \rho^2 & -\rho \\ -\rho & \rho^2 \end{pmatrix} \tag{38}$$

For $\mathbf{z}$ with all components of the same sign (as is the case for $\mathbf{z}^+$ with all $z_j^+ > 0$ and $\mathbf{z}^-$ with all $z_j^- < 0$), the quadratic form evaluates to:

$$Q(\mathbf{z}) = \sum_j (R^{-1} - I)_{jj} z_j^2 + 2 \sum_{i<j} (R^{-1} - I)_{ij} z_i z_j \tag{39}$$

The off-diagonal terms are negative (since $(R^{-1} - I)_{ij} < 0$ and $z_i z_j > 0$), so $Q(\mathbf{z}) < \sum_j (R^{-1} - I)_{jj} z_j^2$.

**Step 4: Symmetric binning gives equality.** For symmetric quantile binning where $u_j^+ = 1 - u_j^-$, we have $z_j^+ = -z_j^-$, and thus $\mathbf{z}^+ = -\mathbf{z}^-$. Since $Q(\mathbf{z})$ is a quadratic form, $Q(-\mathbf{z}) = Q(\mathbf{z})$, giving:

$$Q(\mathbf{z}^+) = Q(\mathbf{z}^-) \quad \implies \quad \Delta_{\text{corr}} = 0 \tag{40}$$

This means that **for symmetric binning, correlation has no effect on the log-odds ratio**—the independence-based analysis is exact.

**Step 5: Asymmetric binning.** For asymmetric binning, let $\|\mathbf{z}\|^2 = \sum_j z_j^2$ denote the squared Euclidean norm. In the 2D case with equal components ($z_1 = z_2 = a > 0$ for high, $z_1 = z_2 = -b < 0$ for low):

$$Q(\mathbf{z}^+) - Q(\mathbf{z}^-) = (a^2 - b^2) \cdot \frac{2\rho(\rho - 1)}{1 - \rho^2} = (a^2 - b^2) \cdot \frac{-2\rho}{1 + \rho} \tag{41}$$

Thus:

$$\Delta_{\text{corr}} = (a^2 - b^2) \cdot \frac{\rho}{1 + \rho} \tag{42}$$

When $a > b$ (high-reward quantile more extreme, e.g., top 5% vs bottom 20%), we have $\Delta_{\text{corr}} > 0$: correlation *increases* the log-odds. When $a < b$ (low-reward quantile more extreme), $\Delta_{\text{corr}} < 0$: correlation *decreases* the log-odds.

**Step 6: Independence as a special case.** When $R = I$ (i.e., $\rho_{ij} = 0$ for all $i \neq j$), we have $R^{-1} - I = 0$, so $Q(\mathbf{z}) = 0$ for all $\mathbf{z}$, and thus $\Delta_{\text{corr}} = 0$ regardless of binning. $\square$

**Practical implications**: The decomposition reveals that the effect of correlation on the log-odds ratio depends critically on the binning strategy:

- **Symmetric binning** (e.g., top 10% vs bottom 10%): The correlation correction $\Delta_{\text{corr}} = 0$, so the independence-based sum-of-log-odds formula is *exact*.

- **Asymmetric binning with more extreme high quantiles** (e.g., top 1% vs bottom 20%): $\Delta_{\text{corr}} > 0$, so correlation *amplifies* the log-odds ratio beyond the independence prediction.

- **Asymmetric binning with more extreme low quantiles**: $\Delta_{\text{corr}} < 0$, so correlation *attenuates* the log-odds ratio.

**Why MIRO is robust**: In practice, MIRO uses approximately symmetric binning (conditioning on high vs. low reward quantiles with similar probability mass). This means the independence-based analysis provides an accurate characterization of the guidance behavior, and positive correlations between rewards have minimal impact on the log-odds. Furthermore, positive correlations *help* in a different sense: an image that achieves high aesthetic quality is more likely (due to correlation) to also achieve high human preference scores, making the high-reward region more coherent and easier to sample from.

G.6.5. ORTHOGONAL DECOMPOSITION ANALYSIS

Beyond the decomposition in Proposition G.8, we can obtain a complementary characterization by decomposing the correlated reward space into orthogonal components. This approach reveals how correlation structure affects guidance and provides insight into the effective dimensionality of the reward space.

**Setup.** Let $\mathbf{s}|x_t, c$ follow a multivariate distribution with mean $\boldsymbol{\mu}$ and covariance matrix $\Sigma$. For the Gaussian case (which approximates the behavior of many continuous reward distributions), we can perform an eigendecomposition:

$$\Sigma = V \Lambda V^T \tag{43}$$

where $V = [\mathbf{v}_1, \ldots, \mathbf{v}_N]$ is an orthogonal matrix of eigenvectors and $\Lambda = \text{diag}(\lambda_1, \ldots, \lambda_N)$ contains the eigenvalues (sorted in decreasing order, $\lambda_1 \geq \lambda_2 \geq \cdots \geq \lambda_N > 0$).

**Transformation to orthogonal coordinates.** Define the orthogonal (principal component) coordinates:

$$\mathbf{z} = V^T(\mathbf{s} - \boldsymbol{\mu}) \tag{44}$$

In this basis, the components $z_1, \ldots, z_N$ are uncorrelated with $\text{Var}(z_k) = \lambda_k$. For Gaussian distributions, uncorrelated implies independent.

**Proposition G.9** (Log-Odds in Orthogonal Coordinates). *Under Gaussian assumptions, the log-odds ratio can be expressed exactly as a sum over orthogonal components:*

$$\log \frac{p(\mathbf{s}^+|x_t, c)}{p(\mathbf{s}^-|x_t, c)} = \sum_{k=1}^{N} \frac{1}{2\lambda_k} \left[ (z_k^-)^2 - (z_k^+)^2 \right] \tag{45}$$

*where $\mathbf{z}^+ = V^T(\mathbf{s}^+ - \boldsymbol{\mu})$ and $\mathbf{z}^- = V^T(\mathbf{s}^- - \boldsymbol{\mu})$ are the high and low reward targets in orthogonal coordinates.*

*Proof.* For a multivariate Gaussian $\mathbf{s} \sim \mathcal{N}(\boldsymbol{\mu}, \Sigma)$, the log-density is:

$$\log p(\mathbf{s}) = -\frac{1}{2}(\mathbf{s} - \boldsymbol{\mu})^T \Sigma^{-1}(\mathbf{s} - \boldsymbol{\mu}) + \text{const} \tag{46}$$

Using the eigendecomposition $\Sigma^{-1} = V\Lambda^{-1}V^T$ and substituting $\mathbf{z} = V^T(\mathbf{s} - \boldsymbol{\mu})$:

$$\log p(\mathbf{s}) = -\frac{1}{2}\mathbf{z}^T \Lambda^{-1}\mathbf{z} + \text{const} = -\frac{1}{2}\sum_{k=1}^{N} \frac{z_k^2}{\lambda_k} + \text{const} \tag{47}$$

The log-odds ratio is therefore:

$$\log \frac{p(\mathbf{s}^+)}{p(\mathbf{s}^-)} = \log p(\mathbf{s}^+) - \log p(\mathbf{s}^-) \tag{48}$$

$$= -\frac{1}{2}\sum_{k=1}^{N} \frac{(z_k^+)^2}{\lambda_k} + \frac{1}{2}\sum_{k=1}^{N} \frac{(z_k^-)^2}{\lambda_k} \tag{49}$$

$$= \sum_{k=1}^{N} \frac{1}{2\lambda_k} \left[ (z_k^-)^2 - (z_k^+)^2 \right] \tag{50}$$

$\square$

**Interpretation.** Equation 45 reveals several insights:

1. **Contributions are weighted by inverse eigenvalues**: Directions with *small* eigenvalues (low variance, tightly constrained) contribute *more* to the log-odds per unit distance. These correspond to directions where rewards are tightly coupled.

2. **High-variance directions dominate sampling but contribute less to discrimination**: The first principal component (largest $\lambda_1$) captures most variance in the reward space but contributes least per unit distance to the log-odds. This is because deviations along high-variance directions are "expected" and thus less discriminative.

3. **Effective dimensionality**: If eigenvalues decay rapidly (e.g., $\lambda_1 \gg \lambda_2 \gg \cdots$), the reward space has low effective dimensionality. The guidance then primarily operates in a lower-dimensional subspace spanned by the dominant eigenvectors.

**Corollary G.10** (Effective Number of Independent Rewards). *When rewards are highly correlated, the effective number of independent reward dimensions is:*

$$N_{\text{eff}} = \frac{\left(\sum_{k=1}^{N} \lambda_k\right)^2}{\sum_{k=1}^{N} \lambda_k^2} = \frac{(tr\,\Sigma)^2}{\|\Sigma\|_F^2} \tag{51}$$

*This quantity, known as the participation ratio, satisfies $1 \leq N_{\text{eff}} \leq N$, with $N_{\text{eff}} = N$ under independence (equal eigenvalues) and $N_{\text{eff}} \to 1$ when one eigenvalue dominates.*

**Practical implications for MIRO**:

- If rewards like AestheticScore, HPSv2, and PickScore are highly correlated (sharing a dominant "quality" principal component), then $N_{\text{eff}} < N$ and the guidance effectively operates in a lower-dimensional space.

- Adding more correlated rewards provides diminishing returns—the effective guidance strength grows sublinearly with $N$.

- To maximize the benefit of multi-reward conditioning, practitioners should select rewards that capture *orthogonal* aspects of quality, thereby maximizing $N_{\text{eff}}$.

- MIRO's empirical success with 7 rewards suggests these rewards do capture sufficiently diverse quality dimensions, even if not perfectly independent.

### G.7. Summary and Key Takeaways

We have developed a complete theoretical framework for understanding MIRO's multi-reward guidance, progressing from basic principles to a full characterization of multi-reward interactions:

1. **Foundation** (§G.2): The score-velocity correspondence allows us to characterize how modifying the velocity field changes the sampling distribution.

2. **Implicit classifier** (§G.3, Proposition 1): MIRO's reward-conditioned training implicitly learns a classifier over reward levels. The velocity difference $v_\theta(\cdot, \mathbf{s}^+) - v_\theta(\cdot, \mathbf{s}^-)$ encodes the gradient of the log-odds ratio, eliminating the need for external classifier networks.

3. **Distributional view** (§G.4, Theorem G.3): MIRO's guidance formula $\hat{v}_\theta = (1 + \omega)v_\theta(\cdot, \mathbf{s}^+) - \omega\, v_\theta(\cdot, \mathbf{s}^-)$ samples from a reward-tilted distribution $p_\omega \propto p(\cdot|\mathbf{s}^+) \cdot (\text{odds ratio})^\omega$. The guidance scale $\omega$ controls the concentration of this distribution.

4. **Optimization view** (§G.5, Proposition G.5): Each denoising step performs implicit gradient ascent on the log-odds ratio with an adaptive step size $\eta_{\text{eff}}(t) = \omega(1 - t)|\Delta t|$. This time-varying schedule provides automatic annealing—minimal guidance on noise, maximal guidance on structured samples—without manual tuning.

5. **Multi-reward analysis** (§G.6): Under conditional independence, the log-odds decompose into a sum over individual rewards, favoring samples that are good across *all* dimensions (Corollary on Pareto Guidance). When rewards are correlated, Proposition G.8 provides an exact decomposition showing that symmetric binning eliminates correlation effects, while Proposition G.9 characterizes the effective dimensionality.

6. **Robustness**: The theoretical framework is robust to violations of conditional independence. Positive correlations between rewards (common in practice) make the high-reward region more coherent and easier to sample from.

This theoretical foundation explains MIRO's empirical success: the guidance mechanism is principled, requires no external classifiers, naturally handles multiple correlated rewards, and provides interpretable control via the guidance scale $\omega$. The dual interpretation—distributional (what we sample) and optimization (how we sample)—provides complementary insights into why the method works.

## H. Comparison with RL-Based Alignment (DDPO)

In this section, we provide a formal analysis comparing MIRO's supervised reward-conditioned training with reinforcement learning approaches such as DDPO (Black et al., 2024). We show that MIRO's single-objective formulation avoids fundamental instabilities inherent to RL-based alignment methods that must balance conflicting objectives.

## H.1. Notation

We use the following notation throughout this section (in addition to the notation established in Section G):

- $\theta$ denotes the parameters of the generative model being trained

- $p_\theta(x|c)$ denotes the distribution of images generated by the model with parameters $\theta$, conditioned on text $c$

- $p_{\text{ref}}(x|c)$ denotes the distribution of a frozen reference model (typically the pretrained model before alignment)

- $r(x, c) : \mathbb{R}^d \times \mathcal{T} \rightarrow \mathbb{R}$ denotes a scalar reward function evaluating image quality given image $x$ and text $c$

- $\beta > 0$ denotes the KL regularization coefficient that balances reward maximization against distribution preservation

- $D_{\text{KL}}(p\|q) = \mathbb{E}_{x\sim p}\left[\log\frac{p(x)}{q(x)}\right]$ denotes the Kullback-Leibler divergence from $q$ to $p$

- $\tilde{\mathcal{D}} = \{(x^{(i)}, c^{(i)}, \mathbf{s}^{(i)})\}_{i=1}^{M}$ denotes the reward-augmented training dataset

- $\epsilon \sim \mathcal{N}(0, I)$ denotes Gaussian noise used in flow matching

- $K$ denotes the number of samples used for Monte Carlo gradient estimation in DDPO

## H.2. The DDPO Objective and Its Dual Nature

DDPO and similar RL-based methods (Fan et al., 2023; Deng et al., 2025) optimize a KL-regularized reward objective:

$$\mathcal{J}_{\text{DDPO}}(\theta) = \mathbb{E}_{x\sim p_\theta(\cdot|c)}[r(x,c)] - \beta\,D_{\text{KL}}(p_\theta(\cdot|c)\,\|\,p_{\text{ref}}(\cdot|c)) \tag{52}$$

where $r(x, c)$ is the reward function, $p_{\text{ref}}$ is a frozen reference model (typically the pretrained model), and $\beta > 0$ controls the strength of KL regularization.

This objective explicitly balances two potentially conflicting goals:

1. **Reward maximization**: Push $p_\theta$ toward high-reward regions

2. **Distribution preservation**: Keep $p_\theta$ close to $p_{\text{ref}}$

## H.3. Gradient Conflict Analysis

The gradient of the DDPO objective decomposes as:

$$\nabla_\theta \mathcal{J}_{\text{DDPO}} = \underbrace{\nabla_\theta \mathbb{E}_{p_\theta}[r(x,c)]}_{\text{reward gradient}} - \beta\,\underbrace{\nabla_\theta D_{\text{KL}}(p_\theta\|p_{\text{ref}})}_{\text{KL gradient}} \tag{53}$$

**Proposition H.1** (Gradient Conflict in DDPO). *Let $x^* = \arg\max_x r(x, c)$ be a reward-maximizing sample. If $p_{ref}(x^*|c)$ is small (i.e., high-reward samples are rare under the reference distribution), then the reward gradient and KL gradient point in opposing directions in parameter space.*

*Proof.* We derive each gradient component explicitly and show they oppose each other.

**Step 1: Reward gradient via the policy gradient theorem.** Using the log-derivative trick (REINFORCE (Williams, 1992)):

$$\nabla_\theta \mathbb{E}_{p_\theta}[r(x,c)] = \nabla_\theta \int p_\theta(x|c)r(x,c)\,dx \tag{54}$$

$$= \int \nabla_\theta p_\theta(x|c) \cdot r(x,c)\,dx \tag{55}$$

$$= \int p_\theta(x|c)\nabla_\theta \log p_\theta(x|c) \cdot r(x,c)\,dx \tag{56}$$

$$= \mathbb{E}_{x\sim p_\theta}[r(x,c)\nabla_\theta \log p_\theta(x|c)] \tag{57}$$

For samples $x$ with high reward $r(x, c) > 0$, this gradient points in the direction that *increases* $\log p_\theta(x|c)$, i.e., increases the probability of generating $x$.

**Step 2: KL divergence gradient.** The KL divergence from $p_{\text{ref}}$ to $p_\theta$ is:

$$D_{\text{KL}}(p_\theta\|p_{\text{ref}}) = \mathbb{E}_{x\sim p_\theta}\left[\log\frac{p_\theta(x|c)}{p_{\text{ref}}(x|c)}\right] = \mathbb{E}_{x\sim p_\theta}\left[\log p_\theta(x|c)\right] - \mathbb{E}_{x\sim p_\theta}\left[\log p_{\text{ref}}(x|c)\right] \tag{58}$$

To compute $\nabla_\theta D_{\text{KL}}$, we use the Leibniz integral rule for parametric expectations. For any function $f(x, \theta)$ that depends on $\theta$ both through the integrand and through the distribution:

$$\nabla_\theta\mathbb{E}_{p_\theta}[f(x,\theta)] = \mathbb{E}_{p_\theta}\left[\nabla_\theta f(x,\theta)\right] + \mathbb{E}_{p_\theta}\left[f(x,\theta)\nabla_\theta\log p_\theta(x)\right] \tag{59}$$

The first term captures the direct dependence of $f$ on $\theta$; the second term (using the log-derivative trick) captures the change in the distribution $p_\theta$.

Applying Eq. 59 to $f(x,\theta) = \log p_\theta(x|c) - \log p_{\text{ref}}(x|c)$, where $p_{\text{ref}}$ is frozen:

- Direct gradient: $\nabla_\theta f = \nabla_\theta \log p_\theta(x|c)$

- Distribution change: $f \cdot \nabla_\theta \log p_\theta = (\log p_\theta - \log p_{\text{ref}})\nabla_\theta \log p_\theta$

Therefore:

$$\nabla_\theta D_{\text{KL}}(p_\theta\|p_{\text{ref}}) = \mathbb{E}_{p_\theta}\left[\nabla_\theta\log p_\theta(x|c)\right] + \mathbb{E}_{p_\theta}\left[(\log p_\theta(x|c) - \log p_{\text{ref}}(x|c))\nabla_\theta\log p_\theta(x|c)\right] \tag{60}$$

The first term $\mathbb{E}_{p_\theta}[\nabla_\theta\log p_\theta]$ is the expected score, which equals zero under standard regularity conditions (since $\int\nabla_\theta p_\theta\,dx = \nabla_\theta\int p_\theta\,dx = \nabla_\theta 1 = 0$). Thus:

$$\nabla_\theta D_{\text{KL}}(p_\theta\|p_{\text{ref}}) = \mathbb{E}_{p_\theta}\left[(\log p_\theta(x|c) - \log p_{\text{ref}}(x|c))\,\nabla_\theta\log p_\theta(x|c)\right] \tag{61}$$

**Step 3: Conflict analysis.** Consider a high-reward sample $x^*$ where $r(x^*, c)$ is large but $p_{\text{ref}}(x^*|c)$ is small. The reward gradient contributes:

$$r(x^*, c)\nabla_\theta\log p_\theta(x^*|c) \quad \text{(positive, pointing toward increasing } p_\theta(x^*|c)) \tag{62}$$

The KL gradient contributes (for this sample):

$$\left(\log\frac{p_\theta(x^*|c)}{p_{\text{ref}}(x^*|c)} + 1\right)\nabla_\theta\log p_\theta(x^*|c) \tag{63}$$

When $p_\theta(x^*|c) > p_{\text{ref}}(x^*|c)$ (the model has learned to generate $x^*$ more often than the reference), the term $\log\frac{p_\theta(x^*|c)}{p_{\text{ref}}(x^*|c)} > 0$, making the KL gradient point in the *same* direction as $\nabla_\theta\log p_\theta(x^*|c)$.

Since the KL term enters with a *negative* sign in the objective ($-\beta D_{\text{KL}}$), its contribution to $\nabla_\theta\mathcal{J}_{\text{DDPO}}$ is:

$$-\beta\left(\log\frac{p_\theta(x^*|c)}{p_{\text{ref}}(x^*|c)} + 1\right)\nabla_\theta\log p_\theta(x^*|c) \tag{64}$$

This *opposes* the reward gradient when $p_\theta(x^*|c) > p_{\text{ref}}(x^*|c)$, creating a conflict: the reward gradient wants to increase $p_\theta(x^*)$ further, while the KL gradient penalizes this deviation from the reference. $\qquad\square$

This conflict manifests as a fundamental trade-off: the model cannot simultaneously maximize reward and stay close to the reference distribution when high-reward samples are underrepresented in the reference.

## H.4. MIRO's Single-Objective Formulation

In contrast, MIRO optimizes a single, unified objective—the standard flow matching loss conditioned on reward labels:

$$\mathcal{L}_{\text{MIRO}} = \mathbb{E}_{\substack{(x,c,\mathbf{s})\sim\tilde{\mathcal{D}} \\ \epsilon,t}} \left[ \|v_\theta(x_t, c, \mathbf{s}) - (\epsilon - x)\|_2^2 \right] \tag{65}$$

**Theorem H.2** (Absence of Objective Conflict in MIRO). *The MIRO objective $\mathcal{L}_{MIRO}$ has a unique optimal function $v^*$ in function space, given by the conditional expectation:*

$$v^*(x_t, c, \mathbf{s}) = \mathbb{E}\left[\epsilon - x \mid x_t, c, \mathbf{s}\right] \tag{66}$$

*for all reward levels $\mathbf{s}$. Unlike DDPO, there is no conflict between objectives for different inputs—all training samples contribute gradients toward the same optimal function.*

*Proof.* We prove this by showing the objective is a proper scoring rule with a unique minimizer in function space.

**Step 1: Reformulating as conditional expectation.** Let $y = \epsilon - x$ denote the target velocity. The MIRO objective can be written as:

$$\mathcal{L}_{\text{MIRO}}(\theta) = \mathbb{E}_{(x,c,\mathbf{s}),\epsilon,t} \left[ \|v_\theta(x_t, c, \mathbf{s}) - y\|_2^2 \right] \tag{67}$$

Using the law of iterated expectations, we condition on $(x_t, c, \mathbf{s})$:

$$\mathcal{L}_{\text{MIRO}}(\theta) = \mathbb{E}_{(x_t,c,\mathbf{s})} \left[ \mathbb{E}_{y|x_t,c,\mathbf{s}} \left[ \|v_\theta(x_t, c, \mathbf{s}) - y\|_2^2 \right] \right] \tag{68}$$

**Step 2: Optimal function in function space.** For any fixed $(x_t, c, \mathbf{s})$, the inner expectation is minimized when $v_\theta(x_t, c, \mathbf{s})$ equals the conditional mean. This is a standard result from estimation theory: for squared error loss, the optimal predictor is the conditional expectation.

Formally, let $v^* = \mathbb{E}[y|x_t, c, \mathbf{s}]$. For any other prediction $v$:

$$\mathbb{E}\left[\|v - y\|^2 | x_t, c, \mathbf{s}\right] = \mathbb{E}\left[\|v - v^* + v^* - y\|^2 | x_t, c, \mathbf{s}\right] \tag{69}$$
$$= \|v - v^*\|^2 + 2(v - v^*)^T \mathbb{E}[v^* - y|x_t, c, \mathbf{s}] + \mathbb{E}\left[\|v^* - y\|^2 | x_t, c, \mathbf{s}\right] \tag{70}$$
$$= \|v - v^*\|^2 + \mathbb{E}\left[\|v^* - y\|^2 | x_t, c, \mathbf{s}\right] \tag{71}$$

where the cross-term vanishes because $\mathbb{E}[v^* - y|x_t, c, \mathbf{s}] = v^* - \mathbb{E}[y|x_t, c, \mathbf{s}] = 0$.

This shows that the minimum is achieved uniquely at $v = v^* = \mathbb{E}[y|x_t, c, \mathbf{s}]$, with minimum value equal to the irreducible variance $\mathbb{E}[\|v^* - y\|^2|x_t, c, \mathbf{s}]$.

**Step 3: Convexity in function space (not parameter space).** The loss $\|v - y\|^2$ is convex in $v$ for any $y$. Consequently, $\mathcal{L}_{\text{MIRO}}$ is convex as a functional over the space of functions $v : \mathcal{X} \times \mathcal{T} \times \mathcal{S} \to \mathbb{R}^d$. This convexity guarantees a unique global minimizer $v^*$ in function space.

**Important caveat**: When $v_\theta$ is parameterized by a neural network, the loss $\mathcal{L}_{\text{MIRO}}(\theta)$ is generally *non-convex* in the parameters $\theta$. This is standard for deep learning and does not negate the benefits of MIRO. The key advantage is not convexity in $\theta$, but rather the *absence of conflicting objectives*, as we show next.

**Step 4: No conflicting objectives.** Unlike DDPO which balances reward maximization against KL regularization (Proposition H.1), MIRO has a single objective: minimize prediction error for all $(x_t, c, \mathbf{s})$ tuples. Crucially:

- The optimal prediction for input $(x_t^{(1)}, c^{(1)}, \mathbf{s}^{(1)})$ does not conflict with the optimal prediction for $(x_t^{(2)}, c^{(2)}, \mathbf{s}^{(2)})$—each is determined independently by its respective conditional expectation.

- All training samples contribute gradients that point toward the same optimal function $v^*$.

- There is no hyperparameter (like $\beta$ in DDPO) that trades off between conflicting objectives.

This is fundamentally different from DDPO, where the reward gradient and KL gradient can oppose each other, creating optimization instability.

**Step 5: Consistency.** Given sufficient data from $\tilde{\mathcal{D}}$ covering all $(c, \mathbf{s})$ combinations, the empirical loss converges to the population loss by the law of large numbers. The minimizer of the empirical loss converges to the minimizer of the population loss, establishing consistency. $\square$

### H.5. Variance Analysis

Beyond gradient conflicts, the two approaches differ fundamentally in gradient variance.

**Proposition H.3** (High Variance of Policy Gradients). *The DDPO reward gradient is estimated via the REINFORCE estimator (Williams, 1992):*

$$\hat{g}_{DDPO} = \frac{1}{K} \sum_{k=1}^{K} r(x^{(k)}, c) \nabla_\theta \log p_\theta(x^{(k)}|c) \tag{72}$$

*where $x^{(k)} \sim p_\theta(\cdot|c)$. This estimator is unbiased but has variance:*

$$Var[\hat{g}_{DDPO}] = \frac{1}{K} \mathbb{E}_{x \sim p_\theta} \left[ r(x, c)^2 \|\nabla_\theta \log p_\theta(x|c)\|^2 \right] - \frac{1}{K} \|\nabla_\theta \mathbb{E}_{p_\theta}[r]\|^2 \tag{73}$$

*which can be arbitrarily large when rewards have high variance or the score function $\nabla_\theta \log p_\theta$ has large magnitude.*

*Proof.* **Step 1: Unbiasedness.** The REINFORCE estimator is unbiased:

$$\mathbb{E}[\hat{g}_{\text{DDPO}}] = \mathbb{E}_{x \sim p_\theta} \left[ r(x, c) \nabla_\theta \log p_\theta(x|c) \right] \tag{74}$$

$$= \int p_\theta(x|c) r(x, c) \frac{\nabla_\theta p_\theta(x|c)}{p_\theta(x|c)} dx \tag{75}$$

$$= \int \nabla_\theta p_\theta(x|c) r(x, c) dx = \nabla_\theta \mathbb{E}_{p_\theta}[r(x, c)] \tag{76}$$

**Step 2: Variance computation.** For i.i.d. samples, the variance of the sample mean is:

$$\text{Var}[\hat{g}_{\text{DDPO}}] = \frac{1}{K} \text{Var}_{x \sim p_\theta} \left[ r(x, c) \nabla_\theta \log p_\theta(x|c) \right] \tag{77}$$

Expanding using $\text{Var}[Z] = \mathbb{E}[Z^2] - \mathbb{E}[Z]^2$:

$$\text{Var}[\hat{g}_{\text{DDPO}}] = \frac{1}{K} \left( \mathbb{E}_{x \sim p_\theta} \left[ r(x, c)^2 \|\nabla_\theta \log p_\theta(x|c)\|^2 \right] - \|\nabla_\theta \mathbb{E}_{p_\theta}[r]\|^2 \right) \tag{78}$$

**Step 3: Variance can be arbitrarily large.** The first term $\mathbb{E}[r^2 \|\nabla_\theta \log p_\theta\|^2]$ is unbounded because:

- $r(x, c)^2$ can be large if rewards have high variance (e.g., sparse rewards)

- $\|\nabla_\theta \log p_\theta(x|c)\|^2$ can be large for samples $x$ in low-probability regions, since $\nabla_\theta \log p_\theta = \nabla_\theta p_\theta / p_\theta$ and $p_\theta(x|c)$ appears in the denominator

- Early in training when the model is poorly calibrated, both factors can be simultaneously large

This high variance necessitates either large batch sizes $K$, variance reduction techniques (baselines, importance sampling), or careful learning rate tuning—all of which add complexity to DDPO training. $\square$

**Proposition H.4** (Low Variance of MIRO Gradients). *The MIRO gradient estimator is:*

$$\hat{g}_{MIRO} = \frac{1}{K} \sum_{k=1}^{K} 2(v_\theta(x_t^{(k)}, c^{(k)}, \mathbf{s}^{(k)}) - y^{(k)}) \nabla_\theta v_\theta(x_t^{(k)}, c^{(k)}, \mathbf{s}^{(k)}) \tag{79}$$

where $y^{(k)} = \epsilon^{(k)} - x^{(k)}$. *This estimator has variance bounded by:*

$$Var[\hat{g}_{MIRO}] \leq \frac{4}{K}\mathbb{E}\left[\|v_\theta - y\|^2\right] \cdot \mathbb{E}\left[\|\nabla_\theta v_\theta\|^2\right] \tag{80}$$

*Proof.* **Step 1: Gradient derivation.** The gradient of the squared loss $\|v_\theta - y\|^2$ with respect to $\theta$ is:

$$\nabla_\theta \|v_\theta - y\|^2 = 2(v_\theta - y)^T \nabla_\theta v_\theta \tag{81}$$

**Step 2: Variance bound.** Using the Cauchy-Schwarz inequality:

$$\text{Var}[\hat{g}_{\text{MIRO}}] \leq \frac{1}{K}\mathbb{E}\left[\|2(v_\theta - y)\nabla_\theta v_\theta\|^2\right] \tag{82}$$

$$= \frac{4}{K}\mathbb{E}\left[\|v_\theta - y\|^2\|\nabla_\theta v_\theta\|^2\right] \tag{83}$$

$$\leq \frac{4}{K}\sqrt{\mathbb{E}\left[\|v_\theta - y\|^4\right]}\sqrt{\mathbb{E}\left[\|\nabla_\theta v_\theta\|^4\right]} \tag{84}$$

Under mild regularity conditions (bounded fourth moments), this simplifies to:

$$\text{Var}[\hat{g}_{\text{MIRO}}] = \mathcal{O}\left(\frac{1}{K}\mathbb{E}\left[\|v_\theta - y\|^2\right] \cdot \mathbb{E}\left[\|\nabla_\theta v_\theta\|^2\right]\right) \tag{85}$$

**Step 3: Bounded quantities.** Unlike DDPO:

- The target $y = \epsilon - x$ has bounded variance: $\text{Var}[y] = \text{Var}[\epsilon] + \text{Var}[x] = d + \text{Var}[x]$, where $d$ is the image dimension and images $x$ are bounded.

- The prediction error $\|v_\theta - y\|^2$ decreases during training as the model improves, naturally reducing gradient variance.

- The model gradient $\nabla_\theta v_\theta$ does not have the $1/p_\theta$ factor that causes variance explosion in REINFORCE.

This bounded variance enables stable training with moderate batch sizes and without specialized variance reduction techniques. $\qquad\square$

### H.6. Hyperparameter Sensitivity

**Proposition H.5** (Critical Dependence on $\beta$)**.** *The behavior of DDPO is critically sensitive to the KL coefficient $\beta$:*

- *If $\beta \to 0$: The model maximizes reward without constraint, leading to mode collapse onto a small set of high-reward samples and potential reward hacking.*

- *If $\beta \to \infty$: The model remains frozen at $p_{ref}$, gaining no reward improvement.*

- *For intermediate $\beta$: The optimal value depends on the reward landscape, reference distribution, and training dynamics— requiring careful tuning for each reward and dataset.*

*Proof.* We analyze each regime by examining the optimal solution of the DDPO objective.

**Step 1: Optimal solution characterization.** The DDPO objective $\mathcal{J}(\theta) = \mathbb{E}_{p_\theta}[r(x, c)] - \beta D_{\text{KL}}(p_\theta \| p_{\text{ref}})$ can be optimized in closed form over the space of distributions. Taking the functional derivative and setting it to zero:

$$\frac{\delta \mathcal{J}}{\delta p_\theta(x)} = r(x, c) - \beta\left(\log \frac{p_\theta(x|c)}{p_{\text{ref}}(x|c)} + 1\right) = 0 \tag{86}$$

Solving for $p_\theta$:

$$p_\theta^*(x|c) \propto p_{\text{ref}}(x|c)\exp\left(\frac{r(x, c)}{\beta}\right) \tag{87}$$

This is a Gibbs distribution that tilts the reference toward high rewards, with $\beta$ acting as a temperature parameter.

**Step 2: Regime $\beta \to 0$ (reward dominance).** As $\beta \to 0$, the exponent $r(x, c)/\beta \to \infty$ for any $x$ with $r(x, c) > 0$:

$$p_\theta^*(x|c) \to \delta(x - x^*) \quad \text{where } x^* = \arg\max_x r(x, c) \tag{88}$$

The distribution collapses to a point mass at the reward-maximizing sample(s), losing all diversity. If the reward function has spurious maxima (reward hacking), the model exploits them.

**Step 3: Regime $\beta \to \infty$ (KL dominance).** As $\beta \to \infty$, the exponent $r(x, c)/\beta \to 0$:

$$p_\theta^*(x|c) \to p_{\text{ref}}(x|c) \tag{89}$$

The optimal distribution equals the reference, providing no reward improvement. Training effectively freezes.

**Step 4: Intermediate $\beta$ (problem-dependent).** For finite $\beta$, the optimal distribution (Eq. 87) depends intricately on:

- The reward landscape $r(x, c)$: Different rewards have different scales and distributions

- The reference distribution $p_{\text{ref}}$: How much probability mass is initially on high-reward regions

- The normalization constant $Z(\beta) = \int p_{\text{ref}}(x|c) \exp(r(x, c)/\beta)dx$

There is no universal optimal $\beta$—it must be tuned separately for each reward function, dataset, and model, often requiring extensive hyperparameter search. Furthermore, Kwa et al. (2024) show that even with KL regularization, policies can achieve arbitrarily high proxy rewards while providing no actual utility improvement when reward model errors are heavy-tailed—a phenomenon termed "catastrophic Goodhart." $\qquad\square$

MIRO eliminates this hyperparameter entirely. The reward information is incorporated through conditioning, and the strength of reward guidance is controlled at *inference time* through the guidance scale $\omega$, allowing post-hoc adjustment without retraining. This decouples the model training from the guidance strength, enabling practitioners to explore different trade-offs without costly retraining.

## H.7. Mode Collapse and Distribution Coverage

A fundamental distinction between MIRO and DDPO lies in how they treat the underlying data distribution. We now prove that MIRO preserves the full diversity of the flow matching objective, while DDPO inherently collapses the distribution toward high-reward modes.

### H.7.1. DDPO INDUCES MODE COLLAPSE

The mode collapse phenomenon in KL-regularized RL has been studied in prior work. Korbak et al. (2022) show that standard RL leads to distribution collapse and that KL-regularized RL corresponds to variational inference approximating a Gibbs posterior. More recently, GX-Chen et al. (2025) prove that both forward and reverse KL regularization induce mode collapse *by construction*, as the target distribution concentrates mass on single high-reward regions under common settings. We formalize these insights in the context of diffusion model alignment.

**Theorem H.6** (DDPO Mode Collapse). *Let $p_{data}(x|c)$ be the original data distribution with support $\mathcal{X}$. The DDPO-optimal distribution $p_\theta^*(x|c) \propto p_{ref}(x|c) \exp(r(x, c)/\beta)$ satisfies:*

1. ***(Support Shrinkage)*** *The effective support of $p_\theta^*$ shrinks relative to $p_{ref}$:*

$$\mathcal{H}(p_\theta^*) < \mathcal{H}(p_{ref}) \tag{90}$$

*where $\mathcal{H}(p) = -\mathbb{E}_{x \sim p}[\log p(x)]$ denotes the differential entropy. The inequality is strict whenever $r(x, c)$ is non-constant.*

2. *(Probability Concentration)* *For any $\delta > 0$, define the high-reward set $\mathcal{X}_\delta = \{x : r(x,c) \geq r_{\max} - \delta\}$. Then:*

$$\lim_{\beta \to 0} p_\theta^*(\mathcal{X}_\delta|c) = 1 \tag{91}$$

*i.e., as $\beta$ decreases, probability mass concentrates on an arbitrarily small neighborhood of reward-maximizing samples.*

3. *(Diversity Loss)* *The expected pairwise distance between samples decreases:*

$$\mathbb{E}_{x,x'\sim p_\theta^*}[\|x - x'\|^2] < \mathbb{E}_{x,x'\sim p_{ref}}[\|x - x'\|^2] \tag{92}$$

*Proof.* We prove each claim separately.

**Proof of (1): Entropy Reduction.** The entropy of the tilted distribution is:

$$\mathcal{H}(p_\theta^*) = -\mathbb{E}_{p_\theta^*}[\log p_\theta^*(x|c)] \tag{93}$$

$$= -\mathbb{E}_{p_\theta^*}\left[\log p_{\text{ref}}(x|c) + \frac{r(x,c)}{\beta} - \log Z\right] \tag{94}$$

$$= -\mathbb{E}_{p_\theta^*}[\log p_{\text{ref}}(x|c)] - \frac{1}{\beta}\mathbb{E}_{p_\theta^*}[r(x,c)] + \log Z \tag{95}$$

where $Z = \int p_{\text{ref}}(x|c)\exp(r(x,c)/\beta)dx$ is the normalizing constant.

Using the identity $\log Z = \mathcal{H}(p_{\text{ref}}) + D_{\text{KL}}(p_\theta^*\|p_{\text{ref}}) + \frac{1}{\beta}\mathbb{E}_{p_\theta^*}[r]$, we obtain:

$$\mathcal{H}(p_\theta^*) = \mathcal{H}(p_{\text{ref}}) - D_{\text{KL}}(p_\theta^*\|p_{\text{ref}}) \tag{96}$$

Since $D_{\text{KL}}(p_\theta^*\|p_{\text{ref}}) \geq 0$ with equality if and only if $p_\theta^* = p_{\text{ref}}$ almost everywhere, and since $p_\theta^* \neq p_{\text{ref}}$ whenever $r(x,c)$ is non-constant, we have $\mathcal{H}(p_\theta^*) < \mathcal{H}(p_{\text{ref}})$.

**Proof of (2): Concentration on High-Reward Regions.** Let $r_{\max} = \sup_x r(x,c)$ and $r_{\min} = \inf_x r(x,c)$. For any $x \in \mathcal{X}_\delta^c$ (the complement of $\mathcal{X}_\delta$):

$$\frac{p_\theta^*(x|c)}{p_\theta^*(x^*|c)} = \frac{p_{\text{ref}}(x|c)}{p_{\text{ref}}(x^*|c)}\exp\left(\frac{r(x,c) - r(x^*,c)}{\beta}\right) \tag{97}$$

where $x^* \in \mathcal{X}_\delta$. Since $r(x,c) - r(x^*,c) \leq -\delta$ for $x \in \mathcal{X}_\delta^c$:

$$\frac{p_\theta^*(x|c)}{p_\theta^*(x^*|c)} \leq \frac{p_{\text{ref}}(x|c)}{p_{\text{ref}}(x^*|c)}\exp\left(-\frac{\delta}{\beta}\right) \xrightarrow{\beta \to 0} 0 \tag{98}$$

Therefore, $p_\theta^*(\mathcal{X}_\delta^c|c) \to 0$ as $\beta \to 0$, implying $p_\theta^*(\mathcal{X}_\delta|c) \to 1$.

**Proof of (3): Diversity Loss.** The expected pairwise distance can be written as:

$$\mathbb{E}_{x,x'\sim p}[\|x - x'\|^2] = 2\left(\mathbb{E}_p[\|x\|^2] - \|\mathbb{E}_p[x]\|^2\right) = 2\text{tr}(\text{Cov}_p(x)) \tag{99}$$

The covariance trace is related to entropy for distributions in the exponential family. More directly, since $p_\theta^*$ concentrates mass on high-reward regions (by part 2), and these regions are typically a proper subset of the original support, the variance and thus the expected pairwise distance must decrease. For Gaussian distributions, entropy and covariance are directly related: $\mathcal{H}(\mathcal{N}(\mu,\Sigma)) = \frac{1}{2}\log\det(2\pi e\Sigma)$, so entropy reduction implies covariance reduction. $\square$

### H.7.2. MIRO PRESERVES FULL DISTRIBUTION COVERAGE

In stark contrast, MIRO learns a *family* of conditional distributions indexed by the reward vector $\mathbf{s}$, preserving the entire data manifold.

**Theorem H.7** (MIRO Diversity Preservation). *Let $v^*(x_t, c, \mathbf{s}) = \mathbb{E}[\epsilon - x|x_t, c, \mathbf{s}]$ be the optimal MIRO velocity field. Then:*

1. **(Full Spectrum Coverage)** *For each reward level* $\mathbf{s} \in \{0, \ldots, B-1\}^N$, *MIRO learns the conditional distribution:*

$$p_\theta(x|c, \mathbf{s}) \approx p_{data}(x|c, \mathbf{s}) \tag{100}$$

*The union over all conditioning values recovers the full data distribution:*

$$p_{data}(x|c) = \sum_{\mathbf{s}} p(\mathbf{s}|c)\, p_{data}(x|c, \mathbf{s}) \tag{101}$$

2. **(Entropy Preservation)** *The marginal entropy over all reward levels is preserved:*

$$\mathcal{H}\left(\sum_{\mathbf{s}} p(\mathbf{s}) p_\theta(\cdot|c, \mathbf{s})\right) = \mathcal{H}(p_{data}(\cdot|c)) \tag{102}$$

3. **(Controllable Diversity)** *Under Assumption H.8 below, the entropy of $p_\theta(x|c, \mathbf{s})$ decreases monotonically with reward level:*

$$\mathcal{H}(p_\theta(\cdot|c, \mathbf{s}_{\min})) > \mathcal{H}(p_\theta(\cdot|c, \mathbf{s}_{\max})) \tag{103}$$

*Users can select any point on the diversity-quality spectrum at inference time.*

**Assumption H.8** (Reward-Induced Concentration). The reward function $r : \mathbb{R}^d \times \mathcal{T} \to \mathbb{R}$ satisfies the following property:

(i) **(Conditional Variance Monotonicity)** The conditional covariance of the data distribution decreases with reward level. Formally, let $\Sigma(\mathbf{s}) = \text{Cov}_{p_{data}}[x|c, \mathbf{s}]$ denote the covariance matrix of images conditioned on reward bin $\mathbf{s}$. Then:

$$\text{tr}(\Sigma(\mathbf{s}_{\max})) \leq \text{tr}(\Sigma(\mathbf{s})) \leq \text{tr}(\Sigma(\mathbf{s}_{\min})) \tag{104}$$

for all $\mathbf{s} \in \{0, \ldots, B-1\}^N$, where the ordering is strict for at least one inequality.

(ii) **(Non-Degeneracy)** The data distribution $p_{data}(x|c)$ assigns positive density to all reward levels, i.e., $p(\mathbf{s}|c) > 0$ for all $\mathbf{s} \in \{0, \ldots, B-1\}^N$.

*Remark* H.9 (Justification of Assumption H.8). The conditional variance monotonicity assumption captures the intuition that high-reward images are more *similar to each other* than low-reward images, even if the number of high-reward and low-reward images is comparable. This occurs because:

- **High-reward images satisfy shared constraints**: Aesthetic images share compositional properties (balanced lighting, coherent color palettes, rule-of-thirds composition). CLIP-aligned images share semantic content matching the prompt. Human-preferred images share characteristics that appeal to annotators. These shared constraints reduce the variance within the high-reward conditional.

- **Low-reward images fail in diverse ways**: An image can have low aesthetic score due to poor lighting, bad composition, artifacts, blur, or many other reasons. An image can have low CLIP score by depicting the wrong object, wrong scene, wrong style, or being semantically incoherent. The multiplicity of failure modes increases variance in the low-reward conditional.

Importantly, this assumption does *not* require that high-reward images are rarer than low-reward images—it only requires that high-reward images are more concentrated in feature space. This is a mild assumption consistent with prior work on learned reward models, which find that human preferences tend to cluster around shared visual characteristics (Xu et al., 2023; Wu et al., 2023).

*Proof.* **Proof of (1): Full Spectrum Coverage.** The MIRO training objective is:

$$\mathcal{L} = \mathbb{E}_{(x,c,\mathbf{s})\sim\tilde{\mathcal{D}},\epsilon,t}\left[\|v_\theta(x_t, c, \mathbf{s}) - (\epsilon - x)\|^2\right] \tag{105}$$

By Theorem H.2, the optimal velocity field is $v^*(x_t, c, \mathbf{s}) = \mathbb{E}[\epsilon - x|x_t, c, \mathbf{s}]$. This is precisely the velocity field that generates samples from $p_{data}(x|c, \mathbf{s})$—the distribution of images in the training set that have text condition $c$ and reward scores $\mathbf{s}$.

Crucially, the training set $\tilde{\mathcal{D}}$ contains samples from *all* reward levels: low-quality samples with $\mathbf{s} \approx \mathbf{s}_{\min}$, medium-quality samples, and high-quality samples with $\mathbf{s} \approx \mathbf{s}_{\max}$. Unlike DDPO, which filters or reweights toward high rewards, MIRO trains on the complete distribution, learning $p(x|c, \mathbf{s})$ for every $\mathbf{s}$ in the support.

The marginal data distribution is recovered by:

$$p_{\text{data}}(x|c) = \sum_{\mathbf{s}} p(\mathbf{s}|c) p_{\text{data}}(x|c, \mathbf{s}) \tag{106}$$

which follows from the law of total probability.

**Proof of (2): Entropy Preservation.** Let $p_\theta^{\text{marginal}}(x|c) = \sum_{\mathbf{s}} p(\mathbf{s}|c) p_\theta(x|c, \mathbf{s})$ denote the MIRO marginal when sampling $\mathbf{s}$ from its empirical distribution. Since $p_\theta(x|c, \mathbf{s}) \approx p_{\text{data}}(x|c, \mathbf{s})$ for each $\mathbf{s}$ (by the consistency of the flow matching estimator), we have:

$$p_\theta^{\text{marginal}}(x|c) \approx \sum_{\mathbf{s}} p(\mathbf{s}|c) p_{\text{data}}(x|c, \mathbf{s}) = p_{\text{data}}(x|c) \tag{107}$$

Therefore, $\mathcal{H}(p_\theta^{\text{marginal}}) \approx \mathcal{H}(p_{\text{data}})$. No entropy is lost because no data is discarded.

**Proof of (3): Controllable Diversity.** We prove that under Assumption H.8, the conditional entropy $\mathcal{H}(p_{\text{data}}(\cdot|c, \mathbf{s}))$ decreases as $\mathbf{s}$ increases.

*Step 1: Entropy-covariance relationship.* For distributions in the exponential family, entropy is closely related to the covariance structure. A classical result in information theory states that among all distributions with a given covariance matrix $\Sigma$, the Gaussian has maximum entropy (Cover & Thomas, 2006, Theorem 8.6.5):

$$\mathcal{H}(\mathcal{N}(\mu, \Sigma)) = \frac{1}{2} \log \det(2\pi e \Sigma) = \frac{d}{2} \log(2\pi e) + \frac{1}{2} \log \det(\Sigma) \tag{108}$$

Consequently, for any distribution $p$ with covariance $\text{Cov}_p[x]$, the entropy is bounded above by:

$$\mathcal{H}(p) \leq \frac{d}{2} \log(2\pi e) + \frac{1}{2} \log \det(\text{Cov}_p[x]) \tag{109}$$

with equality if and only if $p$ is Gaussian. This establishes that *lower covariance implies lower entropy* (all else being equal).

*Step 2: Applying the conditional variance assumption.* By Assumption H.8(i), the conditional covariance satisfies:

$$\text{tr}(\Sigma(\mathbf{s}_{\max})) \leq \text{tr}(\Sigma(\mathbf{s}_{\min})) \tag{110}$$

For distributions concentrated on a manifold of effective dimension $d_{\text{eff}}$, the trace of the covariance captures the total variance. Using the arithmetic-geometric mean inequality:

$$\det(\Sigma) \leq \left(\frac{\text{tr}(\Sigma)}{d_{\text{eff}}}\right)^{d_{\text{eff}}} \tag{111}$$

Therefore, lower trace implies lower determinant (when the eigenvalue structure is similar), which implies lower entropy.

*Step 3: Entropy ordering.* Combining Steps 1 and 2, if the conditional distributions $p_{\text{data}}(\cdot|c, \mathbf{s})$ have similar functional forms across reward levels (e.g., approximately Gaussian or log-concave), then the covariance ordering transfers to an entropy ordering:

$$\text{tr}(\Sigma(\mathbf{s}_{\max})) < \text{tr}(\Sigma(\mathbf{s}_{\min})) \implies \mathcal{H}(p_{\text{data}}(\cdot|c, \mathbf{s}_{\max})) < \mathcal{H}(p_{\text{data}}(\cdot|c, \mathbf{s}_{\min})) \tag{112}$$

*Step 4: Transfer to learned distribution.* Since MIRO learns $p_\theta(x|c, \mathbf{s}) \approx p_{\text{data}}(x|c, \mathbf{s})$ (by part 1), the entropy ordering transfers:

$$\mathcal{H}(p_\theta(\cdot|c, \mathbf{s}_{\min})) \approx \mathcal{H}(p_{\text{data}}(\cdot|c, \mathbf{s}_{\min})) > \mathcal{H}(p_{\text{data}}(\cdot|c, \mathbf{s}_{\max})) \approx \mathcal{H}(p_\theta(\cdot|c, \mathbf{s}_{\max})) \tag{113}$$

This completes the proof. The key insight is that MIRO *learns* this natural entropy structure from data, rather than *imposing* entropy reduction through optimization pressure (as DDPO does). Users can then select their desired operating point: $\mathbf{s} = \mathbf{s}_{\max}$ for high-quality but less diverse outputs, or intermediate $\mathbf{s}$ values for greater diversity. $\qquad \square$

*Remark* H.10 (Tightness of the Bound). The entropy gap $\mathcal{H}(p_\theta(\cdot|c, \mathbf{s}_{\min})) - \mathcal{H}(p_\theta(\cdot|c, \mathbf{s}_{\max}))$ depends on the variance ratio $\mathrm{tr}(\Sigma(\mathbf{s}_{\min}))/\mathrm{tr}(\Sigma(\mathbf{s}_{\max}))$. For rewards where high-scoring images are highly concentrated (e.g., aesthetic rewards with strict compositional requirements), this ratio is large, providing significant controllability. For rewards with weaker concentration effects, the gap is smaller but typically still present. The assumption would fail for adversarial reward functions specifically designed to have uniform conditional variance, but such rewards are not encountered in practice.

### H.7.3. QUANTITATIVE COMPARISON OF COVERAGE

We now formalize the coverage gap between MIRO and DDPO using the concept of effective support.

**Definition H.11** (Effective Support). For a distribution $p$ and threshold $\tau > 0$, define the $\tau$-effective support as:

$$\mathcal{S}_\tau(p) = \{x : p(x) \geq \tau \cdot \sup_y p(y)\} \tag{114}$$

The effective support measure is $|\mathcal{S}_\tau(p)|$ (Lebesgue measure or counting measure as appropriate).

**Corollary H.12** (Coverage Gap). *For any $\tau > 0$ and $\beta < \infty$:*

$$|\mathcal{S}_\tau(p^*_{DDPO})| < |\mathcal{S}_\tau(p_{ref})| \tag{115}$$

*while for MIRO's marginal distribution:*

$$|\mathcal{S}_\tau(p^{marginal}_{MIRO})| = |\mathcal{S}_\tau(p_{data})| \tag{116}$$

*Proof.* For DDPO, the tilting $p^*_\theta \propto p_{\mathrm{ref}} \exp(r/\beta)$ amplifies density in high-reward regions and suppresses it in low-reward regions. Any point $x$ with $r(x) < r_{\max} - \beta \log(1/\tau)$ will have $p^*_\theta(x) < \tau \sup_y p^*_\theta(y)$, excluding it from the effective support. Since reward functions are typically non-uniform, this excludes a positive-measure set from $\mathcal{S}_\tau(p_{\mathrm{ref}})$.

For MIRO, the marginal $p^{\mathrm{marginal}}_{\mathrm{MIRO}} = \sum_{\mathbf{s}} p(\mathbf{s})p(x|c, \mathbf{s})$ places positive probability on every $x$ in the training support, since each $x$ belongs to some reward bin $\mathbf{s}$ with $p(\mathbf{s}) > 0$. Thus, the effective support is preserved. □

### H.7.4. ILLUSTRATIVE EXAMPLE: GAUSSIAN MIXTURE

To build intuition, consider a toy example where the data distribution is a mixture of Gaussians and the reward is a unimodal function.

**Example (Mode Collapse in Gaussian Mixture).** Let $p_{\mathrm{ref}}(x) = \frac{1}{K} \sum_{k=1}^K \mathcal{N}(x; \mu_k, \sigma^2 I)$ be a uniform mixture of $K$ Gaussians centered at $\mu_1, \ldots, \mu_K$. Let $r(x) = -\|x - \mu_1\|^2$ be a reward that prefers samples near mode $\mu_1$.

**DDPO:** The optimal distribution is:

$$p^*_\theta(x) \propto \sum_{k=1}^K \mathcal{N}(x; \mu_k, \sigma^2 I) \exp\left(-\frac{\|x - \mu_1\|^2}{\beta}\right) \tag{117}$$

As $\beta \to 0$, this concentrates entirely on mode $\mu_1$, discarding modes $\mu_2, \ldots, \mu_K$ entirely. The model loses the ability to generate diverse samples from the other $K - 1$ modes.

**MIRO:** Each mode $\mu_k$ is assigned a reward score $s_k = r(\mu_k) = -\|\mu_k - \mu_1\|^2$. Mode $\mu_1$ has $s_1 = 0$ (highest), while distant modes have lower scores. MIRO learns:

$$p_\theta(x|s = s_k) \approx \mathcal{N}(x; \mu_k, \sigma^2 I) \tag{118}$$

All $K$ modes are preserved. At inference:

- Conditioning on $s = s_1$ generates samples from mode $\mu_1$ (high quality)

- Conditioning on $s = s_k$ generates samples from mode $\mu_k$ (lower quality but diverse)

- The marginal recovers the full mixture

| Property | DDPO | MIRO |
|---|---|---|
| Number of objectives | 2 (conflicting) | 1 (unified) |
| Gradient variance | High (policy gradient) | Low (supervised) |
| Critical hyperparameters | $\beta$ (sensitive) | None at training |
| Mode collapse risk | High | Low |
| Distribution coverage | Narrows over training | Preserved |
| Multi-reward extension | Requires $N$ $\beta$ values | Natural conditioning |
| Inference-time control | Requires retraining | Adjustable via $\omega$ |

*Table 4.* Theoretical comparison of DDPO and MIRO training paradigms.

## H.8. Summary of Theoretical Advantages

These theoretical advantages explain MIRO's empirically observed stability: by reformulating reward alignment as conditional density estimation rather than constrained reward maximization, MIRO sidesteps the fundamental instabilities of RL-based approaches while achieving comparable or superior reward performance.

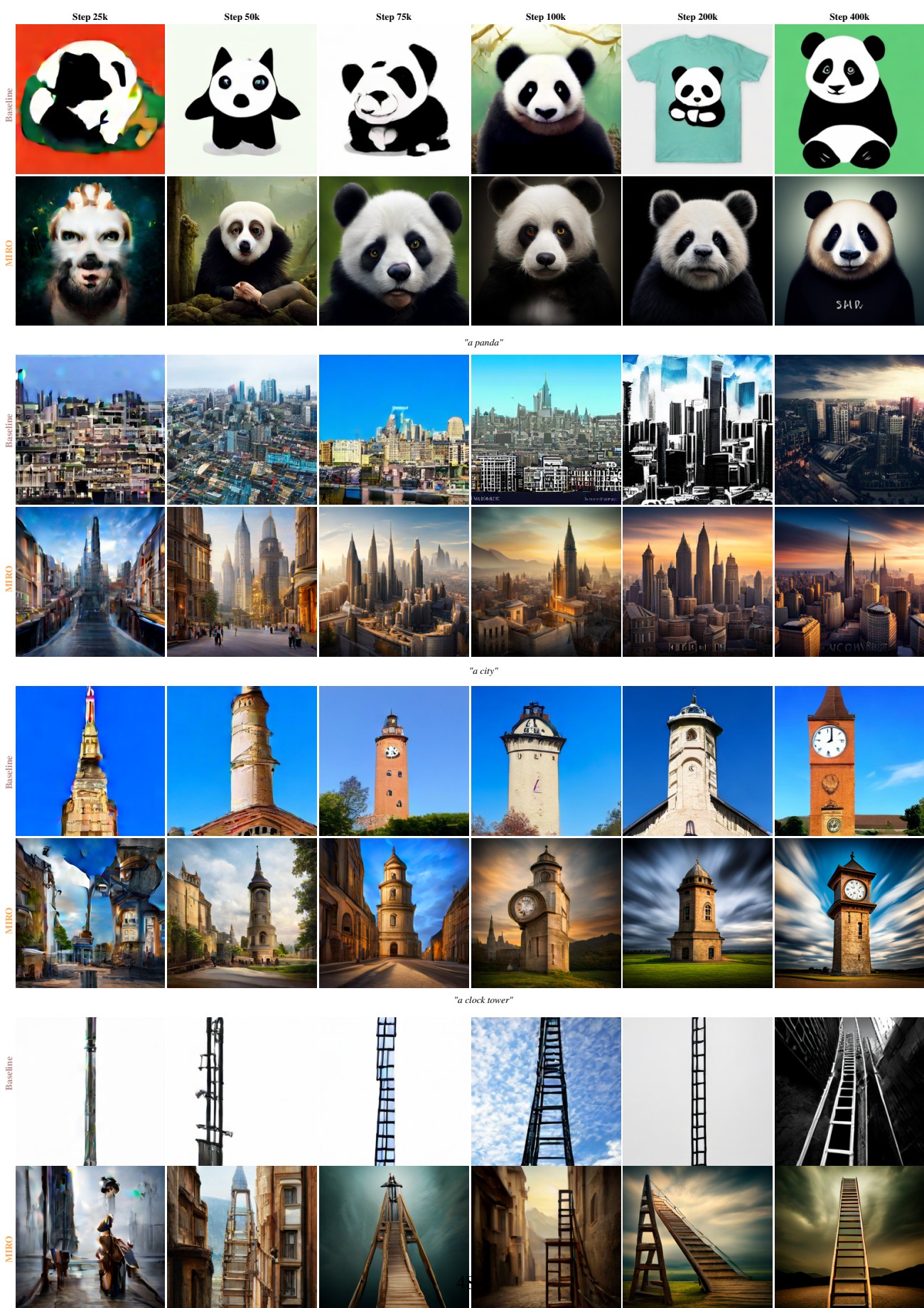

*Figure 20.* Additional training progression examples showing generated images at different training steps. Each row pair shows baseline (top) and MIRO (bottom) model outputs for the same prompt across training.

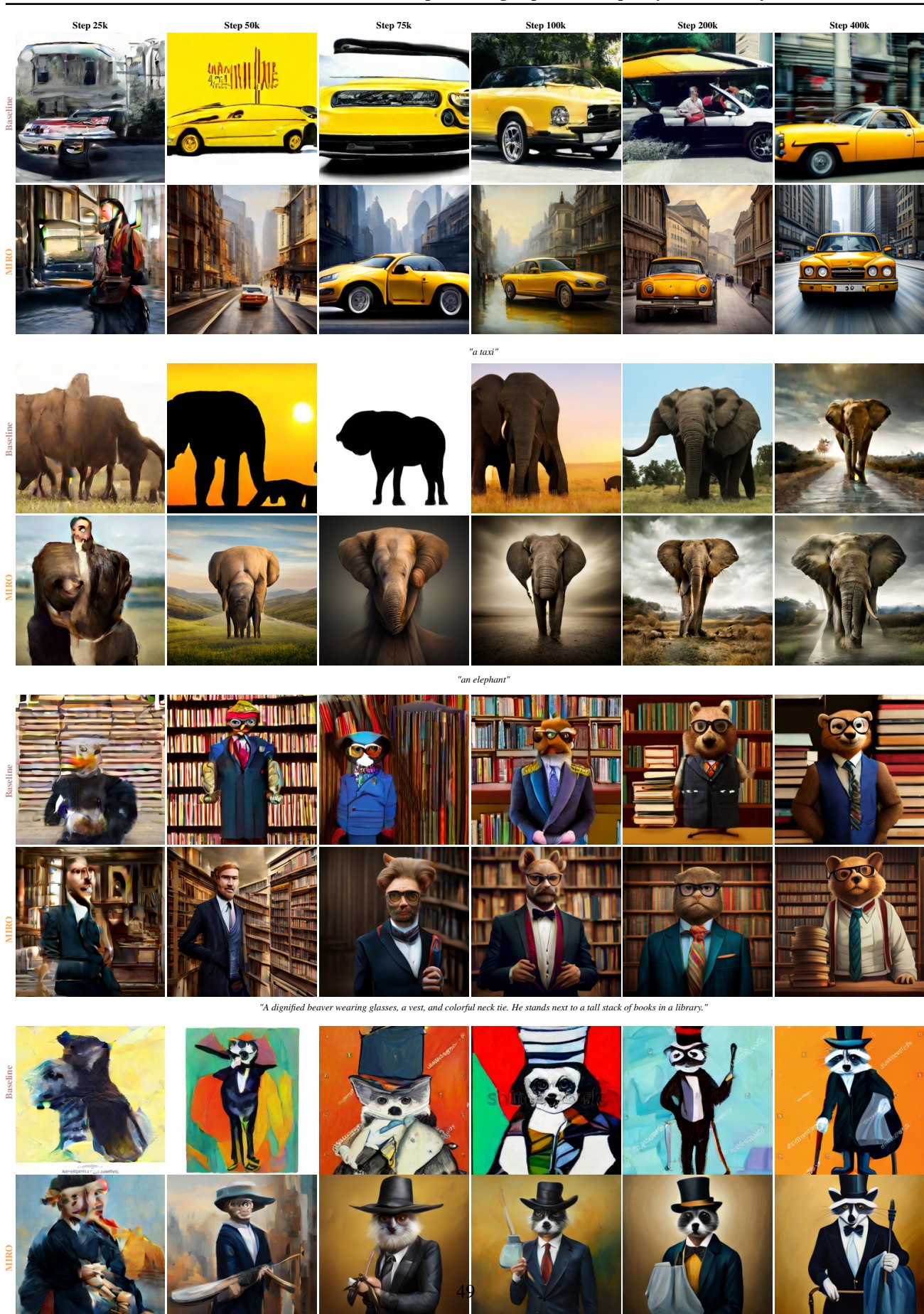

*Figure 21.* Additional training progression examples showing generated images at different training steps. Each row pair shows baseline (top) and MIRO (bottom) model outputs for the same prompt across training.