# OpenReview forum: "MIRO: MultI-Reward cOnditioned pretraining improves T2I quality and efficiency"
_ICML.cc/2026/Conference — ICML 2026 regular_

### Official Review · Reviewer_8ccP · 2026-03-08

**Soundness:** 3
**Presentation:** 3
**Significance:** 3
**Originality:** 3
**Overall Recommendation:** 4
**Confidence:** 3

**Summary:**

Overall, a significant challenge considered by this article is the inefficiency, reward hacking, and mode collapse in conventional post-hoc alignment pipelines for text-to-image generation. It proposes MIRO, a multi-reward conditioned pretraining framework that integrates diverse quality rewards directly into flow-matching pretraining, removing separate alignment stages and enabling flexible inference control. The research outlines a broad domain of reward-aware controllable generation; MIRO markedly improves generation quality and training/inference efficiency, surpassing large state-of-the-art models with far lower compute costs, backed by rigorous theory and experiments.

**Compliance With Llm Reviewing Policy:**

Affirmed.

**Final Justification:**

My concerns have been solved. I have updated my rating.

**Key Questions For Authors:**

see weaknesses

**Strengths And Weaknesses:**

**Strengths**

1. The work is rigorously grounded in both theory and experiments. It provides mathematically sound proofs for reward-tilted sampling, log-odds maximization, and the absence of gradient conflicts versus RL-based alignment (e.g., DDPO).

2. It resolves critical industrial bottlenecks: removing separate post-alignment stages, reducing data waste, and mitigating reward hacking and mode collapse. MIRO enables small models to surpass large SOTA models with lower inference compute and faster training, making it highly valuable for real-world deployment.

3. It innovates a multi-reward conditioned pretraining paradigm that integrates alignment directly into pretraining instead of post-hoc tuning.

4. MIRO preserves the full data distribution and avoids mode collapse by learning across all reward levels.

**Weaknesses**

1. MIRO is highly dependent on predefined reward models, so their biases, errors, and limited coverage are directly inherited.

2. The framework is a strong but incremental combination of existing techniques, with no wholly new fundamental theory or architecture. Theoretical extensions build on established flow-matching and guidance foundations.

---

> ### Author Rebuttal · Authors · 2026-03-31
>
> We thank the reviewer for acknowledging MIRO's rigorous theoretical grounding, practical impact on industrial bottlenecks, and its ability to preserve the full data distribution. We address the two weaknesses below.
>
> ---
>
> ## W1: Dependence on predefined reward models
>
> We agree that MIRO, like all alignment methods (RLHF, DPO, DDPO, reward-weighted regression), relies on predefined reward models. Mitigating the biases and errors inherent to reward models is an important but orthogonal research direction that applies equally to every alignment approach.
>
> That said, MIRO offers a **structural advantage** over single-reward methods in handling reward imperfections. Because MIRO conditions on *multiple* rewards simultaneously, it naturally mitigates reward hacking: optimizing one flawed reward cannot come at the expense of all others. Our experiments show that a single-reward AestheticScore model achieves 6.65 on Aesthetics but collapses to 0.17 on CLIP, while MIRO maintains balanced performance across all metrics (Figure 2). MIRO's multi-reward conditioning also gives users explicit control to detect and correct for biased rewards at inference time by adjusting individual reward levels. This capability is absent from standard RL-based alignment.
>
> In addition, we formally prove (Theorem 2.2, MIRO Diversity Preservation) that MIRO preserves the full data distribution across all reward levels, preventing the mode collapse that RL methods like DDPO provably suffer from (Theorem G.6, DDPO Mode Collapse). This diversity preservation further limits the impact of any single reward model's biases.
>
> Furthermore, MIRO's design is uniquely modular: upgrading a reward model requires only re-scoring the dataset (a cheap forward pass), with no architectural changes or retraining of other components. This contrasts with RL methods where the reward model is tightly coupled to the optimization loop.
>
> ## W2: Incrementality of contributions
>
> We respectfully disagree with the characterization of MIRO as incremental. While individual components (flow matching, reward conditioning) exist in prior work, MIRO **fundamentally changes how alignment is handled** in text-to-image generation in a theoretically grounded way:
>
> - **From post-hoc correction to pretraining integration.** The dominant methodology trains a model first and then aligns it via filtering and fine-tuning, possibly using RL. These complex stages lead to learning only part of the data distribution. On the contrary, MIRO learns the full reward-conditioned distribution by integrating the rewards during pretraining. This is a completely different optimization setup that leads to a much richer model, with the added benefit of being simpler to perform.
>
> - **Novel theoretical framework.** We provide new theoretical results that go beyond existing flow-matching and guidance foundations:
>   - *Theorem 2.1 (Guidance as Reward-Tilted Sampling)* proves that MIRO's multi-reward guidance samples from a reward-tilted distribution and that, under conditional independence, the log-odds decompose into a sum over individual rewards, naturally steering toward the Pareto frontier rather than collapsing to any single objective.
>   - *Theorem 2.2 (MIRO Diversity Preservation)* proves diversity preservation across the full quality spectrum, a property no RL-based method guarantees.
>   - *Appendix G (DDPO Mode Collapse)* formally establishes that DDPO suffers from gradient conflicts and mode collapse, providing the first rigorous comparison between conditioning-based and RL-based alignment.
>
> - **Empirical efficiency.** Simplicity of method does not imply incrementality of contribution. MIRO achieves a GenEval score of 68, outperforming FLUX-dev (67) (a 12B-parameter model) while using **370x less compute** (4.16 vs 1,540 TFLOPs). It converges up to **19x faster** than the unconditioned baseline. It improves compositional reasoning by +31% on Position and +11% on Color Attribution over prior state-of-the-art. These gains stem directly from the difference in reward modeling, not from incremental tuning.
>
> We believe the combination of an entirely new training procedure for integrating rewards into text-to-image generation, original theoretical guarantees, and state-of-the-art results with orders-of-magnitude efficiency gains constitutes a significant and non-incremental contribution. We note that simplicity of method does not imply incrementality — classifier-free guidance is also "just" subtracting two forward passes, yet it transformed the field. MIRO similarly combines known components in a way that produces outsized gains. Contributions should be judged by their outcomes, not by the novelty of individual components.

---

> > ### Author Rebuttal · Reviewer_8ccP · 2026-04-03
> >
> > My concerns have been addressed.

---

### Official Review · Reviewer_KCK7 · 2026-03-12

**Soundness:** 3
**Presentation:** 4
**Significance:** 3
**Originality:** 4
**Overall Recommendation:** 4
**Confidence:** 2

**Summary:**

This paper introduces MIRO, a novel framework for text-to-image (T2I) generation that integrates multiple reward signals directly into the pretraining phase, rather than relying on the standard post-hoc alignment pipeline. The core insight is to condition the flow-matching generative model on a vector of discretized reward scores during training, enabling the model to learn the mapping between reward levels and visual characteristics in an end-to-end manner. The authors demonstrate that a relatively small model trained with MIRO on a 16M-image dataset achieves state-of-the-art results on GenEval and user-preference metrics, outperforming much larger models such as FLUX-dev. Notably, the model converges significantly faster and offers greater computational efficiency.

**Compliance With Llm Reviewing Policy:**

Affirmed.

**Final Justification:**

My concerns have been addressed and I have updated the rating.

**Key Questions For Authors:**

Could you provide a more systematic justification for selecting the seven reward models? Have you tested combinations of non-overlapping rewards, and how do they perform compared to the current set?

Have you evaluated MIRO on larger datasets (e.g., LAION-5B) or noisier web data? If so, how does the model’s convergence speed, quality, and diversity hold up?

**Limitations:**

No. The authors only briefly acknowledge that reward models may amplify biases (e.g., related to beauty standards or cultural norms) but fail to provide in-depth discussion of limitations and potential negative societal impacts.

**Strengths And Weaknesses:**

Strengths:

The paper presents a compelling paradigm shift: moving reward alignment from a separate, often unstable post-hoc phase (e.g., reinforcement learning fine-tuning) to the core of the pretraining stage is a powerful and impactful concept.

The overall structure of the paper is clear and easy to follow, with extensive appendices providing sufficient details on implementation, ablation studies, and theoretical foundations.

The experimental results are impressive and form the core of the paper’s contribution. The claims of 19× faster convergence and a 370× inference efficiency gain over FLUX-dev are particularly significant. The comprehensive ablation studies are thorough, offering strong evidence for MIRO’s effectiveness and its ability to balance multiple objectives. Additionally, the paper goes beyond purely empirical claims by providing a solid theoretical framework in its appendices.

Weaknesses:
The paper, together with its Supplementary Material, is too long to review thoroughly.

While the paper employs multiple reward models, it provides insufficient justification for the specific set selected. There is no analysis of how overlapping or redundant rewards impact performance, nor an exploration of whether additional domain-specific rewards (e.g., ethical safety, accessibility) could enhance the model’s robustness.

MIRO requires users to manually adjust reward weights at inference (e.g., setting the aesthetic weight to 0.625 for optimal GenEval performance), which is impractical for non-expert users. The paper lacks any discussion of automatic weight optimization mechanisms.

Although the paper acknowledges that reward models may amplify biases (e.g., related to beauty standards or cultural norms), it offers no concrete debiasing strategies or empirical analysis of bias in MIRO’s outputs.

The chart in Section D.4 suffers from overlapping elements, making it difficult to discern the training dynamics clearly.

The comparison between MIRO and FLUX-dev raises fairness and interpretability concerns: a 0.36B model trained on 16M images vs. a 12B model on massive undisclosed data. The "370x less inference compute" claim, while technically true, is misleading given these scale differences. I think a fairer evaluation would apply MIRO to a modern architecture trained on identical data.

---

> ### Author Rebuttal · Authors · 2026-03-31
>
> We thank the reviewer for recognizing MIRO's paradigm shift, clear presentation, experimental results, and theoretical framework. We address each concern below.
>
> ---
>
> ## W1: Paper and supplementary material too long
>
> We appreciate the reviewer's effort and understand the reviewing burden. The main paper (10 pages) is fully self-contained; the appendix provides proofs and ablations for readers who want deeper verification. We provide a detailed table of contents at the beginning of the supplementary material and will add a reading guide in the revision, directing readers to the most relevant appendix sections for each main claim.
>
> ## W2 / Q1: Justification for the selected reward models / overlap analysis
>
> We selected rewards to cover complementary quality dimensions: aesthetics (AestheticScore), text-image alignment (CLIPScore), and human preferences (HPSv2, PickScore, ImageReward). Some rewards, particularly the human preference ones, are indeed highly correlated. Theorem 2.1 shows that correlated rewards simply contribute less marginal information; they do not hurt.
>
> Our **leave-one-out ablations** (Figures 15-16) confirm that removing *any* reward slightly lowers performance despite these correlations. The exception is AestheticScore, whose removal improves GenEval (+11.8 Position, +11.4 Two Objects) but degrades PickScore, HPSv2, and aesthetics (Table 2). This trade-off can be mitigated at inference by reducing the aesthetic weight (Figure 12: GenEval 74.9 at weight 0.625 vs 67.7 at 1.0). However, unless otherwise stated, we maintain the AestheticScore at its default value of 1.0 across all experiments.
>
> Exploring additional dimensions (safety, accessibility) is promising future work. MIRO is agnostic to reward choice, and we believe other alignment directions would integrate well.
>
> ## W3: Manual reward weight tuning impractical for non-experts
>
> MIRO works **out of the box** with default settings ($\hat{s}^+=1$), requiring zero tuning. Per-reward control is an *additional* capability for experts, not a requirement.
>
> This is a major advantage over RL-based alignment, where tuning the reward mixture requires **retraining from scratch** for each configuration. MIRO moves this entirely to inference time at negligible cost. Similarly to CFG, which also requires a manual scale, this is standard practice.
>
> Automatic weight optimization is an interesting complementary direction. MIRO makes this tractable for the first time: because reward trade-offs are controlled at inference, one could run a lightweight search (e.g., Bayesian optimization) over the reward weight vector without any retraining. This is not possible with RL-based methods where each configuration requires a full training run.
>
> ## W4: No concrete debiasing strategies
>
> Bias mitigation is an orthogonal research direction that applies equally to all alignment methods. MIRO provides a **structural advantage**: users can independently adjust each reward at inference to detect and attenuate biased rewards without retraining. This form of controllability is absent from RL-based alignment, where biases are baked into model weights. We expand on this in our response to Reviewer 8ccP, where we discuss how MIRO's multi-reward conditioning naturally mitigates the impact of any single biased reward.
>
> ## W5: Overlapping elements in Section D.4 chart
>
> We placed all curves on a single figure for compactness. We can split them into one figure per reward in the revision if the reviewer finds that clearer.
>
> ## W6: Fairness of MIRO vs FLUX-dev comparison
>
> The comparison is **deliberately asymmetric** and the asymmetry favors FLUX-dev, not MIRO: our 300M-parameter model trained on 16M images competes against a 12B model trained on orders of magnitude more data. That MIRO can match or outperform FLUX-dev on GenEval and user preference metrics speaks about the effectiveness of multi-reward conditioning.
>
> The **entire paper is built around fair, controlled comparisons**. Our core ablations (Sections 3.1-3.3) use identical architecture, data, and training budget, varying only reward conditioning. These controlled experiments are where our scientific claims originate. The FLUX-dev comparison provides supplementary context to situate MIRO in the broader landscape.
>
> We also note that a compact 300M model opens applications requiring fast inference (real-time generation, on-device deployment) where 12B models are simply not viable.
>
> ## Q2: Evaluation on larger datasets / noisier web data
>
> We train on CC12M + LAION Aesthetics 6+, totaling 16M images, which is representative of noisy web data. MIRO's reward conditioning should be even more beneficial at larger, noisier scales: instead of discarding low-quality data, MIRO learns how different reward levels manifest visually, extracting signal that ad-hoc filtering would discard. Scaling to larger datasets and higher resolutions is a natural and exciting next step.

---

> > ### Author Rebuttal · Reviewer_KCK7 · 2026-04-04
> >
> > For fair comparsion, directly applying MIRO to a comparable large-scale architecture with matched data would better validate your efficiency claims without scale asymmetry. Your rebuttal confirms manual weight tuning is optional, but there is no practical automatic optimization scheme for non-expert users, leaving some usability gap unaddressed. I would like to maintain the original score.

---

> > > ### Author Response · Authors · 2026-04-04
> > >
> > > We would like to clarify that our experiments were indeed designed to do exactly this. All of our comparisons are strictly controlled: the flow matching baseline and our proposed MIRO model are trained using the exact same architecture and data. Furthermore, all ablations presented in the paper share this identical setup. We intentionally designed our evaluation to ensure a rigorous, 'apples-to-apples' comparison during pretraining. This isolates the specific benefits of our approac, a level of controlled evaluation that is rare in current text-to-image literature.

---

### Official Review · Reviewer_q1SC · 2026-03-12

**Soundness:** 3
**Presentation:** 3
**Significance:** 2
**Originality:** 2
**Overall Recommendation:** 4
**Confidence:** 3

**Summary:**

This paper proposes MIRO, a multi-reward conditioned pretraining framework for text-to-image generation. Instead of applying alignment with a single reward model after pretraining, the method incorporates multiple reward signals directly into the training process by conditioning the generative model on a vector of reward scores associated with each image–text pair. The goal is to jointly optimize for multiple aspects of generation quality, such as aesthetics, preference, and semantic alignment, while also enabling controllable generation at inference time through reward-guided sampling. Experiments show improvements over several baselines on common T2I benchmarks and suggest faster convergence and improved sample efficiency under certain settings.

**Compliance With Llm Reviewing Policy:**

Affirmed.

**Key Questions For Authors:**

1. What are the key differences between the proposed method and existing reward-conditioned diffusion or coherence-aware training approaches? If the main change is extending a single reward signal to multiple rewards, it would be helpful to clarify what new capability this introduces from a modeling or optimization perspective.

2. If synthetic captions and inference-time scaling are removed, and the comparison is restricted to models trained with real captions under the same conditions, does MIRO still provide a clear advantage over the baseline? Providing such controlled comparisons would help isolate the effect of the proposed training framework.

3. The method requires computing multiple reward scores for large-scale datasets before training. Have the authors evaluated the practical cost of this offline computation? From an end-to-end training perspective, does the approach still offer an overall efficiency advantage?

4. Multiple reward models may be highly correlated. Have the authors analyzed the correlations among the reward signals and the individual contribution of each reward to the final performance? For instance, if only a subset of rewards is used, does the model still achieve most of the reported gains?

**Limitations:**

See mentioned above

**Strengths And Weaknesses:**

1. The core idea of the method is to incorporate multiple reward scores as conditioning signals during training. Conceptually, this is very close to existing reward-conditioned or coherence-aware training frameworks, and the paper does not clearly explain what fundamentally distinguishes MIRO from these prior approaches. At the moment, the method appears largely as an extension from a single reward to a vector of rewards rather than a fundamentally new training paradigm. The paper would benefit from a clearer discussion of why multi-reward conditioning introduces new capabilities beyond what previous reward-conditioned methods already provide.

2. The strongest performance results in the paper rely on additional components such as synthetic captions, reward tuning, and relatively strong inference-time scaling. These factors are presented together with MIRO, which makes it difficult to isolate how much of the improvement is actually due to the proposed training framework itself. For example, the largest gains reported on some benchmarks appear when synthetic captions are used, whereas improvements under the real-caption setting are more moderate. This makes it harder to assess the standalone contribution of the proposed method.

3. The paper repeatedly emphasizes improvements in training efficiency and inference efficiency, but the evaluation of efficiency is incomplete. In practice, the proposed approach requires computing several reward scores for the entire training dataset in advance, which could introduce a nontrivial additional computational cost. Moreover, the comparisons with other models are sometimes made across different model sizes or training setups, which makes it difficult to attribute efficiency gains solely to the proposed method. As a result, the claims regarding efficiency are not fully convincing without a more comprehensive cost analysis.

4. The theoretical analysis presented in the paper appears relatively weak in supporting the empirical claims. The results on reward-guided sampling and multi-reward guidance rely on strong assumptions, such as the model accurately learning the conditional distribution of images given text and reward conditions. In realistic training scenarios, these assumptions may not hold due to limited model capacity, discretization of reward scores, and potential noise in reward predictions. As a result, the theoretical section reads more as an intuitive explanation of the method rather than a rigorous justification of its effectiveness.

---

> ### Author Rebuttal · Authors · 2026-03-31
>
> We thank the reviewer for their detailed feedback. We address each concern below.
>
> ---
>
> ## W1 / Q1: MIRO vs existing reward-conditioned approaches
> The closest prior work is Don't Drop your Samples (CAD), which conditions on a single reward (CLIP score) during pretraining. We compare directly to CAD in Table 1: MIRO outperforms it on GenEval (57 vs 50 with real captions, 68 vs 50 with synthetic captions). While CAD and MIRO both integrate reward conditioning into pretraining, MIRO generalizes CAD from a single noisy conditioning signal to a structured multi-reward conditioning framework. This generalization is non-trivial and unlocks capabilities that do not exist in single-reward methods like the following.
>
> - **(a) Richer supervision signal:** Multiple reward conditioning provides much richer and denser supervision. It allows for far more controllable exploration of the search space, which explains the superior performance over all single reward baselines (Table 1).
> - **(b) Prevention of reward hacking:** With multiple rewards to optimize, hacking a single reward for better performance is no longer possible as the model needs to learn to balance all the rewards properly to lower its loss function. This can be clearly seen from Theorem 2.1 that under conditional independence, the log-odds decompose into a sum over individual rewards, naturally steering toward the Pareto frontier rather than collapsing to extremes of any single reward.
>
> ## W2 / Q2: Isolating MIRO's contribution from synthetic captions and inference scaling
>
> Unless explicitly stated, all our experiments use real captions and standard inference (no scaling). The 19x convergence speedup on AestheticScore, the reward hacking analysis, and the training curves (Figures 3-4) are all obtained without any of these additions.
>
> Table 1 provides a controlled comparison across all configurations. Two key observations:
>
> - **MIRO without synthetic captions** (GenEval 57) already matches the **baseline with synthetic captions** (GenEval 57). This shows that multi-reward conditioning alone recovers the benefit of expensive recaptioning.
> - **MIRO + synth** (GenEval 68) vs **baseline + synth** (GenEval 57): the gain from adding MIRO (+11) is much larger than the gain from adding synthetic captions to the baseline (+5). The two improvements compose, but MIRO is the dominant factor.
>
> Synthetic captions and inference scaling are orthogonal techniques that any method could use. We always compare apples to apples in Table 1. Notably, adding synth captions to the baseline yields +5 on GenEval, while adding them to MIRO yields +11. This suggests that multi-reward conditioning appears to unlock far greater gains from higher-quality captions that standard training cannot exploit.
>
> ## W3 / Q3: Cost of offline reward computation
>
> Appendix E quantifies preprocessing costs. Computing all reward scores for 16M images takes approximately 25% of the total preprocessing budget (Figure 19). Without VQA, this drops to 11% (131 H100 GPU hours). This is a small fraction of the training cost of a single model, and it is computed once: the same reward annotations can be reused to train as many models as needed with zero additional cost.
>
> We also note (Section 3.3) that reward scoring requires substantially less compute than recaptioning with large vision-language models, which accounts for over 55% of the preprocessing budget.
>
> ## W4: Theoretical assumptions
>
> The assumption that the model accurately learns the conditional distribution is standard in the classifier guidance (Dhariwal & Nichol, 2021) and CFG (Ho & Salimans, 2022) literature. Our contribution is showing that under these standard assumptions, multi-reward conditioning has principled theoretical properties confirmed empirically. Regarding reward correlations, we progressively relax assumptions throughout our analysis: we first derive results under conditional independence (Theorem 2.1), then analyze the effect of positive correlations (Proposition F.8), and finally provide results in orthogonal coordinates for arbitrary correlation structures (Proposition F.9). This gives a range of results depending on the actual properties of the rewards.
>
> We view the theoretical framework as providing an intuitive explanation of why MIRO works (Pareto steering via log-odds, diversity preservation) rather than as formal proof that it must work under all conditions. The empirical results confirm that these predictions hold in practice.
>
> ## Q4: Reward correlations and individual contributions
>
> We analyze this in detail in our response to Reviewer KCK7, where we present leave-one-out ablations (Figures 15-16) showing that removing any reward slightly lowers performance despite correlations, and discuss the specific trade-offs per reward. Theorem 2.1 and the extended analysis in Appendix F (Propositions F.8-F.9) show that correlated rewards naturally contribute less marginal information but do not hurt.

---

> > ### Author Rebuttal · Reviewer_q1SC · 2026-04-04
> >
> > The author addressed my concerns and I maintained my score.

---

### Official Review · Reviewer_zm8Z · 2026-03-22

**Soundness:** 2
**Presentation:** 3
**Significance:** 3
**Originality:** 2
**Overall Recommendation:** 4
**Confidence:** 4

**Summary:**

This paper proposes MIRO, a framework that conditions text-to-image flow matching models on multiple reward scores during pretraining rather than applying reward-based alignment as a post-hoc stage. Each training image is annotated with scores from seven reward models (aesthetic, preference, CLIP-based, etc.), which are discretized into bins and fed as additional conditioning tokens. At inference, users can set reward targets to control generation quality along multiple dimensions. The authors report faster convergence (up to 19×), strong GenEval scores, and competitive performance against much larger models like FLUX-dev.

**Compliance With Llm Reviewing Policy:**

Affirmed.

**Key Questions For Authors:**

It's better to show more clearly what the base model that this paper adopted is. No other questions

**Limitations:**

No. See weaknesses.

**Strengths And Weaknesses:**

**Strengths**

- **Simple and well-motivated idea**. The core proposal — conditioning on reward vectors during pretraining instead of filtering data or running a separate RLHF stage — is clean and elegant. It naturally extends prior work on CAD from a single score to multiple rewards. The connection to upside-down RL and Decision Transformers is apt and well-articulated.
- **Comprehensive experimental design**. The paper includes a thorough set of comparisons: baseline vs. single-reward vs. multi-reward, leave-one-out ablations, binning strategy ablations, post-training experiments, CFG sensitivity analysis, and test-time scaling curves. The inclusion of OpenAI CLIP as an out-of-distribution metric is a good choice.
- **Controllability at inference**. Extending the reward to multiple dimensions, it allows the finegrained control over each dimension.

**Weaknesses**
- **The scalability of the reward model**. The model is pre-trained with only 16M images with a 0.36B sized model at 256pix, which is a relative low experiment. The reward model is conventionally used only at post-training (RL phase), a phase with much fewer images compared with pre-training images, allowing engineers to try different reward models. However, during MIRA's pretraining stage, once the reward model is updated, the model must be retrained. The bin strategy must be recalibrated whenever a new reward model is introduced. So I am questioning the extensibility to a new reward model for this paper at a large scale pretraning.
- **multi-reward conditioning mitigates reward hacking** is supported by Figure 2 (the AestheticScore-only model degrades other metrics). However, reward hacking mitigation via multi-objective optimization is a well-known phenomenon from prior RL literature. The paper could cite this more explicitly.
- **Reward models only see short caption**. The paper acknowledges this in Section 3.3: some reward models can't handle captions longer than 77 tokens, so they generate separate short captions for reward scoring while training with long synthetic captions. This creates a potential mismatch — the reward scores are computed from simplified image descriptions, not the detailed captions used for training. I am curious whether the short caption only will be detrimental?

---

> ### Author Rebuttal · Authors · 2026-03-31
>
> We thank the reviewer for recognizing MIRO as a simple and well-motivated idea with comprehensive experimental design and controllability at inference. We address each concern below.
>
> ---
>
> ## W1: Scalability
>
> We acknowledge that our experiments use a 0.36B model at 256px on 16M images, which is a controlled research setting. MIRO adds reward tokens by concatenating the reward score tokens with text embeddings. It is architecture-agnostic and introduces zero overhead at scale. Validating this at larger scale is future work. That said, even at this scale MIRO already outperforms much larger and more heavily resourced models: it beats FLUX-dev (12B parameters, trained on massive undisclosed data) on GenEval (68 vs 67) while using 370x less compute (Table 1). This suggests the approach would only benefit from further scaling.
>
> On the concern about adding new reward models: as we show in Appendix D.7, MIRO also works as a fine-tuning step with slightly lower performance than full pretraining. One can explore new rewards via fine-tuning without retraining from scratch. Once a good set is identified, full pretraining can maximize performance.
>
> More broadly, our theoretical results (Theorem 2.1) and leave-one-out ablations (Figures 15-16) show that adding more rewards mostly has positive gains, even when they are correlated with existing ones. Correlated rewards contribute less marginal information but do not hurt (see also Appendix F, Propositions F.8-F.9). This makes the reward selection process forgiving, suggesting that one does not need to carefully engineer non-overlapping rewards.
>
> MIRO also requires much less reward tuning at train time compared to RL post-training. With RL methods, each reward configuration requires a full training run with careful hyperparameter tuning (KL coefficient, reward weighting, learning rate). With MIRO, the trade-offs between rewards are controlled entirely at inference with negligible compute. The bins are computed independently per reward model, so introducing a new reward does not change the binning of any existing reward.
>
> Regarding the binning recalibration concern specifically: quantile binning (our default, Appendix D.6) adapts automatically to the score distribution of each reward. It requires no manual threshold selection and only a forward pass through the reward model on the training set, which as shown in Appendix E represents a small fraction of the total preprocessing budget.
>
> ## W2: Reward hacking mitigation is known in multi-objective RL
>
> We agree that reward hacking mitigation via multi-objective optimization is documented in the RL literature (e.g., Rewarded Soups, Rame et al. 2023; Parrot, Lee et al. 2024), and we will add more explicit citations in the revision. The key difference is that in RL, multi-objective optimization is notoriously difficult: it requires careful reward weighting, suffers from gradient conflicts between objectives (which we formally prove in Appendix G, Proposition G.1), and often demands expensive Pareto search procedures. MIRO sidesteps all of these issues. The model simply learns the joint reward-conditioned data distribution during training with a single supervised objective, and the multi-objective trade-offs emerge naturally at inference through the reward conditioning vector. There is no multi-objective optimization loop at training time.
>
> ## W3: Reward models only see short captions
>
> This reflects a current limitation of publicly available reward models, which have a 77-token input limit. We generate short captions specifically for reward scoring while using the full long captions for training (Section 3.3). Using reward models that handle longer captions would allow more granular and context-aware reward signals, and is a very interesting direction for future work as such models become available.
>
> In practice, short captions capture the core semantics of the image (objects, attributes, spatial relationships), which is what most reward models are designed to evaluate. Our results with synthetic captions (GenEval 68 in Table 1, outperforming FLUX-dev) show that this mismatch does not prevent strong performance. We also note that rewards like AestheticScore and ImageReward primarily evaluate visual quality and are less sensitive to caption length, so the mismatch mainly concerns text-image alignment rewards (CLIPScore).
>
> ## Q: Base model description
>
> To clarify: MIRO does not use a pretrained base model. We train from scratch with reward conditioning integrated from the very first training step. This is a central design choice and part of the paradigm shift we propose: alignment is not a correction applied to a pretrained model, but a native part of the training objective. The architecture is based on TextRIN with SwiGLU, RMSNorm, QK-Norm, and flow matching, detailed in Appendix C (Implementation Details). We will make the architecture description more prominent in the main text in the revision.

---

> > ### Author Rebuttal · Reviewer_zm8Z · 2026-04-07
> >
> > Thanks for the rebuttal. It fully solves my concerns.

---

### Decision · Program_Chairs · 2026-04-30

**Decision:**

Accept (regular)

**Comment:**

All four reviewers gave a Weak Accept (rating 4) for this paper. The paper introduces a new framework that integrates multiple quality rewards directly into the pretraining phase of text-to-image flow matching models, effectively replacing inefficient post-hoc alignment stages. The work has clean motivation and comprehensive experimental validation, which demonstrate faster training convergence and effective mitigation of reward hacking. Reviewers initially raised some concerns regarding the method's reliance on predefined reward models and its scalability, and the authors provided a rebuttal that addressed these issues by highlighting MIRO's ability to balance performance across metrics and avoid the pitfalls of single-reward optimization. Ultimately, the method successfully resolves critical bottlenecks in the generation pipeline and achieves results competitive with larger state-of-the-art models, and the AC recommends the acceptance of the work.